



# Inventory, motion and acceleration of rock glaciers in Ile Alatau and Kungöy Ala-Too, northern Tien Shan, since the 1950s

Andreas Kääb[1], Tazio Strozzi[2], Tobias Bolch[3], Rafael Caduff[2], Håkon Trefall[1], Markus Stoffel[4,5,6], Alexander Kokarev[7]

[1] Department of Geosciences, University of Oslo, Oslo, 0316, Norway
[2] GAMMA Remote Sensing, Gümligen, Switzerland
[3] Geography and Sustainable Development, University of St Andrews, Scotland, UK
[4] Climatic Change Impacts and Risks in the Anthropocene (C-CIA), Institute for Environmental Sciences, University of Geneva, Switzerland
[5] Department of Earth Sciences, University of Geneva, Switzerland
[6] Department F.-A. Forel for Environmental and Aquatic Sciences, University of Geneva, Switzerland
[7] Institute of Geography of the Republic of Kazakhstan, 050010 Almaty, Kazakhstan

*Correspondence to*: Andreas Kääb (kaeaeb@geo.uio.no)

**Abstract.** Spatio-temporal patterns of rock glacier creep have rarely been studied outside the densely populated European Alps. This study investigates the spatial and temporal variability of rock glacier motion in the Ile Alatau and Kungöy Ala-Too mountain ranges, northern Tien Shan. Over the study region of more than 3000 km$^2$, an inventory of slope movements is constructed using a large number of radar interferograms and high-resolution optical imagery. The inventory includes more than 900 landforms, of which around 550 are interpreted as rock glaciers. Out of the active rock glaciers, 45 are characterised by a rate of motion exceeding 1 m/a. From these fast rock glaciers we select six and study them in more detail (Gorodetzky, Morenny, Archaly, Ordzhonikidze, Karakoram and Kugalan Tash rock glaciers) using offset tracking between airphotos, and historical and modern very high resolution optical satellite data. Most of them show an overall increase of decadal surface velocities from the 1950s onwards with speeds being roughly two to three times higher in recent years compared to the 1950s and 1960s. This development indicates a possible significant increase in sediment and ice fluxes through rock glaciers and implies that – when compared to glacier shrinkage – periglacial sediment transport in the region seems to gain importance relative to glacial sediment transport. Those rock glacier fronts reaching the valley floors show a strongly compressive flow regime, and changes in speeds further upstream affect them only in a damped way. The only rock glacier investigated in detail that does not exhibit an overall increase in speed since the 1950s is Gorodetsky where glacier retreat and dead-ice degradation seem to have decoupled the rock glacier from its supply by glacial sediments and ice.



# 1 Introduction

Frozen debris on mountain slopes can accumulate, deform under gravity, and form striking tongue-shaped landforms, up to a kilometre wide and several kilometres long – so called rock glaciers (or rock glaciers) (Haeberli, 1985; Martin and Whalley, 1987; Barsch, 1996). Rock glaciers can be defined as *"the visible expression of cumulative deformation by long-term creep of ice/debris mixtures under permafrost conditions"* (Berthling, 2011), a definition that we will follow in this study. Other definitions are based more on the morphology of the features, irrespective of whether they are connected to frozen ground, glacial activity, or even rock avalanches (Whalley and Azizi, 2003). Prominent rock glaciers are found in most cold mountain environments on Earth (Barsch, 1996; Jones et al., 2018). Their debris content typically stems directly from rock walls situated above them, but debris can also be of glacial origin. In a similar way, the ice content, enabling rock glaciers to deform, is directly incorporated from glaciers; alternatively it can evolve from freezing liquid precipitation or melt water (Humlum, 1998). The long-term preservation of ice content in rock glaciers under permafrost conditions, situated under a rocky active layer, enable these landforms to creep over time periods of thousands of years (Haeberli et al., 1999; Krainer et al., 2015). The thermal conditions below the freezing point and the ice content of rock glaciers make their creep inherently sensitive to climatic conditions and it is able to change over different time scales (Kääb et al., 2007; Delaloye et al., 2010; Sorg et al., 2015). Besides their climatic and geomorphic significance, rock glaciers can also be the source of rockfall and debris flows, and thus evolve into local-scale natural hazards (Stoffel and Huggel, 2012; Deline et al., 2015; Schoeneich et al., 2015). Some rock glaciers in the European Alps were even found to destabilise (Kaufmann and Ladstädter, 2008; Roer et al., 2008; Scapozza et al., 2014; Bodin et al., 2016; Kummert et al., 2017; Scotti et al., 2017).

Besides rock glacier distribution, rock glacier creep speed is considered a valuable indicator of environmental, in particular of climatic and ground-thermal conditions. Rock glacier creep speed is expected to increase with ground temperature (Kääb et al., 2007; Arenson et al., 2015; Müller et al., 2016), although it will also be influenced substantially by a number of other factors such as topography (e.g., slope), provision of ice and debris, temporal and vertical variations in ice content, rheology of the frozen debris, thickness, as well as advection or internal production of water (Jansen and Hergarten, 2006; Cicoira et al., 2019; Kenner et al., 2019). The response of rock glacier creep speeds to rising ground temperatures is expected to be especially variable around thawing conditions, i.e. in cases in which ice content starts to degrade or melt, or in cases where advected water plays a major role (Kääb et al., 2007; Delaloye et al., 2010; Arenson et al., 2015; Hartl et al., 2016).

From the above it becomes clear that inventorying contemporary rock glacier motion and monitoring rock glacier speed over time may contribute substantially to a better understanding of the climatic and geomorphic significance of rock glacier creep, its response to climatic changes, and impacts from spatio-temporal variations of rock glacier creep. Inventorying rock glacier motion seems currently best done using satellite radar interferometry. This method enables detecting slow slope movement, not least rock glacier motion, over large regions and hundreds of individual landforms (Villarroel et al., 2018). The approach has been used to develop rock glacier motion inventories for the Swiss Alps (Strozzi et al., 2004; Barboux et al., 2014), Sierra Nevada, California (Liu et al., 2013), Northern Iceland (Lilleoren et al., 2013), Brooks Range, Alaska (Rick et al., 2015),



Carpathians, Romania (Necsoiu et al., 2016), north-eastern Tien Shan, Xinjiang/China (Wang et al., 2017), and the dry Andes of Argentina and Chile (Villarroel et al., 2018). These studies mainly relied on satellite L-band synthetic aperture radar (SAR) data such as from the JERS and PALSAR instruments, and C-band data such as from ERS and Sentinel-1. In addition to measured displacement rates, the categorisation of landforms into motion magnitudes (cm/day, cm/month, dm/month, cm/year,

etc.) has been proven useful (Barboux et al., 2014).

Rock glacier speed was shown to vary at a range of time scales, including seasonal, annual, decennial, or centennial (Delaloye et al., 2010; Scapozza et al., 2014). For a few rock glaciers worldwide, mostly concentrated in the European Alps, important time series exist with seasonal to decennial time resolutions (more details below). Most of these measurements are based on terrestrial geodetic surveys. This implies that such time series cannot be established retrospectively for locations that have not

been monitored in the past. Satellite radar interferometry is mainly available for 21$^{st}$-century rock glacier creep velocities. The only means to measure 20$^{th}$-century rock glacier surface velocities at yet unmeasured locations and to reconstruct rock glacier speed variations at climatic time scales (i.e. at roughly decennial time scales) and back into the last century is by offset tracking in repeat historic and modern air and satellite imagery (Kääb et al., 2007; Kaufmann, 2012; Sorg et al., 2015; Monnier and Kinnard, 2017).

A limited number of studies that have presented data on decennial variations in rock glacier speed in the past have drawn a contrasting picture of decadal-scale increases in speed, but also reported decreases or phases of decrease, as well as episodes of stable creep speeds (Schneider and Schneider, 2001; Janke, 2005; Krainer and He, 2006; Bodin et al., 2009; Delaloye et al., 2010; Kellerer-Pirklbauer and Kaufmann, 2012; Hartl et al., 2016; PERMOS, 2016; Eriksen et al., 2018). In some of these cases, the variations in creep velocity followed decadal variations in mean annual air temperatures. All these studies underline

the influence of variations in mean annual air or ground temperature on rock glacier speed, through mechanisms such as heat conduction or melt water advection (Ikeda et al., 2008; Kenner et al., 2017; Kenner et al., 2019). Decrease in speed was also shown to coincide with a significant surface lowering, thus hinting to a degradation of ice-rich permafrost (Bodin et al., 2009). Again, by far most of the above few studies on decennial variations in rock glacier speed stem from the European Alps.

The aim of the present study is to inventory, for the first time, contemporary rock glacier creep velocities over two mountain

ranges in the northern Tien Shan and to reconstruct speed changes since the mid-20$^{th}$ century for a few selected landforms, with the specific goal to eventually investigate spatio-temporal variations in rock glacier creep for a region outside the European Alps. After introducing the study area and the substantial body of permafrost/rock glacier-related work already available for it, we explain the data and methods used in the present study – satellite radar interferometry and offset tracking between repeat optical data. After presenting the corresponding results, we discuss and combine them towards overall

conclusions.



## 2 Study area and previous work

The mountain ranges of Ile Alatau and Kungöy Ala-Too (former names: Zailijskij and Kungej Alatau) are located at the border of Kyrgyzstan and Kazakhstan, Central Asia (Fig. 1). They rise from the Kazakh steppe to the north and Lake Isyk-Köl (Issyk-Kul, 1608 m asl) to the south up to an elevation of almost 5000 m asl. Ongoing tectonic activity has caused many earthquakes with the largest of magnitudes 8.0 and more (Yadav and Kulieshius, 1992; Lukk et al., 1995; Tibaldi and Graziotto, 1997), and stronger earthquakes have triggered a large number of landslides and rock avalanches (Delevaux et al., 2001; Korjenkov et al., 2004; Strom and Korup, 2006). The dominant lithology of the mountain ridges is granite, but weaker schists build the lower-lying saddles as well as larger parts of the valleys.

The climate of the study region is continental with mean annual precipitation exceeding 1000 mm on the northern slopes and less than 800 mm on the southern ridges and for elevations of about 3000 m asl (Bolch, 2007; Narama et al., 2010). Minimum precipitation is typically recorded during winter under the Siberian High. Maximum precipitation is found for spring and early summer. The zero-degree isotherm is estimated at slightly above 2700 m asl. (Bolch, 2007). Ground temperature measurements showed that permafrost occurrence is likely above 3200 m asl. (Gorbunov, 1996). Air temperatures have been increasing since the 1970s with annual rates of about 0.01 °C, and higher rates during late summer and winter (Bolch, 2007; Marchenko et al., 2007; Narama et al., 2010; Sorg et al., 2012). Ground temperature measurements at about 3300 m asl. show an increase between 0.3 °C and 0.6 °C from the 1970s to 2004 (Marchenko et al., 2007). All rock glaciers investigated here in more detail using photogrammetry (Gorodetzky, Morenny, Archaly, Ordzhonikidze, Karakoram and Kugalan Tash) originate in areas where permafrost is very likely whereas the fronts of the larger rock glaciers are located at elevations slightly below the lower boundary of regional permafrost occurrence (Bolch and Gorbunov, 2014). In the hemispheric-scale, 1km-resolution permafrost model by Obu et al. (2019), all these rock glaciers have permafrost probabilities > 0.5, and modelled mean annual ground temperatures (MAGT) between -0 and -2 °C. In line with the results from Bolch and Gorbunov (2014), the model indicates that Karakoram rock glacier reaches down into non-permafrost terrain (MAGT ~+1.4 °C).

Rock glaciers are widespread in the study area, and some are remarkably large with areas exceeding 2 km². A first rock glacier inventory was compiled based on aerial photography from the 1970s. More than 1000 rock glaciers (amongst about 180 inactive ones) were identified in Ile Alatau and Kungöy Ala-Too covering a total area of about 90 km² (Gorbunov and Titkov, 1989; Gorbunov et al., 1998). The average altitude of the active rock glaciers is 3400 m asl., the lowest elevation of an active rock glacier front is less than 2700 m asl. Most rock glaciers in Ile Alatau are on north-facing slopes, whereas in Kungöy Ala Too the southern aspect prevails. The highest rock glacier density is found in the central and western part of Ile Alatau as well as in some north-facing side branches of the Chon-Kemin and Chilik valleys where the percentage of glaciated terrain is 10-20% (Titkov, 1988; Gorbunov and Titkov, 1989). Rock glacier density decreases to the outer parts of the mountain ranges due to decreasing elevations of the ridges. In a more recent inventory, Bolch and Gorbunov (2014) identified 72 rock glaciers in the central part of Ile and Kungöy Ala Too based on Landsat ETM+ imagery and aided by aerial images as well as high-resolution satellite images from GoogleEarth. Blöthe et al. (2019) map rock glaciers over High Mountain Asia, including our



study region, with the purpose to statistically model and detect those landforms that divert, block or else impact on river channels. One of our study rock glaciers, Ordzhonikidze, is explicitly mentioned in their study, but also Karakoram and Kugalan Tash rock glaciers studied here clearly divert the river channels at their snout.

Displacement measurements of larger boulders on few larger rock glaciers using ground surveys and aerial imagery from 1969, 1979 and 1994 revealed annual speeds between less than 1m/a to extraordinarily high rates of more than 11 m/a (Gorbunov et al., 1992). On average, there was an increase of the surface speed found between the periods 1969 – 1979 and 1979 – 1994, except for the Karakorum rock glacier, which is at the lowest elevation. Displacement rates of rocks vary greatly within rock glaciers and there are some areas where latest measurements show inactivity (Gorbunov et al., 1992; Kokarev et al., 1997) Most of the measured rocks showed displacements in down-valley directions, but those rock glaciers reaching the valley floor also showed lateral displacement, pointing to a spread onto the valley floor. It was suggested that some rock glaciers in Tien Shan are subject to rapid advances, maybe caused by earthquakes. This might be the reason why some rock glaciers have been shown to advance into forested areas (Gorbunov, 1983). Four of the rock glaciers studied in Gorbunov et al. (1992) are also investigated in detail in this study, and results are compared in the discussion section of this contribution.

Sorg et al. (2015) used tree-ring growth anomalies from 280 trees growing on rock glaciers in the study area, or close to it, as an indicator for rock glacier activity and compared their results to climatic data and detailed glacier mass balances for the period 1895 – 2011. They find that the activity of the rock glaciers studied correlates well with decadal variations of summer air temperatures and glacier mass balance – rock glacier activity/tree growth anomalies are positively linked to high summer temperatures, or negative glacier mass balances, respectively. The authors also use preliminary offset tracking between repeat aerial and satellite imagery to show that on a multi-decadal scale the variations in rock glacier speeds show more contrasting and site-specific trends, with accelerations and decelerations, both varying over time and even within the same rock glacier system. They attribute this behaviour partially to different statuses of the rock glaciers on their path towards inactivation under continued atmospheric warming in the region. It is the offset tracking work started in Sorg et al. (2015) that we extend in the second part of the present study in terms of data and time steps, number of rock glaciers, and methods used. The specific rock glaciers studied to this end are Morenny, Archaly, Gorodetzky, Ordzhonokidze, Kugalan Tash and Karakorum (Fig. 1). More details about these rock glaciers are introduced under the results section for the offset tracking work realized here.

## 3 Data and methods

### 3.1 Radar interferometry

An inventory of rock glaciers and other periglacial processes in the Ile and Kungöy Ranges of Northern Tien Shan was in this study compiled by visual analysis of differential SAR interferograms and satellite optical images. We computed differential SAR interferograms (DInSAR) with short baselines and time intervals between 1 day and one year from the ERS-1/2 tandem mission (1998-1999), ALOS-1 PALSAR-1 (2006-2010), ALOS-2 PALSAR-2 (2014-2016) and Sentinel-1 (2015-2018, Table A1). Images acquired along both ascending and descending geometries and during summer (snow-free) and winter (frozen



snow) conditions were employed. For topographic reference and orthorectification, we computed a DEM from TanDEM-X data at a spatial resolution of 10 m. Bistatic TanDEM-X acquisitions along ascending (13 Jan 2012 and 24 Jan 2012) and descending (15 Dec 2013 and 6 Jan 2014) orbits were combined to reduce problems of areas masked by layover and shadow, which differently affect opposite slopes along north-south oriented valleys in mountainous regions (Ambrosi et al., 2018). In general, we estimate the errors due to atmospheric distortions and noise in the DInSAR images to be in the order of ¼ of a phase cycle (Strozzi et al., 2001). All processing was done using the GAMMA radar software.

By combining interpretation of all interferograms, signals related to the motion of periglacial phenomena over various time intervals were first identified from the wrapped interferometric signals following the recommendations given in Barboux et al. (2014) and marked as point features. In a second step, the extent of the InSAR movement signals was mapped based on the interferograms, and in addition supported by optical high-resolution imagery from GoogleEarth/Map and Bing Maps. Next, the deformation rates of the mapped landforms were classified using five classes (i.e. 0-2, 2-10, 10-50, 50-100 and >100 cm/a). In cases where the estimated velocity was close to the upper boundary of a velocity class, the polygon was assigned to the faster class. The same was done with natural variations of surface displacement rates, i.e. if two or more classes were present during the observation time-span, the higher displacement rate was used to determine the velocity class. Because the interferometry-derived displacement rates are measured along the satellite line-of-sight, we empirically considered the line of sight in the assessment of the state of activity of the mapped landforms. In addition to the rock glaciers as main objective of this study, movements related to solifluction / debris movements, dead ice / subsidence, debris-covered glaciers and ice-cored moraines were classified.

In this final step of our analysis, the specific interferometric fringe pattern, optical images and the shaded relief of the DEM were visually interpreted to classify the process types associated to the observed slope movements. High-resolution satellite images were the main source for the classification in one of the above classes. In cases for missing information due to low resolution or cloud cover in the images, the interferograms were consulted for the interpretation. E.g. rock glaciers usually show an increase of the velocity to the inner region of the mapped polygon, have a lobe-structure and clear fronts. The temporal signature of the rock glacier velocity is usually not affected by short term changes. In contrast, debris-covered glaciers show stronger seasonal variations in speed, and thermokarstic features such as supraglacial ponds, where, in addition, ice is sometimes visible in high-resolution optical images. Melting dead ice bodies have characteristic spot-like fringe patterns and are visible only in late summer interferograms. Similar appearance in the interferograms can be observed for debris movements (soli-/gelifluction). This process type is typically bound to inclined slopes but lacks large-scale compressional features (lobes). Ice-cored moraines are bound to the former lateral or terminal moraines of a retreated glacier. For cases where the InSAR signal could not be unambiguously assigned to a process type and also optical images were not of help due to a lack of good quality high-resolution optical images, for instance, an "undefined" class was introduced.

Above the maximum inventory movement rate of 1 m/a, motion could not be quantified anymore, but detection was still well possible. This maximum measurable displacement rate does not only depend on the wavelength used, but also on the movement type (e.g., spatially coherent vs. locally variable) and terrain (e.g., terrain slope relative to radar incidence angle, or movement



azimuth relative to satellite azimuth). In the case of Sentinel-1 interferograms, for instance, assuming a maximum measurable displacement rate of half a wavelength (2.8 cm) (Villarroel et al., 2018) gives for the shortest temporal baselines available over the study area (12 days) a maximum value of 85 cm/a.

## 3.2 Matching of repeat optical images

For four sites with rock glaciers in the study area (Gorodetsky/Morenny/Archaly; Ordzhonokidze; Kugalan Tash; Karakorum; Fig. 1), decadal-scale variations of speed were estimated based on offset tracking between repeat optical images. The radiometric quality, distortions and sensor types used in this study are very different. Roughly, we used three categories of data: historic airphotos, historic spy satellite images, and contemporary very-high-resolution satellite data (Table A2). The contemporary satellite images from the DigitalGlobe constellation or Pléiades were oriented using standard procedures based

on the provided rational polynomial function coefficients, and checked with a few control points extracted from GoogleEarth. Over the Gorodetsky/Morenny/Archaly rock glacier group, Pléiades stereo data were available. A corresponding digital elevation model (DEM) was therefore constructed, and the satellite images were orthorectified based on this DEM. For the other three sites without own DEMs produced, orthoimages were computed using the C-band SRTM DEM (30-m resolution), the ALOS PRISM AW3D DEM (commercial version, 5-m resolution version), and the TanDEM-X WorldDEM (commercial

version, 12-m resolution version). The High-Mountain-Asia DEM (Shean, 2017) – produced with stereo data from the DigitalGlobe archive – was also investigated; it turned out to be very good for the Karakoram and Ordzhonikidze rock glaciers, but showed large voids over the Kugalan Tash rock glacier. Based on the analysis of these different DEMs such as from DEM differences, level of detail, and void cover, and based on the distortion patterns over stable ground between the corresponding orthoimages produced, it was decided to use the TanDEM-X WorldDEM for the generation of the orthoimages, except for

Gorodetsky/Morenny/Archaly where the Pléiades DEM was used. At the time of processing, the TanDEM-X WorldDEM appeared to be the most complete, detailed and consistent DEM over all sites. The orthoimages generated from the contemporary optical satellite data served as source of georeference for the older images.

The airphotos used in this study stem from Soviet-era archives, and the digital copies available to us were of very variable quality, typically scans of copies of the original images using consumer-grade scanners, or similar. The images had thus

significant distortions and very variable radiometric contrast. No camera calibrations were available. In addition, we used high-resolution declassified spy satellite images from Corona missions (KH-4A and KH-4B) (Altmaier and Kany, 2002; Bolch et al., 2008; Narama et al., 2010; Goerlich et al., 2017). These images have a complex panoramic geometry, in addition to film distortions. Thus, to both the airphotos and Corona data, no rigorous sensor models were applied but rather empirical models were used consisting of 4[th]-order rational polynomials computed from around 40-50 ground control points at each site. Ground

control was extracted from the above contemporary satellite orthoimages and DEMs. All processing of the optical data was done using the PCI Geomatica software (version 2018sp1).

Eventually, horizontal displacements were measured between pairs of the above orthoimages using standard subpixel-level normalized cross-correlation (Kääb and Vollmer, 2000; Debella-Gilo and Kääb, 2011; Kääb, 2014). For the contemporary



satellite images and some particularly good 20th-century images, entire flow fields could be measured successfully. For the majority of the older image data, the resulting velocity fields had in parts substantial areas were image matching failed, as indicated by low correlation values, or vectors pointing randomly at different directions. Time series were only constructed for points that had reliable measurements for all periods, i.e. measurements that passed thresholds on correlation coefficients and manual removal of obvious outliers. The time series shown in the following results and figures are average time series for all individual measurements within manually delineated zones. These zones are also shown in the figures. The purpose of creating such zones was to focus on central rock glacier parts with substantial movement, and to exclude rock glacier parts that were considered to be less representative, for instance due to location at the margins of the rock glacier (system) or due to uncertain measurements. Even for these averaging zones, the low number of suitable measurements in the historical images typically limited the choice of clusters of points with available time series. The accuracy of temporal speed variations was then evaluated using three measures: (i) speeds on stable ground (for Gorodetsky, Morenny, Archaly only), (ii) variation of speeds within the point clusters (spatial variance), and (iii) variation of speed variations between periods and clusters (spatio-temporal variance). Potential uncertainties in the time series are thus given as the normalized median absolute deviation (NMAD; Höhle and Höhle, 2009) of stable ground speeds (red error bars, (i)), as the standard deviation of speeds over one period (blue error bars, (ii)), and as standard deviation of speed differences between two periods (grey error bars, (iii)). The error budget resulting from the highly variable quality of the base images, their georeferencing, from the DEMs used for their orthorectification, and the offset tracking procedure is examined in more detail in the discussion section of this study.

We also relate the advance rates of rock glaciers to the surface speeds above the fronts. The ratio between both quantities is a function of the ice content, which melts out at the front, and the vertical profile of horizontal speeds, i.e. the total material flux into the front (Kääb and Reichmuth, 2005). We roughly estimate the advance rates by measuring some ten manually selected distances along the main flow direction between repeat positions of the front feet in flickering orthoimages of modern era high-resolution satellite images. It is clear that such estimate of advance is coarse, and difficult anyway due to local variability from e.g. falling rocks and erosions, and our aim is thus only to investigate according large differences between rock glaciers in a qualitative way.

## 4 Results

### 4.1 Rock glacier motion inventory

The inventory of rock glacier creep speed and movement of periglacial and glacial processes over the Ile and Kungöy Ranges of the northern Tien Shan covers a surface of more than 3000 km² and includes more than 900 landforms, namely 551 rock glaciers, 184 solifluction features / debris movements, 46 dead-ice / subsidence forms, 18 debris-covered glaciers, 14 ice-cored moraines. Finally, 98 features remain undefined due to the lack of good quality high-resolution images available at the time the inventory was created. Of the active rock glaciers, which are homogeneously distributed over the study region, 45 are characterised by a rate of motion exceeding 1 m/a. We identified 21 large rock glaciers exceeding 1 km² in size with the largest



being about 2 km². The mean rock glacier size is 0.27 km². Few of the largest rock glaciers have also the largest elevation difference of up to 1,500 m. Several rock glaciers extend below 3000 m asl. with the lowermost elevation of active rock glacier fronts being at about 2700 m asl., amongst them Karakorum and Ordzhonikidse rock glaciers, which are analysed in more detail in this study. The highest rooting zone of a rock glacier is situated above 4000 m and the mean elevation is 3480 m. Six

5 rock glaciers within the class > 1 m/a are further analysed in this study by matching of repeat optical images: Gorodetsky/Morenny/Archaly, Ordzhonikidze, Karakoram, and Kugalan Tash (Fig. 1).

A selected Sentinel-1 interferogram for all these rock glaciers demonstrates how the motion cannot be quantified over large parts of the rock glaciers as no consistent movement fringes are present (Fig. 2). However, the moving areas can still be detected and mapped even if movement fringes cannot be reliably unwrapped (i.e. integrated) into displacement rates or

10 coherence is completely lost.





**Figure 1: Movement inventory of rock glaciers and other forms of surface deformation over parts of the Ile Alatau and Kungöy Ala-Too, northern Tien Shan. Displacement rates are based on satellite radar interferometry, interpretation of process types are based in addition on high-resolution optical satellite images. The glacier mask is from the Randolph Glacier Inventory (RGI) (Pfeffer et al., 2014). The panel to the lower left indicates the location of the study area (red rectangle).**

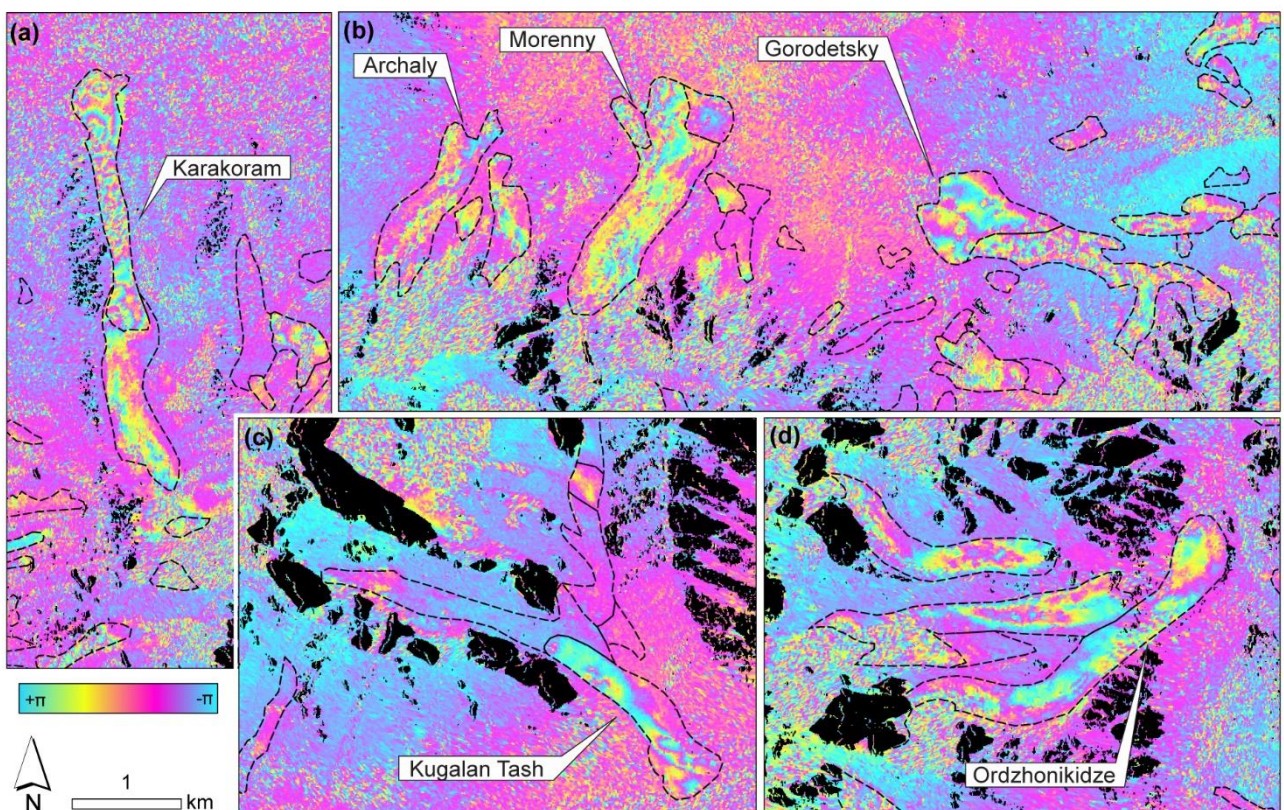

**Figure 2: Raw, orthorectified interferogram between Sentinel-1 radar data of 24 Aug and 5 Sep 2018 (12 days). Black areas are due to radar layover and shadow. The black dashed lines are the inventory outline polygons from our motion and process-type classes. Panels (a)-(d) show interferogram sections of the rock glaciers investigated in detail photogrammetrically.**





## 4.2 Detailed flow fields and sixty-year time series

### 4.2.1 Gorodetsky, Morenny and Archaly

Photogrammetric velocities between 2012 and 2016 (for exact dates and image sources see Table A2) over Gorodetsky rock glacier (Fig. 3a) show a pattern of several zones of high speed (Fig. 3c). Highest speeds of up to 1.4 m/a are found along the

lower southern margin of the rock glacier, which agrees well with the interferogram (Fig. 2) where strongest signs of coherence loss (noisy fringes) are found over the same zone. Gorodetsky rock glacier seems to be nourished primarily by five glaciers, three of which are visible in Figure 3a, whereas two are located more to the east. The zones between these glaciers and the main rock glacier do not show any significant movement anymore. Close inspection of the repeat high-resolution optical images rather reveals a mixture of zones characterized by rapidly changing thermokarst features, such as melt ponds, melt

depressions, or small slope instabilities, and zones that appear stable and without any visible, ice-related activity. Together, these lines of evidence suggest that the rock glacier has lost direct contact with the glaciers and their associated sediment and ice flux as a result of the ongoing glacier retreat.

A time series of speeds for a cluster of points (ca. 45) in the central portion of the rock glacier (white dashed outline in Fig. 3c) suggests a decrease of speed over 1953-1964-1971-2012. By contrast, over the period 2012-2016-2018, we measured a

slight increase in speed, though hardly statistically significant. The large gap of data between 1971 and 2012 prevents a more detailed assessment and only enables measurement of average speeds over a large part of the observation period.

The results for Archaly and Morenny rock glaciers are comparable to those obtained for Gorodetsky. The zones between the nourishing glaciers and the rock glaciers (Fig. 4c) show clear signs of thermokarstic ice degradation and slow/no movement as well, but the evidence is spatially less extensive than at Gorodetsky. A striking fast-creeping branch (up to 2.5 m/a) diverts

from the mainly south-north flowing Morenny rock glacier towards the northwest (Fig. 4e). The InSAR results (Fig. 2) are consistent with the velocity field (Fig. 4e) in that they show coherent motion fringes over much of the slower-moving part of the rock glacier, but strong noise from phase coherence loss over the fast-moving lateral branch. Overall, Archaly rock glacier moves faster than Morenny (Fig. 4d), which is also reflected in the more decorrelated interferometric phases (Fig. 2). Though, also on the central part of Morenny rock glacier speeds are in the order of 1 m/a, i.e. consistent with the interferogram

interpretations (Fig. 1). The time series for Archaly (~40 points) and Morenny (~75 points) rock glaciers (Fig. 4a and b) look similar to Gorodetsky, with the clear exception of increasing speeds over the period 1953-1964-1971.

The long time series of airphotos and satellite images allow an estimation of advance rates of the rock glaciers, and relate these to surface speeds above the fronts. Advance rates 2012-2018 at the southern branch of Gorodetsky rock glacier are roughly 0.8 m/a, and the 2012-2016 surface speeds around 1.2 m/a, giving a ratio of around 0.67. In addition we measure an average

advance 1953-2012 of 0.7 m/a. The fast-creeping western branch of Morenny rock glacier shows very little advance, but surface speeds above the front of 2 m/a, giving a very small advance ratio between both, equivalent to a very shallow sheet of creeping material (or very ice rich material). Archaly rock glacier showed advance rates of roughly up to 0.3 m/a, with 2012-2016 surface speeds of around 1.1 m/a, and thus an advance ratio of around 0.3.





**Figure 3: Velocities and speed variations on Gorodetsky rock glacier. (a) Overview image; Planet image of 7 Sept 2016 in near-infrared – red – green R-G-B composition. (b) Average time series of speeds for a cluster of points (white dashed outline in panel c). Error bars give normalized median absolute deviations (NMAD) of stable ground speeds (red), standard deviations of speeds within the point cluster (blue), and standard deviations of speed differences within the measurement periods (grey) in m/a. Blue triangles on the time axis indicate years of imagery used. (c) Colour-coded velocity field with vectors superimposed from image matching between 2012 and 2016 very high-resolution satellite images (for exact dates and sensors see Table A2). Background image: Pléiades, 27 Aug 2016 © CNES (2016), and Airbus DS (2016), all rights reserved.**











**Figure 4: Velocities and speed variations on Archaly and Morenny rock glaciers. (a)(b) Average time series of speeds for clusters of points (white dashed outline in panels d and e). Error bars give normalized median absolute deviations (NMAD) of stable ground speeds (red), standard deviations of speeds within the point cluster (blue), and standard deviations of speed differences within the measurement periods (grey) in m/a. Blue triangles on the time axis indicate**

**years of imagery used. (c) Overview image; Planet image of 7 Sept 2016 in near-infrared false colour. (d)(e) Colour-coded velocity fields with vectors superimposed from image matching between 2012 and 2016 very high-resolution satellite images (for exact dates and sensors see Table A2). Background image: Pléiades, 27 Aug 2016, © CNES (2016), and Airbus DS (2016), all rights reserved.**

**4.2.2 Karakoram**

The velocity field on Karakoram rock glacier (Fig. 5a,c) shows one coherent stream of material nourished from a partially ice-covered rock wall (Fig. 5a). A few thermokarst depressions and lakes are found in the root zone of the rock glacier, but a spatially coherent material flux can be discerned from the nourishing ice and debris reservoir into the main rock glacier. The lowermost part of the rock glacier hosts significant vegetation including dozens of trees (Sorg et al., 2015), visible in red colour

in the infrared false colour composite Fig. 5a. Maximum speeds recorded over 2009-2017 reach up to 6.5 m/a. These high speeds are consistent with the interferogram given in Fig. 2, where strong phase noise is visible. The high speeds and associated surface changes result in a situation in which points with valid velocity measurements over all measurement periods can only be found in the lowermost portion of the rock glacier. There, the average of the point cluster (ca. 20 points) shows a steadily increasing speed over the period 1960-1971-1980-1988-2001-2009-2017 (Fig. 5b). Available airphotos from 1977, 1985 and

2008 have not been used in this time series because they were of too low quality to enable reliable measurements, or their dates were too close to other image dates so that measurements did not exceed the noise level. The advance rate of Karakoram rock glacier can only be measured in a meaningful way in the north-western portion of the rock glacier front as the other parts of the front terminate in a river. In this north-western part of the front, the advance rate (~1 m/a) is very similar to the surface speeds above the front, leading to a ratio close to 1. This points to block movement of the entire rock glacier column and very

little ice content left at the front.



**Figure 5: Velocities and speed variations on Karakoram rock glacier. (a) Overview image; Planet image of 7 Sept 2016 in near-infrared false colour. (b) Average time series of speeds for a cluster of points (white dashed outline in panel c). Error bars give standard deviations of speeds within the point cluster (blue), and standard deviations of speed differences within the measurement periods (grey) in m/a. Blue triangles on the time axis indicate years of imagery used. (c) Colour-coded velocity field with vectors superimposed from image matching between 2009 and 2017 very high-resolution satellite images (for exact dates and sensors see Table A2). Background image: GeoEye, 29 Jul 2017.**



### 4.2.3 Ordzhonikidze

Ordzhonikidze rock glacier is a system of several streams, all nourished from cirques with small glaciers (Fig. 6a). The lowermost part of the longest stream reaches into forested terrain and carries vegetation, including dozens of trees (shown in red in Fig. 6a) (Sorg et al., 2015). For all individual rock glacier streams a coherent flux of material seems to exist from the cirque glaciers into the rock glacier(s). Highest speeds of up to 4 m/a over 2016-2018 are found on a steep narrow part of the longest stream. The interferometric fringe pattern (Fig. 2) and the interferometric interpretations (Fig. 1) are consistent with photogrammetric measurements, with phase noise over the fastest rock glacier parts, and more coherent fringes over the slower parts. Velocity time series were computed for four zones (A: ~ 60 points; B: ~100; C: ~70; D: ~50; Fig. 6e) and over the periods 1953-1966-1971-1985-2009-2013-2016-2018 (for exact image dates and types see Tab. A2). A period of higher speeds over 1966-1971 is found for all four zones but increased velocities only barely exceed the error margins (Fig. 6b-d). Although with variable timing and to variable degrees, we detect an overall increase in speed for all rock glacier zones investigated. Rock glacier advance rate ratios are difficult to characterize because of the many different frontal parts. Roughly, at the front below zone A, advance rates are in the order of 0.7 m/a and speeds above this front are between 1 and 2 m/a (corresponding to advance ratios of around 0.35 – 0.7). At the front below zone D, the advance is in the order of 0.7 m/a, and surface speeds are around 1.2 m/a (with a ratio of around 0.6), based on advance rates and speeds over the period 2009-2018.









**Figure 6: Velocities and speed variations across the Ordzhonikidze rock glacier system. (a) Overview image; GeoEye image of 6 Oct 2013 in near-infrared false colour, with the southernmost part from a Planet image of 7 Sep 2016. (b-d) Average time series of speeds for clusters of points (zones A-D; white dashed outline in panel e). Error bars give standard deviations of speeds within the point clusters (blue), and standard deviations of speed differences within the measurement periods (grey) in m/a. Blue triangles on the time axis indicate years of imagery used. (e) Colour-coded velocity field with vectors superimposed from image matching between 2016 and 2018 very high-resolution satellite images (for exact dates and sensors see Table A2). Background image: GeoEye, 6 Oct 2013.**

### 4.2.4 Kugalan Tash

Kugalan Tash rock glacier is nourished from a cirque with no significant ice bodies remaining at the present day (Fig. 7a). Our earliest airphotos of 1956 show clearly more ice cover than today, both in the rock walls and at their base. Movement rates between this ice and the sediment reservoir into the main rock glacier are smaller than the significance level so that the supply of sediment and glacial ice to the rock glacier system as a transition from debris-covered ice into frozen debris cannot be ascertained. Only very little vegetation is growing on the rock glacier (Fig. 7a) (Sorg et al., 2015). Maximum speeds found over 2012-2014 are around 2.5 m/a. The photogrammetrically measured rock glacier stream is well reflected in the interferogram depicted in Fig. 2, with coherent fringes in the slower parts, and phase noise due to coherence loss over the faster parts. Speed time series for 1956-1973-1988-2004-2012-2014 were measured for two zones (A: ~20 points; B: ~30). Zone A shows a steady increase in speeds (Fig. 7c). The behaviour of velocities in zone B is very similar to that observed in zone A but more damped with a smaller range of change. Noteworthy, changes documented for zone B (not shown) are within their error bars and cannot thus be considered statistically significant. At this site, we did not use Corona satellite data of 1971 as the airphotos of 1973 were of better quality. Rock glacier advance rates 2004-2012 are around 0.4 m/a, for speeds recorded above the front of around 1.2 m/a, corresponding to an advance ratio of around 0.3.



**Figure 7: Velocities and speed variations on Kugalan Tash rock glacier. (a) Overview image; Planet image of 7 Sept 2016 in near-infrared false colour. (b) Colour-coded velocity field with vectors superimposed from image matching between 2012 and 2014 very high-resolution satellite images (for exact dates and sensors see Table A2). Background image: WorldView1, 23 Sep 2012. (c) Average time series of speeds for cluster of points A (white dashed outline in panel b). Error bars give standard deviations of speeds within the point clusters (blue), and standard deviations of speed differences within the measurement periods (grey) in m/a. Blue triangles on the time axis indicate years of imagery used.**



## 5 Discussion

### 5.1 Radar interferometry versus photogrammetry

The photogrammetric velocity fields derived from repeat airphotos and very high-resolution optical satellite data in this study agree reasonably very well with the radar interferograms and their interpretation. Noteworthy, both datasets have been obtained

independently as the photogrammetric velocity fields were not accessible to the team processing and interpreting radar interferometry data, and vice versa. Nonetheless, all rock glaciers investigated using photogrammetry were classified correctly from the interferograms (Fig. 1). When having a closer look at the full resolution of the radar interferograms, however, small differences can be seen. As radar interferometry provides in principle only one motion component (line-of-sight) – by contrast to photogrammetry where two horizontal components are obtained – any discrimination of lateral from vertical movements is

obviously more difficult in radar interferograms. By way of example, a small elongated zone of phase coherence loss (phase noise) can be detected in the eastern sector of the main part of Gorodetsky rock glacier (Fig. 2), and has been interpreted as a fast-moving zone within the rock glacier (Fig. 1). Photogrammetric investigation, however, shows that this zone should be interpreted rather as a zone of thermokarstic erosion, a process that can obviously cause loss of interferometric phase coherence. One needs to underline, however, that radar interferometry and multitemporal photogrammetry require very

different datasets and that the two approaches usually serve different purposes. Radar interferometry is typically used to reliably map the activity of a large number of rock glaciers over large areas, a target that can only be achieved by using free, weather-independent, and easily accessible baseline data (e.g. Sentinel-1). By contrast, photogrammetric approaches as the ones used here, require very specific, highly-resolved data that cover only small areas, and can be expensive or else difficult to obtain.

In addition, we would also like to stress that the motion patterns seen in the raw radar interferograms (Fig. 2) agree very well with the photogrammetric velocity fields. Fast moving rock glacier parts are characterized by interferometric phase noise, whereas slower-moving parts show more coherent interferometric fringes (e.g., Fig. 8). In the 12-day interferograms displayed in Figs. 2 and 8, the transition between both cases seems to be indeed at speeds of around 1 m/a, which is in good agreement with estimates from Villarroel et al. (2018).

Whereas our motion classes are based on quantitative measurements, it is clear that our landform classification involves some uncertainty especially for process types that naturally have no sharp boundary between them, in spatial terms or in terms of process definition, but rather transitions, such as the downstream transition from a debris-covered glacier into a (glacier-derived) rock glacier. Further it is clear, that the radar and optical data available at the time of classification, for instance their resolution, will influence the class decision.





**Figure 8: Ordzhonikidze rock glacier. Section of a interferogram between Sentinel-1 radar data of 24 Aug and 5 Sep 2018 (12 days; slightly smaller section than Fig. 2d) with vectors superimposed from image matching between 2016 and 2018 very-high resolution satellite images (for exact dates and sensors see Table A2). The white dashed lines are the inventory outline polygons from our motion and process-type classes.**

## 5.2 Uncertainties in the photogrammetric measurements

In this section, the error budget of our photogrammetric displacement measurements, and thus of our speed time series, is analysed. Error budgets are typically composed of (a) overall shifts between the orthorectified data, (b) lateral shifts in the orthoimages due to errors in the DEM used for orthorectification, (c) distortions in the raw data or in the sensor model that propagate into the orthoimages, and (d) image matching uncertainties and errors (Kääb et al., 2016; Kääb et al., 2017, 2019).



Biases of type (a) are much minimized by our processing scheme and should not influence our results in a significant way. One orthorectified very-highresolution satellite scene is used as master for georeferencing of all other images over a site, and image sections are additionally co-registered over stable ground during the offset tracking procedure.

Orthorectification offsets due to DEM errors, i.e. error budget component (b), are more difficult to quantify precisely. They

will in particular affect the airphotos we used as those have typically much larger opening angles (field of view versus flying height) compared to the satellite images. In the very worst case, a DEM error could translate 1:1 to a lateral offset. We can estimate such orthorectification offsets from stable ground offsets (Figs. 3b and 4a-b), but we cannot be sure that these estimates are representative for the rock glacier surfaces as the elevations there could have changed over time with respect to the DEM used for orthorectification (here the TanDEM-X WorldDEM), or DEM errors on rock glaciers could be

systematically different from DEM errors outside the rock glaciers. In this study, we computed a number of orthoimages using different DEMs (see Section 3.2) and analysed offsets in them in order to select the best DEM. Another, more stringent way to estimate orthorectification errors would be to analyse offsets between stereo orthoimages, i.e. different images of the same scene (typically stereo images) rectified by the same DEM (Kääb and Vollmer, 2000; Kääb et al., 2016; Altena and Kääb, 2017). We had no suitable image data available to perform such tests, though.

Distortions in the raw data or sensor model used, i.e. errors of type (c), are especially expected for the Corona spy satellite data and some of the airphotos obtained. We have attempted to minimize such effects by using 4th-order rational polynomial sensor models computed from around 40-50 ground control points per image. These errors can be estimated from stable ground offsets as well. By contrast to the orthorectification offsets, it can be assumed that image or model distortions are independent of the image content, and affect thus stable areas in a similar way than the rock glaciers.

The final error category (d) concerns matching errors. These can range from gross outliers to fine matching inaccuracies. Gross outliers in our displacement measurements are eliminated by thresholds on the correlation coefficient and a manual selection of points that have consistent measurements in all measurements periods for a site. Thereby, those vectors appearing inconsistent with respect to surrounding vectors (i.e. mainly sections of the vector field with random variation of directions) have been disregarded. In addition, we made sure that the time series obtained from mean speeds for point clusters were

consistent with the time series obtained from median values for the point clusters, as the latter are less sensitive to outliers. The remaining matching accuracy varies certainly much with the corresponding image quality, including resolution, contrast, and/or noise. Effects from matching inaccuracies are minimized by averaging speeds of point clusters over several tens of points. Again, this matching accuracy can be estimated from stable ground offsets, in particular for surfaces that have similar visual contrast as the rock glaciers.

In stable ground offsets, the combination of error types (b)-(d) is present. For Gorodetsky, Morenny and Archaly rock glaciers, we compared the normalized median absolute deviations (NMAD) of stable ground offsets to the standard deviations of speeds within point clusters, and to the standard deviation of speed differences to find that the latter two seem to be on overall a more conservative uncertainty measure, which we thus show for the other time series. The latter two standard deviations contain also natural and, thus, non-erroneous variability in speeds and speed differences. On the other hand, we know that





orthorectification errors (b) might be larger on the rock glaciers than around them, so that we prefer to use an error measure that is more conservative than the stable ground offsets.

## 5.3 Speed time series

The photogrammetric velocity fields from this study show a wide spatial variation of rock glacier speeds, with the maximum

speeds of up to 6.5 m/a found on Karakoram rock glacier. Whereas the Karakoram, Ordzhonikidze, and Kugalan Tash rock glaciers, or rock glacier systems, consist of well identifiable individual streams, Gorodetsky, Morenny and Archaly rock glaciers display more complex flow patterns, with notable divergence in the downstream direction.

This study also shows that the temporal variations of rock glacier speeds from repeat airphotos and satellite data, available since the 1950s, displays a contrasting picture. The only feature in the time series common to all rock glaciers studied in more

detail is indeed an increase in speeds in recent years, but results clearly differ in their levels of statistical significance.

Karakoram rock glacier shows a consistent increase in speed over the entire study period, but due to its fast movement, which much limits reliable measurements over all measurement periods and all sections, this can only be confirmed for the lowermost part of the rock glacier. Through the manual tracking of 15 individual rocks in airphotos over the period 1969-1977-1984, Gorbunov et al. (1992) find a similar behaviour of speed increase in the lowermost part of the rock glacier (their measurement

points 1 and 3) (Gorbunov and Titkov, 1989). For the parts further up, the authors documented a decrease in speed over their two study periods, interpreting these changes as a wave-like acceleration/deceleration process as the rock glacier material is travelling downstream. For a few points in the middle of the rock glacier (around points 6-9 of Gorbunov and Titkov, 1989), where consistent image matching was possible from our data, we find a deceleration of movements over the period 1964-1971, but a continuous acceleration thereafter (over the full period 1971-1980-1988-2001-2008-2017). Speeds documented for 1980-

1988 were still below the values recorded for 1964-1971, but during 1980-1988 already almost back to the 1964-1971 level. The few measurement points for this estimate are not sufficient to compute reliable error statistics, though. Given the different measurement locations in Gorbunov et al. (1992) and the large displacement uncertainties in that study, it is still consistent with the present study regarding an overall rock glacier acceleration 1964-2017 for the lowermost part. In addition, our 1964-1971-1980 speed variations (Fig 5a) are close to their statistical significance, and we cannot clearly exclude a slight speed

decrease over these periods, as suggested by Gorbunov et al. (1992) for the middle rock glacier section.

It therefore seems reasonable to assume that the lowermost rock glacier part, which deforms under a regime of strong longitudinal compression, as seen from the photogrammetric velocity fields and consistent with the many transverse ridges (Kääb and Weber, 2004), responds rather passively to flux variations in the upper rock glacier parts and thus displays these variations in a dampened manner. In addition, and as indicated by the advance ratio, the ice content of the front could be quite

low. Both for the lowermost rock glacier part (Fig. 5a) and the few points documented in the middle part, speeds have more than doubled over 1964-2017. The results from the present study are also consistent with those from Sorg et al. (2015). Noteworthy, both studies were based on the same data, but the present work uses improved processing approaches and extends




the time series. As such, the measurement points used in Sorg et al. (2015) extend less further down compared to the present study, so that the speeds found in their work are, on average, higher compared to the present study.

Kugalan Tash rock glacier shows an overall increase of speeds over 1956-2014 as well, but at a lower significance level than Karakoram rock glacier (Fig. 7c). The increase in speeds on the lowermost rock glacier part (zone B) is rather dampened. The latter part is under a strong compressive creep regime and reaches and diverts the river, similar to the situation observed for the lowermost part of Karakoram rock glacier. The photogrammetric measurements in Sorg et al. (2015) show, by contrast to the reprocessing and extension performed here, higher speeds during 1956-1971, a feature that we cannot confirm in the new assessment anymore. The use of an improved DEM in our new processing and other improvements related to co-registration of images are able to explain this difference. No independent measurements exist, however, from other studies that could be used for comparison with our results.

Overall, all investigated zones on the Ordzhonikidze rock glacier system show an acceleration over the period 1953-2018 as well, a phenomenon that is most clearly visible in the higher-lying northern rock glacier (zone A in Fig. 6e). A striking peak in speeds is visible over the period 1966-1971 (Fig. 6b-d), for which we do not, however, have a clear explanation. Meteorological and glacier mass balance data (Figure 2 in Sorg et al. (2015), and (Bolch and Marchenko, 2009)) show that this period covers mostly years with rather cold summer temperatures in the region, and balanced glacier mass budgets, i.e. not particularly warm air temperatures or large melt water productions. Importantly however, and consistent with the glacier mass balances, snow measurements at two stations close to the rock glacier (Bolshoje Almatinskoje Ozero, Great Almatinka Lake , 2516 m asl., and Mynzhilki, 3017 m asl.) (for locations see Bolch and Marchenko, 2009) show strongly above-average snow heights during the 1960s (MeteorologicalService, 1968). Using a dendrochronological rock glacier activity index, Sorg et al. (2015) show that the period 1960-1970 is characterized by particularly high movement activity at Ordzhonikidze rock glacier. Such activity peak is not found for Karakoram rock glacier and another rock glacier in the region studied in the latter work (Figure S8 in Sorg et al. (2015)). Increased permafrost and ground ice temperatures due to strong surface insulation, and perhaps enhanced amounts of snow melt water, from exceptionally large snow heights are thus a reasonable explanation for the peak in rock glacier speed during the 1960s. We find no evidence that an earthquake could have caused enhanced displacements (Rosenwinkel et al., 2015). We did not find any indications of particular distortions in the images that could have caused erroneous displacement values. No measurements exist from other studies that could be used to compare our results.

Interestingly, Gorodetsky, Morenny, and Archaly rock glaciers are the only systems investigated in this study that are not showing a clear increase of speed over the study period. At the same time, however, these are also the sites for which data availability was least good and for which no images existed between1971 and 2012. The increase in speeds over 2012-2016-2018 is at the limit of statistical significance (Figs 3b and 4a-b). Archaly and Morenny rock glaciers show a similar behaviour over time, and a peak in speed over 1964-1971, similar to Ordzhonikidze rock glacier, which is just 15 km to the northeast, at the northern slope of the Ile Alatau, and likely under similar meteorological conditions. The agreement between Archaly, Morenny and Ordzhonikidze rock glaciers with respect to the peak in speeds in the second half of the 1960s, and the high



dendrochronological activity during this time on Ordzhonikidze rock glacier (Sorg et al., 2015) suggest that this feature in speed variations could actually be a reliable measurement, possibly related to large snow heights and associated increased ground temperatures.

Gorbunov et al. (1992) investigated displacements of blocks on Archaly and Morenny rock glaciers over 1969-1984 and find an overall increase in speeds between 1969-1979 and 1979-1984. Over shorter time scales (1973-1977-1978-1982), the authors first identify a decrease in speeds and then again an increase, leading them to suggest the existence of pulse-like speed variations. Our measurements contain no details over these study periods but it seems very reasonable to assume that speeds did actually vary within our 1971-2012 measurement period.

By contrast to Archaly and Morenny rock glaciers, Gorodetsky rock glacier showed even higher speeds during 1953-1964 as compared to 1964-1971. A reason for the overall decrease in speed for this rock glacier over the study period and the weakest speed increase since 2012 (Fig. 3b) could be that this rock glacier seems decoupled from its nourishing ice and debris reservoir. Measurements for some blocks by Gorbunov et al. (1992) for 1964-1984 reveal speeds similar to our measurements, specifically slightly higher than our 2012-2016 speeds.

In-situ measurements of the lower outline of the Gorodetzky rock glacier front revealed that advance rates for 1923-1980 around the location where also we measured the advance are in the order of 0.8 m/a (Gorbunov and Titkov, 1989; Kokarev et al., 1997), i.e. equal to our coarse measurements of 0.8 m/a for 2012-2018, and slightly higher than our 1953-2012 advance of 0.7 m/a. Advance rates increased to about 1.6 m/a in the 1990s (Bolch and Marchenko, 2009). As our 2012-2018 estimates are significantly lower than the latter value for the 1990s, they might be an indication for a decrease in advance rate. But our 1953-2012 estimate of 0.7 m/a is not compatible with this assumption unless there were particularly low advance rates between the 1990s and 2012. Given the coarse character of our image-based advance estimates and the high variability of rock glacier advance over short time scales, it is difficult to compare our estimates in detail to the previous measurements, though.

## 5.4 Wider geomorphic implications

Given the increase in air temperatures and the negative glacier mass balances in the region during the study period (Bolch, 2007; Sorg et al., 2012; Sorg et al., 2015), the behaviour observed in the study area confirms findings from Europe and from theoretical considerations that rock glacier creep speeds increase overall under atmospheric warming (Kääb et al., 2007; Delaloye et al., 2010; PERMOS, 2016; Eriksen et al., 2018). Surface speeds for most rock glaciers photogrammetrically investigated in our study were roughly two to three times higher in recent years than during the 1950s and 1960s. This indicates a significant increase in sediment and ice fluxes through these rock glaciers, and potentially for the entire study region if one assumes that speed increases apply in overall also to the other rock glaciers mapped using radar interferometry. Together with the shrinkage of glaciers in the region (Narama et al., 2010; Farinotti et al., 2015; Severskiy et al., 2016; Treichler et al., 2019), this implies that periglacial sediment transport seems to gain currently importance relative to glacial sediment transport (c.f., Monnier and Kinnard, 2017; Seligman et al., 2019).





The ratios between rock glacier advance rates and surface speeds over the rock glaciers studied in detail cover almost the full range between very small ratios and 1, with most ratios between 0.3 and 0.7. The ratio between both quantities is a function of the ice content, which melts out at the front, and the vertical profile of horizontal speeds, i.e. the total material flux into the front (Kääb and Reichmuth, 2005).

All rock glaciers studied in detail in this study are derived from contemporary or former glaciers. The size of these glaciers, which will have been (substantially) different between the study rock glaciers, could have (had) an important influence on the volume of the rock glacier ice cores. Differences in extent and thickness of these ice cores between the rock glaciers, which we do not know in sufficient detail, could well play a role in the temporal variability of surface speeds and in the response of speeds to atmospheric warming.

## 10 6 Conclusions

Using satellite radar interferometry, we have compiled an activity inventory of rock glaciers and related landforms for the Ile Alatau and Kungöy Ala-Too ranges of northern Tien Shan. We confirm that these ranges host a large number of rock glaciers and identify a large number of comparably fast-moving (>1 m/a), individual rock glaciers. The radar-interferometric interpretations derived in this study agree well with photogrammetric offset tracking measurements based on repeat, very high-

resolution optical satellite data, both in terms of overall motion class and surface velocity patterns. When using 12-day Sentinel-1 interferograms, speeds of up to roughly 1 m/a can be measured, above which phase decorrelation makes precise quantification of speeds impossible. However, the outlines and lower limit of the velocity of the surface can be identified using the assumptions stated above.

Overall, we find an increase in surface speed for selected rock glaciers from the 1950s on. This confirms theoretical

considerations of overall rock glacier acceleration under atmospheric warming, and observational evidence that was so far mainly available for the European Alps. Our finding suggests that for many rock glaciers of our movement inventory speeds might have been lower during recent decades than contained in this contemporary map. The two to three-fold increase in surface speeds for most time series constructed between the 1950s-1960s and recent years implies a substantial increase in sediment and ice fluxes through these rock glaciers. Relative to the shrinking glaciers in the region, this means a higher

periglacial sediment transport compared to a shrinking glacial sediment transport.

Together with the overall increase in speeds over time, we find also decadal speed variations. The most striking peak in speeds is estimated for the second half of the 1960's for Morenny, Archaly and Ordzhonikidze rock glaciers on the northern slope of the study region in the Ile Alatau. A possible explanation for this feature are particularly large snow heights in the region during the period of concern that could have contributed to enhanced surface insulation during winter, and associated reduced

permafrost cooling and ground ice warming, and enhanced advection of snow melt water into the rock glaciers.

The rock glacier fronts that reach valley floors show a strongly compressive flow regime, and changes or increases, respectively, in speeds further upstream affect them in a damped way. The fronts might respond rather more passive to upstream speed changes than actively produce own changes.

The only rock glacier speed time series that does not exhibit overall increase in speed is found for Gorodetsky rock glacier, a rock glacier where glacier retreat and dead-ice degradation seem to have decoupled the rock glacier from supply by glacial sediments and ice. Despite this reduction in nourishment, Gorodetsky rock glacier still shows substantial creep rates suggesting the persistence of ice-rich permafrost and its deformation under gravity. Decreases in speed due to decreasing supply, as perhaps indicated for the beginning of the study period (Fig. 3b), could have been compensated in parts by increases in speeds due to increasing deformation of the ice core following increased air temperatures.

All rock glaciers for which velocity time series have been measured in this study are derived from actual or former glaciers. This connection has certainly influence on the rock glacier speeds and their spatio-temporal changes, as suggested above for Gorodetsky rock glacier. Whereas the amount of ground ice in the rock glaciers and the related ice supply from glaciers above them cannot be easily quantified, the combination of elevation change time series over the rock glaciers and their contributing glaciers with the kinematics of this study might in a next study better illuminate the mass balance of these connected land

forms and the influence of this connection on their response to climatic changes.

**Code availability**

The image matching code used for this study (Correlation Image AnalysiS, CIAS) is available from
http://www.mn.uio.no/icemass (Kääb, 2014). The software used for radar interferometry, the GAMMA remote sensing software, is commercial.

**Data availability**

Sentinel-1 data are freely available from the ESA/EC Copernicus Sentinels Scientific Data Hub at https://scihub.copernicus.eu (Copernicus, 2019). The rock glacier and slope instability inventory compiled for this study from differential SAR
interferograms and optical imagery, 1998-2016, Northern Tien Shan, is available from the Permafrost Information System (PerSys, https://apgc.awi.de/group/persys) (PerSys, 2019). ERS-1/2 and ALOS-1 PALSAR-1 data are available from ESA through registration procedures. ALOS-2 PALSAR-2 data are available from JAXA, and TanDEM-X data from DLR through proposal application procedures. Planet data are not openly available as Planet is a commercial company. However, scientific access schemes to these data exist (https://www.planet.com/markets/education-and-research/). Data from DigitalGlobe



satellites (GeoEye, Ikonos, WorldView, Quickbird) and Pléiades are commercial, but programs to facilitate academic access exist.



## Appendix

**Table A1: Main characteristics of the differential SAR interferograms used. Date format: YYYYMMDD**

| Satellite | Orbit | Date 1 | Date 2 | Perp. Baseline | Time Interval |
|---|---|---|---|---|---|
| ERS-1/2 | Descending | 19980707 | 19980708 | -455.5 m | 1 day |
| ERS-1/2 | Descending | 19990413 | 19990414 | -118.1 m | 1 day |
| ERS-1/2 | Ascending | 19990331 | 19990401 | 59.1 m | 1 day |
| ERS-1/2 | Ascending | 19990505 | 19990506 | 69.6 m | 1 day |
| ERS-1/2 | Ascending | 19990609 | 19990610 | 34.8 m | 1 day |
| ALOS-1 PALSAR-1 | Ascending | 20070903 | 20071019 | 305.6 m | 46 days |
| ALOS-1 PALSAR-1 | Ascending | 20080119 | 20080305 | 650.5 m | 46 days |
| ALOS-1 PALSAR-1 | Ascending | 20080305 | 20080420 | 460.9 m | 46 days |
| ALOS-1 PALSAR-1 | Ascending | 20080905 | 20081021 | 650.4 m | 46 days |
| ALOS-1 PALSAR-1 | Ascending | 20090121 | 20090308 | 460.8 m | 46 days |
| ALOS-1 PALSAR-1 | Ascending | 20100124 | 20100311 | 627.3 m | 46 days |
| ALOS-1 PALSAR-1 | Ascending | 20070116 | 20090121 | 225.9 m | 736 days |
| ALOS-1 PALSAR-1 | Ascending | 20070903 | 20100727 | 946.6 m | 1058 days |
| ALOS-1 PALSAR-1 | Ascending | 20071019 | 20100727 | 641.0 m | 1012 days |
| ALOS-1 PALSAR-1 | Ascending | 20070817 | 20071002 | 251.6 m | 46 days |
| ALOS-1 PALSAR-1 | Ascending | 20080102 | 20080217 | 994.8 m | 46 days |
| ALOS-1 PALSAR-1 | Ascending | 20080819 | 20081004 | 957.3 m | 46 days |
| ALOS-1 PALSAR-1 | Ascending | 20090104 | 20090219 | 497.8 m | 46 days |
| ALOS-1 PALSAR-1 | Ascending | 20100107 | 20100222 | 710.6 m | 46 days |
| ALOS-1 PALSAR-1 | Ascending | 20100525 | 20100710 | 65.9 m | 46 days |
| ALOS-1 PALSAR-1 | Ascending | 20100710 | 20100825 | 451.9 m | 46 days |
| ALOS-1 PALSAR-1 | Ascending | 20100825 | 20101010 | 389.7 m | 46 days |
| ALOS-1 PALSAR-1 | Ascending | 20100710 | 20101010 | 841.6 m | 92 days |
| ALOS-1 PALSAR-1 | Ascending | 20061230 | 20090104 | 6.9 m | 736 days |
| ALOS-1 PALSAR-1 | Ascending | 20070817 | 20090822 | -1038.9 m | 736 days |
| ALOS-1 PALSAR-1 | Ascending | 20071002 | 20100825 | 1030.7 m | 1058 days |
| ALOS-2 PALSAR-2 | Descending | 20150306 | 20150320 | -220.8 m | 14 days |
| ALOS-2 PALSAR-2 | Ascending | 20141005 | 20141214 | 180.2 m | 70 days |
| ALOS-2 PALSAR-2 | Ascending | 20141005 | 20151004 | 142.6 m | 364 days |
| ALOS-2 PALSAR-2 | Ascending | 20141005 | 20160724 | 57.4 m | 658 days |
| ALOS-2 PALSAR-2 | Ascending | 20141214 | 20150222 | 64.2 m | 70 days |
| ALOS-2 PALSAR-2 | Ascending | 20151004 | 20160724 | -85.2 m | 294 days |
| ALOS-2 PALSAR-2 | Ascending | 20160724 | 20161030 | -24.4 m | 98 days |
| Sentinel-1 | Descending | 20150712 | 20150724 | -170.5 m | 12 days |
| Sentinel-1 | Descending | 20150712 | 20150829 | 27.0 m | 36 days |
| Sentinel-1 | Descending | 20160718 | 20160811 | -65.9 m | 24 days |
| Sentinel-1 | Descending | 20160811 | 20160904 | 102.6 m | 24 days |
| Sentinel-1 | Descending | 20160904 | 20160928 | -50.4 m | 24 days |
| Sentinel-1 | Descending | 20160928 | 20161022 | -74.0 m | 24 days |
| Sentinel-1 | Descending | 20150712 | 20160718 | 15.9 m | 372 days |
| Sentinel-1 | Descending | 20150829 | 20160811 | -77.0 m | 348 days |
| Sentinel-1 | Ascending | 20150723 | 20150816 | 26.5 m | 24 days |
| Sentinel-1 | Ascending | 20150816 | 20150909 | -36.6 m | 24 days |
| Sentinel-1 | Ascending | 20160729 | 20160822 | 79.2 m | 24 days |
| Sentinel-1 | Ascending | 20160822 | 20160915 | -14.7 m | 24 days |
| Sentinel-1 | Ascending | 20150723 | 20150909 | -10.1 m | 48 days |
| Sentinel-1 | Ascending | 20160729 | 20160915 | 64.5 m | 48 days |
| Sentinel-1 | Ascending | 20150723 | 20160729 | -53.4 m | 372 days |
| Sentinel-1 | Ascending | 20150816 | 20160822 | -0.8 m | 372 days |
| Sentinel-1 | Ascending | 20150909 | 20160915 | 21.1 m | 372 days |
| Sentinel-1 | Descending | 20180824 | 20180905 | 51 m | 12 days |
| Sentinel-1 | Descending | 20180824 | 20180917 | 13 m | 24 days |
| Sentinel-1 | Ascending | 20180825 | 20180906 | -72 m | 12 days |
| Sentinel-1 | Ascending | 20180825 | 20180918 | -79 m | 24 days |
| Sentinel-1 | Ascending | 20180906 | 20180918 | -6 m | 12 days |



**Table A2: Airphotos and optical satellite data used. The column "Gorodetsky …" includes Morenny and Archaly rock glaciers. Date format: YYYY MMDD. Note, the days and months of acquisition on the airphotos were not always clearly identified and could be wrong in a few cases by several days to a month.**

| Gorodetsky … | Karakoram | Ordzhonikidze | Kugalan Tash | Image source |
|---|---|---|---|---|
| 1953 | | 1953 | | Airphoto |
| | | | 1956 0902 | Airphoto |
| | 1964 0819 | | | Airphoto |
| 1964 1006 | | | | Corona |
| | | 1966 0913 | | Airphoto |
| | 1971 0917 | 1971 0917 | 1971 0917 | Corona |
| | | | 1973 0917 | Airphoto |
| | 1977 0818 | | | Airphoto |
| | 1980 0712 | | 1980 0712 | Airphoto |
| | 1985 0825 | 1985 0726 | | Airphoto |
| | 1988 0810 | | 1988 0810 | Airphoto |
| 1990 0915 | | | | Airphoto |
| | 2001 1114 | | | Ikonos |
| | | | 2004 0712 | Airphoto |
| | 2008 0813 | | | WorldView |
| | 2009 0801 | | | Quickbird |
| | | 2009 0806 | | WorldView |
| 2012 0809 | | | | GeoEye |
| | | | 2012 0923 | WorldView |
| | | 2013 1006 | | GeoEye |
| | | | 2014 0710 | WorldView |
| 2016 0827 | | | | Pléiades |
| | 2017 0729 | | | GeoEye |
| 2018 0823 | | 2018 0727 | | WorldView |

**Author contribution**

A.K. performed the photogrammetric analyses and wrote the paper. T.S. and R.C. performed the radar interferometric analyses and wrote the paper. T.B. assisted with analyses and wrote the paper. H.T. assisted with photogrammetric analyses. M.S. contributed to discussions and edited the paper.

10  **Competing interests**

All authors declare that they have no competing interests.



**Acknowledgements**

This work was funded by the ESA projects GlobPermafrost, Permafrost-cci, Glaciers_cci, and the ESA EarthExplorer10 Mission Advisory Group, and the European Research Council under the European Union's Seventh Framework Programme. We are grateful to the providers of free data for this study; European Space Agency (ESA) / European Commission (EC)

Copernicus for Sentinel-1 data, ESA for ERS-1/2 and ALOS-1 PALSAR-1 data, the German Aerospace Center (DLR) for TanDEM-X data, the Japan Aerospace Exploration Agency (JAXA) for ALOS-2 PALSAR-2 data, and Planet for their cubesat data via Planet's Ambassadors Program. ERS-1/2 and ALOS-1 PALSAR-1 data were provided by ESA, courtesy of C1F.6504. ALOS-2 PALSAR-2 images were provided by JAXA, courtesy of RA4-1058. TanDEM-X data was provided by DLR, courtesy of Wegmulle_NTI_INSA3397. We are grateful to CNES/Airbus DS for the provision of the Pléiades satellite data

within the Pléiades Glacier Observatory facilitated by Etienne Berthier (LEGOS) and Delphine Fontannaz (CNES).

**Financial support**

This work was funded by the ESA projects GlobPermafrost (40001161196/15/I-NB), Permafrost_CCI (4000123681/18/I-NB), Glaciers_CCI (4000109873/14/I-NB, 4000127593/19/I-NS), and the ESA EarthExplorer10 Mission Advisory Group (4000127656/19/NL/FF/gp), and by the European Research Council under the European Union's Seventh Framework

Programme (FP/2007-2013) / ERC grant agreement no. 320816.





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
