# Peer review of "Inventory and changes of rock glacier creep speeds in Ile Alatau and Kungöy Ala-Too, northern Tien Shan, since the 1950s"

_The Cryosphere, 2020_

## Referee Comment (RC1) · Alessandro Cicoira (Referee) · 25 Aug 2020

Cicoira Alessandro (Referee) alessandro.cicoira@unifr.ch

General comments:

This manuscript investigates the distribution and the motion of a large sample of active rock glaciers (551 landforms) in norther Tien Shan. The rock glacier inventory is not exhaustive and does not include inactive and relict landforms. In addition, the

movement of some other landforms, such as debris-covered glaciers and ice-cored moraines, have been classified (900 landforms in total). A combination of satellite based radar interferometry and feature tracking of optical imagery allows the classification of activity classes for all the landforms mapped. For six rock glaciers the authors also perform the calculation of inter-annual variations in surface speed over a period of almost 70 years. By doing so, they provide the first long-term regional investigation of rock glacier motion in the Tien Shan. The methodology is thoroughly described and the uncertainties properly addressed, and the authors discuss the limitations of the manuscript, overall with high scientific rigor. The conclusions are reasonable and coherent with previous research. However, the conclusions are supported by little and at times controversial evidence and should be discussed with more caution. A strong link to Sorg et al. (2015) is evident and well explained in the introduction, but the discussion is only briefly addressing the relation between the two manuscripts. In general, a more detailed discussion would help placing the manuscript in the context of current research and highlight its novelty (extension of rock glacier kinematic observations in the Tien Shan massive). Several minor revisions and a few possible additions to the manuscript are suggested in the specific comments below. Concluding, I consider the manuscript well suited for publication in The Cryosphere after minor revisions.

Specific comments:

Page 1, Line 23: Please delete "very" in "very high resolution data".

Page 1, Line 23: Most of them reads three out of six. It would be very interesting to include an analysis of the regional scale evolution of surface speed, which would largely improve the confidence in the general conclusions. This might be possible on the basis of the readily available InSar or optical data. The resolution of such an analysis might be lower compared to the one provided for the six study cases investigated in detail. If this analysis is not done, it should be made clear in the text that the conclusions are still rather speculative and based on preliminary results and similitudes to other regions (Alps).

Page 1, Line 26: The comparison between rock glacier and glacier sediment transfer is poorly constrained in the manuscript. Please provide more information and a context. Please add some relevant references from the literature, and extend the discussion. This might require some additional analysis and a quantitative regional-scale comparison of the two contributions.

Page 1, Line 27: The relation between the stress regime (compressive flow regime) and changes in speed upstream is mentioned, but only superficially discussed, without any reference to rock glacier dynamics and their mass fluxes.

Page 1, Line 28: The conclusion that the Gorodetsky Rock Glacier does not show an acceleration due to the decoupling from its rooting zone contradicts the previous point. It also partly contradicts the hypothesis that rock glacier acceleration is due to warming air and ground temperatures. The two mechanisms are not exclusive, but their interaction is not straight forward and requires a more detailed discussion in the manuscript (and possibly additional analysis).

Page 1, Entire Abstract: I suggest to rewrite the abstract in a more concise and specific fashion, in order to highlight the novelty and the merit of the manuscript.

Page 2, Line 2: Rock glacier move due to the deformation of debris-ice mixtures (as you also mention several times later in the text). Please correct this first sentence, which now suggests that they move due to the deformation of the frozen debris only.

Page 2, Line 3: Please correct the repetition: "rock glaciers, or (rock glaciers)." I can imagine that it is a typo for the one-word version "rockglaciers".

Page 2, Line 10: Due to the ongoing debate about the genesis of rock glaciers and the origin of their constitutive material, I find the text insufficient for an introduction. I would avoid confusion and keep this topic out of the intro, also because it is not needed nor addressed further in the manuscript. If the authors believe that this is essential for their manuscript, I suggest that they extend the introduction and address the topic in more

detail.

Page 2, Line 17: The concept of destabilization might be easily misunderstood and confused for structural instabilities. This being not the case, some additional explanation might be needed to present the concept of rock glacier destabilization. However, here again, I suggest to omit this point, because it is not needed in the manuscript. Alternatively, introduce it in more detail.

Page 2, Line 21: please add citations to Arenson and Springman (2005)a-b.

Page 2, Line 25: please be careful when considering the different processes. The response of permafrost creep to increasing temperature is thought to follow a power law. In addition, the role of water can enhance, also in a non-linear fashion, the response of creep to temperatures approaching melting conditions.

Page 2, Line 26: please consider the following references, which are in my opinion amongst the most important publications regarding this topic: Ikeda et al, (2008), Buchli et al, (2018) and Cicoira et al, (2019)b.

Page 2, Line 29: please specify "impacts".

Page 2, Line 30: inventorying rock glaciers is typically done on the basis of a combination of optical and topographical data. Kinematic information (e.g. InSar) remains an optional and non-sufficient data source. Please be more precise in the text. The authors might refer to the Baseline Concepts for inventorying rock glaciers which are currently being elaborated within the International Permafrost Association IPA.

Page 3, Line 19: please be more specific in the wording. "followed" is not a technical term and might be misunderstood. I suggest to use the key-terms: qualitative – similar patterns, statistical correlation, phase lag, thermal offset, non-linear.

Page 3, Line 20: The influence of temperature forcing through heat conduction on rock glacier dynamics has been quantitatively investigated in detail in e.g. Kääb et al, (2007) and Cicoira et al, (2019)a-b. The authors might want to discriminate between qualitative

and quantitative studies that have investigated the processes controlling variations in rock glacier creep and include the state-of-the-art knowledge on the topic.

Page 3, Line 21: The influence of variations in ground temperature through melt water advection has been shown to be negligible for the case of the Furggwanghorn in Buchli et al, (2018) and for the Ritigraben Rock Glacier in Cicoira et al, (2019). I am not aware of any study where the hypothesis (in the submitted manuscript) was tested on the basis of observational data nor modelling studies. If such study exist, please include the reference in the text in an explicit fashion. The study of Ikeda (cited in the text), also concurs to the hypothesis that rock glacier creep is controlled by variations in the effective stresses, rather than variations in ground temperatures (being these close to the melting point).

Page 3, line 21: as a general comment, I see the need of general revisions of the text about the processes controlling rock glacier creep.

Page 3, Line 24: At this point, the reader would expect temperature, precipitation and snow cover data to be analysed along the creep rates. I believe that the study presents enough new insights and does not need this additional step, but I suggest the authors to explain why it has not been done.

Page 4, Line 15: please specify the depth of the ground temperature measurements.

Page 5, Line 7: please consider replacing "rock" with "boulder" in all the appropriate cases.

Page 5, Line 8: add a point at the end of the sentence.

Page 5, Line 24: Please explain the reasons for the choice of the six rock glaciers.

Page 6, Line 1: I am not sure what "frozen snow" means. Please consider replacing this formulation with a more specific terminology.

Page 6, Line 12: The choice of assigning a polygon to the next class when the velocities

are close to its upper limit seems unjustified to me. Also, what is the reasoning behind the division for the first classes (0-2, 2-10)? The next two classes are (half) an order of magnitude, so I wonder why also the first two are not consistent.

Page 6, Line 13: please specify the nature of the mentioned variations. (spatial variations?)

Page 6, Line 15: is there a reason why the vector of the observed displacements has not been corrected according to topography? Maybe extend on this point.

Page 6, Line 23: is this sentence a list of the criteria used for identifying and locate the rock glaciers? Currently, the sentence is somehow lost in the text. Please consider rephrasing it.

Page 6, Line 24: short-term variations in rock glacier velocity have been observed at many sites worldwide. Seasonal and even weekly oscillations are observed consistently. The statement is therefore unjustified. I suggest to support it with specific evidence for the study area (if available) or to discuss it in more detail. See Haeberli, (1985), Wirz et al, (2016), Strozzi et al, (2020).

Page 6, Line 25: it is now not clear to me weather the analysis has been conducted over multiple time steps. Maybe this has not been explained clearly enough in the text, or I just misunderstood it here.

Page 6, Line 19-30: is this paragraph a list of the criteria used to classify a rock glacier and distinguish it from a rock glacier? Please be more specific and explain in detail the concepts that have been used. I suggest to reference to the ongoing action group on rock glacier inventories and kinematics of the IPA.

Page 8, Line 11: please explain why the measure of accuracy was performed on stable ground only for three of the six field sites.

Page 8, Line 18-24: The results of this very interesting analysis are only briefly described in the manuscript. Also the discussion seem to me not sufficient to address

this point. I suggest to include more details in the manuscript.

Page 10, Line 10: please specify that only (part of ) the labelled rock glaciers are further investigated in the photogrammetric analysis, and not all the visible polygons in the figure.

Page 10, Line 10: please consider indicating the value of the wavelength for the data in the figure.

Page 10, Line 10: it is impossible from the figure to distinguish between the different classes of the polygons. This information is present only in Fig. 1 at a very low resolution. Consider improving the level of detail in this (Fig. 2) or in the following figures (Fig. 3-7).

Page 11, Line 3: consider replacing "photogrammetric velocities" with a more detailed terminology. (such as "Surface velocities calculated by offset tracking").

Page 11, Line 4-12: the determination of the origin of the ice and sediment constituting the rock glacier requires more than the observation of spatial connection, or as in this case, the (legit) supposition of past spatial connection. The state of inactivity of the current glacier forefield (called in the text "zone between glacier and main rock glacier") only shows that the connection is not currently present, but is not sufficient to imply that this (dynamical- and sedimentological) connection was present in the past. Even in this case, it would have been limited to the period when the glacier advanced to its maximum (LIA). No information on the climatic and sedimentological setting of the rock glacier is provided in order to commence such an analysis. Without entering in too much detail, I suggest to limit the discussion to the spatial connection (glacier forefield connected, according to Delaloye and . . . 2018) and avoid speculation regarding the "nourishment" and genesis of the landform.

Page 11, Line 15: please provide quantitative evaluation of the observed trend and its statistical significance.

Page 11, Line 18: as above, consider removing the concept of "nourishment" from the paragraph.

Page 11, Line 19: consider removing "striking" or replace it with a more technical and specific adjective.

Page 11, Line 25: please describe in more detail the differences between the surface speed for this early period.

Page 11, Line 27: I suggest to improve this paragraph and give a better summary of the results for this analysis. Also, write explicitly what the calculated ice-content would be. It is implicit in the ratio, but the reader might be helped by some repetition here. Consider calculating it for all the available time steps and include an estimation of the uncertainty in the results.

Page 14, Line 11: In figure 4c, the debris-covered glacier and (for what I can see) the glacier forefield are not shown and it is impossible to verify the presence of a material flux from the rock wall to the glacier. As previously, I consider the observational evidence insufficient to support the statement. I suggest to simply avoid the point, which is in my opinion not important for the present manuscript, or to include more data and analysis to support this thesis.

Page 14, Line 24: the deformation profile (on the vertical dimension) also has an important influence on this calculation. Why is it not mentioned here? Please consider spending some words about this point to make the text clearer.

Page 15, Figure 5: it would be interesting to see the displacement on stable terrain.

Page 16, Line 2: what is the mass flux at the boundary between the glacier and the rock glacier? What is the absolute value of the surface speed? It is very hard to see this from the figure provided. In general, a similar comment as for the nourishment above.

Page 18, Line 15: here the authors state that the sediment transfer between glacier and

rock glacier cannot be determined with the available data. I agree with the conclusion, but still, I would like a more quantitative discussion. Otherwise, I suggest again to completely avoid this point. A possible analysis would investigate the relation between acceleration and max fluxes along flow lines.

Page 18, Line 15: more information about the vegetation could be interesting. What is the size and what are the species growing on the rock glacier? Is this information available from field expeditions?

Page 18, Line 21: the fact that the observed signal from feature tracking is lower than the noise does not allow to conclude that the rock glacier has accelerated. I don't understand why the authors mention "statistical significance" in their argument.

Page 22, Line 1: I agree with the statements, but I would like a more quantitative evaluation of the different error types.

Page 23, Line 2: I agree with the authors. Still, it would be interesting to see the values of the errors on stable ground.

Page 23, Line 7: the divergence of a vector field is a well defined term and as far as I understand this is not what the authors mean. I suggest to replace this substantive with a more pertinent description of the differences observed in the velocity field.

Page 23, Line 10: I have not found any values for the statistical analysis of the velocity time series. I warmly suggest to implement this analysis in a quantitative way. If the authors prefer not to, I would be much more careful talking about statistical significance.

Page 23, Line 25: it would be valuable to have a better quantification of the "strong compressional regime" by means of e.g. strain rates (accompanied by proper interpretation or even with the calculation of the internal stresses).

Page 23, Line 27-29: First, it would be very welcome to see the original results for the upper part. Second, I don't agree with the statement that the lower part responds passively. It is a matter of dynamics, thus of mass and momentum fluxes. I suggest

to change the formulation, possibly mentioning that the dampened response (I would appreciate an illustration of this) is due to topographical setting and the corresponding dynamic behaviour of the rock glacier. (In detail it could be that the mass flux is mostly compensated by variations in thickness or in mass input rather than variations in velocities, but such a statement should be supported by more evidence.)

Page 23, Line 29: please repeat the advance rate and the surface velocity, and consider discussing the result in more detail – also with a possible range of quantitative values of the ice content.

Page 23, Line 32: given the strong similarity between the two publications, this point might require some more discussion. I would suggest also one or two figures in the appendix or some additional comparisons.

Page 24, Line 10: If I understand this correctly, it means that it is not possible to conclude which one of the two studies is more accurate. If this is the case, please state it more clearly in the text.

Page 24, Line 24: consider citing Cicoira et al., 2019b, where it has been shown that the seasonal and inter-annual variations in rock glacier flow are mostly controlled by variations in snow melt rates and liquid precipitation, rather than in air temperature. Other very relevant citations are Buchli et al., 2018 and Ikeda et al., 2008.

Page 25: in general, in the "5.3 Speed time series" paragraph, more discussion relative to the results and their validity would enhance the validity of the manuscript. Most of the discussion is a very precise comparison to Gorbunov et al., (1992), which could be probably summarized in one or maximum two paragraphs. I suggest to highlight more the originality of the manuscript and discuss better its strength and weaknesses.

Page 25, Line 26: I don't see the link the negative glacier mass balance. Please avoid this point or argument in more detail the linking mechanism.

Page 25, Line 29: it would be very interesting to quantify this sediment transfer. I

suggest using a simple assumption for the rock glacier thickness (e.g. constant value of 20 meters, see Cicoira et al., 2020) and estimate the overall ice/sediment transport rates for the periglacial environment. This would be a major result, and is not very difficult to calculate (although the uncertainty will be large).

Page 25, Line 31: such a statement definitely requires a quantification of both the sediment transfer.

Page 26, Line 4: this point is very interesting but insufficiently discussed. I suggest the authors to implement it both in a qualitative and in a quantitative fashion.

Page 26, Line 5: This statement appears unjustified to me. As far as I know, no quantitative (and conclusive) evidence that a rock glacier derived from a glacier exist. As for this manuscript, there is not sufficient evidence supporting the statement. Often, an interaction (more or less important) has happened during the LIA. I suggest to rewrite this last paragraph with more focus on the novelty of the manuscript (it is not the geomorphological genesis of the rock glaciers).

Page 26, Line 18: it is not so clear to me which assumptions. Please be more explicit in the conclusions.

Page 26, Line 26: the time scale considered in the manuscript is (almost) only decennial. I would rephrase this sentence and highlight the fact that the original observations in speed also show past periods of acceleration at the investigated temporal scale.

Page 27, Line 4: I am not completely convinced by this statement. The quantification of the trends and their significance for each rock glacier might make this point more convincing and increase the confidence in the results and the conclusions.

Page 27, Line 10: I don't agree at all with this statement. This is very speculative and not supported sufficiently by the evidence provided in the study. As above, I suggest to discuss it in more detail or to avoid this point, which is in my opinion not relevant for the manuscript.

Page 27, Line 10-15: on the contrary I welcome the topic as an outlook for future studies. With this last comment, I thank the authors for an interesting piece of research.

References:

Arenson, L.U., Springman, S.M., 2005a. Mathematical descriptions for the behaviour of ice-rich frozen soils at temperatures close to 0âŮęC. Can. Geotech. J.42, 431–442. https://doi .org /10 .1139 /t04 -109.

Arenson, L.U., Springman, S.M., 2005b. Triaxial constant stress and constant strain rate tests on ice-rich permafrost samples. Can. Geotech. J.42, 412–430. https://doi .org /10 .1139 /t04 -111.

Cicoira, A., Beutel, J., Faillettaz, Vieli, A., 2019b. Water controls the seasonal rhythm of rock glacier flow. EPSL 528, https://doi.org/10.1016/j.epsl.2019.115844

Cicoira, A., Marcer, M., Faillettaz, J., Gärtner-Roer, I., Bodin, X., Arenson, L. U., Vieli, A., 2020. A general theory of rock glacier creep based on in-situ and remote sensing observations. PPP (in review).

Delaloye et al., 2018. Rock glacier inventories and kinematics: a new IPA Action Group. EUCOP5. Buchli, T., Kos, A., Limpach, P., Merz, K., Zhou, X., Springman, S.M., 2018. Kine-matic investigations on the Furggwanghorn Rock Glacier, Switzerland. Permafr. Periglac. Process.29, 3–20.

Haeberli, W., 1985. Creep of Mountain Permafrost. Mitteilungen der Versuchsanstalt für Wasserbau, Hydrologie und Glaziologie der ETH Zürich, vol.77.

Ikeda, A., Matsuoka, N., Kääb, A., 2008. Fast deformation of perennially frozen debris in a warm rock glacier in the Swiss Alps: an effect of liquid water. J. Geophys. Res., Earth Surf.113. https://doi .org /10 .1029 /2007JF000859 .f01021.

Wirz, V., Gruber, S., Purves, R.S., Beutel, J., Gärtner-Roer, I., Gubler, S., Vieli, A., 2016. Short-term velocity variations at three rock glaciers and their relationship with

meteorological conditions. Earth Surf. Dyn.4, 103–123. https://doi .org /10 .5194 /esurf -4 -103 -2016.

---

## Referee Comment (RC2) · Philippe Schoeneich (Referee) · 28 Sep 2020

It is a very high quality paper, written by some of the best specialists of the topic and of the methods used. The paper provides a valuable and comprehensive dataset, on an area for which few data were available so far.

The work presented uses a combination of various methods and various types of optical and satellite imagery. A combination that was rarely achieved at such a level in a single study. This allows a cross-checking and validation of the results, which therefore appear as very robust. The use of image archives allows a back-analysis from the 1950s onwards and the interpretation of the evolution of velocities over time. If this

is not new, the above mentionned combination of various image types and methods allows a higher and more detailed temporal resolution, and reveals short lived velocity changes that could not be observed by using a single type of images.

The paper provides therefore an innovative contribution both for the results and the methodological approach.

Therefore, there are no fundamental comments, and the paper should be accepted with the minor improvements listed below or proposed by others.

p. 4 – l. 6 Âń Âă landslides and rock avalanches Âă Âż Âă: there is a problem of vocabulary. In English, the word Âń Âă landslide Âă Âż is used as generic term for all types of mass movements on slopes, and thus includes rock avalanches. The authors probably wanted to distinguish slides (= Âń Âă Rutschungen Âă Âż, Âń Âă glissements Âă Âż) from rock avalanches. The adequate term in this case is Âń Âă slide Âă Âż. If they mean deep seated slope movements (= Âń Âă Sackungen Âă Âż, Âń Âă Talzuschub Âă Âż, Âń Âă tassements Âă Âż, Âń Âă glissement rocheux Âă Âż) the most adequate concept would be DSGSD (for Deep Seated Gravitational Slope Deformation). As generic term for designating all mass movements, we recommend to use Âń Âă mass movements Âă Âż instead of Âń Âă landslide Âă Âż, the latter term causing much confusion among french or german speaking readers.

Fig. 1 + part 4.1 Âă: the figure legend indicates 1 Âń Âă landslide Âă Âż (see also comment above), but this is not mentionned in the text.

Fig. 1 is small and hardly lisible. It should be provided in full format as downloadable supplementary data

Figures 3, 4, 5, 6 and 7 Âă: the velocity color scale is not the same in all figures, which hinders a direct comparison of the different cases and can lead to misinterpretation. In figure 4 for instance, at a first and quick view, the Morenny RG could appear as slower than the Archaly RG, which is not the case. A common velocity color scale would be

better.

p. 24, l. 20-25 Ăă: you mention the influence of snow thickness/insulation and amount of meltwater as factors possibly explaining the acceleration in the 1960s. I agree with this general statement, but it can be refined. The data series of the Laurichard rock glacier (see Bodin et al. 2009, you already have in your references) and ground surface temerature measurements at many places, show that the most relevant factor is the onset date of the snow cover. An early onset, for instance in October, before the coldest days of November-December, prevents the ground from cooling and keeps the accumulated summer heat in the ground. On the opposite, a late onset of the snow cover, by end of December of even January, will allow a strong cooling of the ground surface and a deep seasonal frost in non permafrost areas. The Laurichard time series shows that the reaction is assymetric Ăă: it needs several Âń Ăăwarm ĂăÂż years (years allowing a warming of the ground, with hot summer and/or early snow cover) to induce an acceleration, but a single Âń Ăăcold ĂăÂż winter (with late onset of the snow cover) is sufficient to reduce velocities. Do your meteorological data allow to establish the onset date of the snow cover (insulation is provided with a minimal thickness in the order of 60-80 cm) Ăă? If yes, it could reinforce your interpretation.

Author contributions Ăă: there are two AK among the authors Ăă! Who is AK Ăă? And what was the contribution of the second one Ăă?

Acknowledgements Ăă: the two first lines are duplicated in the Financial support section. Remove from here, and leave under Financial support.

References

Check formatting of abridged journal names Ăă: - for PPP Ăă: your text mentions Permafrost Periglac. (Processes is missing) – numerous references - p. 35 Kääb et al. 2017 Ăă: NHESS (Sciences missing)

p. 33 - Gorbunov 1983 Ăă: volume missing
p. 33 - Haeberli 1985 Ăă: volume number in the collection missing

p. 34 – Kaufmann 2012 Ăă: journal missing

p. 36 – Roer et al. 2008 Ăă: incomplete (volume, pages, ...)

Please also note the supplement to this comment:
https://tc.copernicus.org/preprints/tc-2020-109/tc-2020-109-RC2-supplement.pdf

---

## Referee Comment (RC3) · Line Rouyet (Referee) · 2 Oct 2020

Kääb et al. inventoried active rock glaciers and other periglacial landforms in Tien Shan using InSAR and further analyzed the spatial-temporal patterns of six fast-moving glacial-derived rock glaciers using optical offset tracking. They present a nice piece of work that shows the value of radar and optical remote sensing for mapping and monitoring creeping permafrost landforms over a large and hard-to-access areas. The study focuses on a region that had not been intensively documented before. The contribution is well suitable for a publication in The Cryosphere and likely to become a reference in remote sensing of mountain permafrost.

[Figure]

I have no major concern regarding the way the study has been designed but I think the manuscript could benefit from some clarifications about the inventory procedure, a better visualization of the results and an extended discussion on the findings. Here I develop three main elements. I also listed some complementary suggestions of correction at the end of the review.

–Main comment 1. Inventory and movement classification–

For the movement types, I think it should rather be named 'landform types' as e.g. a rock glacier or a moraine is not a process but a landform that has been shaped by a periglacial or glacial process. Consider also renaming 'solifluction / debris movement' (not sure what debris movement means; a rock glacier is also composed of debris. Scree deposit?). 'Dead ice / subsidence' could just be 'thermokarst'?

About the velocity classes in the inventory: The choice of the velocity classes needs at least to be explained. 0-2, 2-10, 10-50, 50-100, >100 does not look so intuitive to me. Why not equal intervals: e.g. 0-25, 25-30, 50-75, 75-100. Or 0-3, 3-10, 10-30, 30-100, >100? Or just an order-of-magnitude (cm/yr, cm-dm/yr, dm/yr, dm/yr-m/yr, m/yr), as now recommended by the IPA action group 'rock glacier inventories and kinematics'? The criteria for defining the class are also not fully clear: when writing that 'if two or more classes were present during the observation time-span, the higher displacement rate was used to determine the velocity class' (l.14-15, p.6), does it mean that a little fast-moving part of the rock glacier could lead to classify the whole landform as for ex. >1m? If so, that sounds a bit bold to me. Maybe it should just be acknowledged in the discussion that it could lead to a rate overestimation. Similarly, at l.3-4 (p.7), it is acknowledged that the max. measurable rate for a C-band with 12d repeat-pass is 85 cm/a. To my understanding, a decorrelated part in a 12d-interferogram means >85 cm/a, but so not necessarily >100, right? In these cases, were the 1d ERS interferograms used to ensure a correct classification? If yes, it could just be mentioned.

–Main comment 2. Figures–

Most of the figures could benefit from slightly more job to clearly disseminate the findings. Considering the volume of work that the study represents, it is a bit a pity if the presentation does not fully help to maximize the understanding of the results.

Figure 1: This map is hard to read. I understand the wish to be comprehensive, but it is most likely too much information in once and the color contrast does not allow for differentiating the velocity classes for specific landforms. I would suggest making two maps: one with only the different movement types, without any velocity classes. And another focusing on rock glaciers (the main point of the article) with a different color for each velocity class (for ex. green to red, blue to red). A similar map for the other landform types can potentially be placed in Supplementary.

Figure 2: Add the equivalent in cm of the phase color scale. Missing information about SAR geometry (LOS arrow or mention of the ascending/descending geometry in legend).

Figures 3-7: It would be more comfortable for the reader if the optical images always had the same extent and scale than the velocity field maps. The error bars (especially the red) are hard to see due to the transparency. The arrows of the photogrammetry results are sometimes impossible to see and interpret (Fig. 5-7). Less importantly: it could be nice to have altitude references at some locations on the optical images.

Figure 8: Add the equivalent in cm of the phase color scale. Missing information about SAR geometry. Here, as the background map from the interferogram is not directly related to the size of the arrows (contrary to Fig. 3-7), it would be good to add a legend of the arrow length. It is a really nice figure to show the agreement of both methods, as well as the value of combining them. What about adding the same for the five other rock glaciers (potentially in Supplementary)?

–Main comment 3. Time series and acceleration–

There is something that does not completely add up in the interpretation of the tem-

poral variations: contradictory information that gets the reader a bit lost at the end of the paper. The title includes 'acceleration of rock glaciers' and it is presented in the abstract that five of the six temporally investigated rock glaciers exhibit acceleration (l.23-24, p.1). However, at several locations, the authors acknowledge that the statistical significance is not always good enough to confirm the trends (several cases with std dev of speed differences greater than the actual measured difference, e.g. Fig. 7c). As explained at l.28-29 (p.24), 'Gorodetsky, Morenny and Archaly rock glaciers [...] are not showing a clear increase of speed over the study period'. Gorodetsky, Moreeny and Archaly are three out of six rock glaciers, which does not fit anymore with the findings' summary in the abstract (l.28, p.1) or the conclusion (l.4, p.27). Maybe I misunderstood something, but in that case, it is probably possible to be clearer in the text.

The statement 'the behaviour observed in the study area confirms findings [...] that rock glacier creep speed increase overall under atmospheric warming' (l.23-24, p.25) sounds a bit too bold to me considering that no comparison with temperature is provided in the study. Other elements could be discussed in greater details. In 5.4, the authors remind that 'all rock glaciers studies in detail in this study are derived from contemporary or former glaciers' (l.5, p.26). Some fuzzy thoughts here: When did the glacial direct connection expected to have stopped? During the time span of the study (50th-70th)? If yes, are some of the variations related to the glacial dynamics instead of the permafrost creep, or due to the transition between the two flow types? What if you had also analyzed talus-derived and slower landforms? Not to say that it should be done here, but it could be discussed as a prospect. Some of the conclusions here are maybe only valid for glacial and extreme (>1m) cases.

I don't think this comment requires critical changes. The results are well described in Section 4, but the discussion could be improved (especially 5.4) and the abstract/conclusion (and even the title?) better matched the actual findings.

—

–Complementary comments–

Page 1:

- Title: 'acceleration'. Maybe consider 'Inventory and spatial-temporal trends of. . .' See main comment 3.

- l.23 and l.28: 'most of them'. . . 'The only rock glacier'. . . Maybe just write the actual number in both cases: five of them – the only, if this is really the case. See also main comment 3

- l.30: maybe missing a general sentence in the abstract about what does it tell, what is the relevance of the study in a broad sense?

—

Page 2:

- l.3: rock glaciers (or rock-glaciers) I guess

- l.9: 'In a similar way, the ice content [. . .] can be directly incorporated. . .'

- l.19: As rock glacier distribution is also a valuable environmental indicator, I would suggest removing the four first words of the sentence.

- l.21: '. . .although it is also influenced by. . .'

- l.22: '. . .temporal and spatial variations of ice content. . .' I guess lateral variations are as important as vertical?

- l.25: '. . .variable around thawing conditions'. Maybe 'sensitive' instead?

- l.25: '. . .i.e. in cases the ice content starts to degrade. . .'

- l.29: 'impacts from spatio-temporal variations'. Impacts on what in this context?

—

Page 3:

- l.3-5: Start of the sentence ('in addition to . . .,') does not sound necessary to me. Instead, maybe: 'The categorization of landforms into classified movement rates or orders of magnitude (e.g. cm/day, . . ..)'

- l.8: (more details below) is not really useful.

- l.17: 'constant speeds' instead of 'stable speeds'?

- l.19-20: repetition in these two sentences (creep velocity <-> temperature)

- l.24: 'for the first time' sounds a bit weird considering the extensive literature presented in review 2 and reference to Sorg et al. article. What about 'The aim of the present study is to systematically inventory contemporary rock glacier crepp velocities. . .'

- l.27-30: last two sentences are maybe not necessary.

—

Page 4:

- l.5-6: potential to rephrase here 'with the largest of magnitude8.0 and more [. . .] and stronger earthquakes. . .' Stronger than 8 and more? . . .

- l.9-10: Sth unclear with that sentence: '. . .and for elevations of about 3000 m asl.' Does it mean 'For elevations of about 3000 m asl, mean annual precipitation exceeds 1000 mm on the northern slopes and is less than 800 mm on the southern ridges.'?

- l.18: larger than what? '. . .permafrost is very likely although the fronts of the large rock glaciers are located. . .'

- l.25: 'inactive ones': maybe wise to define here the activity categories. What about relict landforms? Are they considered in the study? In the activity morphologically determined or using InSAR/photogrammetry?

- l.33: 'Blöthe et al. (2019) mapped...'

—

Page 5:

- l.7: 'Displacement rate of individual blocks vary greatly...'

- l.9: 'blocks'? Replace 'down-valley' by 'slope direction'?

- l.17-18: New sentence at 'Rock glacier activity/tree growth anomalies...'? Respectively? To what? Not clear.

- l.21: 'different statuses of the rock glaciers on their path towards...' -> 'different stages'?

- l.25: remove 'for the offset tracking work realized here.'

- l.30: 'short spatial baseline and time intervals between 1 day and three years...' According to your table A1, the max. interval is over one year.

—

Page 6:

- l.5: I guess $\frac{1}{4}$ of a phase cycle' will be abstract for most of the readers. Maybe add what it means in terms of displacement?

- l.9: I think that 1-2 sentences could added here to quickly explain the principle. Count the fringes on coherent parts and relate it to velocity considering the wavelength interval between pairs + use decorrelation information. In the paper, you show only 12d Sentinel examples, but it is important to be clear that you used a large stack with different wavelengths/intervals/periods to get different detection capability and avoid misinterpretation of short-term velocity variations. Someone who does not know the technique could misunderstand if not quickly explained here.

- l.11: Why these classes? See main comment 1.

- l.14: 'the highest displacement rate. . .' See also main comment 1.

- l.15-16: 'We empirically considered the line of sight in the assessment': It could be explained what it practically means. If the slopes significantly deviating from LOS and between two classes, the highest cause it is considered as underestimated?

- l.16-18: last sentence of paragraph could be moved after first sentence of the next one (l.20).

- l.25: Check this sentence. I doubt you get coherent information on supraglacial ponds.

- l.27-28: Similar appearance to what? Check this sentence. I don't see why solifluction would give spot-like fringe patterns. Also: debris movement does sounds like right terminology. See main comment 1.

- l.30: Redundant. Remove 'also optical images were not of help due to'.

- l.32: Consider using everywhere 100 cm/yr instead of 1 m/yr, as in Fig.1

—

Page 7:

- l.2-3: >85 could also mean 90, which is in 50-100 class. See main comment 1.

- l.30 and 32: 'the above': not necessary.

—

Page 8:

- l.4: thresholds on correlation coefficients -> the actual values of the thresholds could be documented.

- l.12: the point (ii) sounds odd to me: looking at the chosen areas on the figures, they are significant velocity variations that are likely to be natural (for ex. C on Fig.6). Not possible to select areas with more similar patterns? What about the correlation

coefficients, the average correlation cannot be used as an accuracy estimate?

—

Page 10:

- Legend Figure 2: 'Wrapped orthorectified interferogram'? Raw sounds weird, and not especially right considering the corrections described in Section 3.1.

—

Page 11:

- l.5: 'agree well with the interferograms'. Plural? To emphasize that several interferograms have been investigated.

- l.20: 'they show coherent fringes'

- l.24: 'the central part of Morenny rock glacier speeds are in the order of 1 m/a: It does not look like that in Fig.4, most of the rock glacier is blue and the averaged time series show values clearly under 1, even for the highest parts of the series...

—

Page 14:

- l.16: see main comment 1: in theory, noise could also mean 90 cm/yr, so can we really say consistent?

- l.24: 'This points to block movement of the entire rock glacier column...' Sounds odd, potential for rephrasing.

—

Page 16:

- l.6: ref. to Fig. 6e

- l.12: 'because of the many different frontal parts' -> 'because of heterogeneities'?

—

Page 18:

- l.19-20: I don't see the point of adding zone B if the results are not shown. Either the tseries are shown somewhere (potentially in Supplementary) or you maybe don't mention this part at all? About 'changes documented for zone B [..] are within their error bars': looks like there is the similar problem for A since the 90th (Fig.7 c).

—

Page 19:

Figure 7: Remove area B) if not used? See also main comment 2.

—

Page 20:

- l.4: 'reasonably very' -> just reasonably?

- l.6-7: 'all rock glaciers [...] were classified correctly from the interferograms': does not sound right for Morenny (see comment page 11, l.24)

- l.22-23: see main comment 1: so 85 cm/yr is approximate as 1m/yr?

—

Page 22:

- l. 2: missing space 'very-highresolution'

- l.4: 'more difficult to quantify precisely': Sounds weird cause I don't think biases a) have been quantify in the manuscript.

- l.6: 'could translate 1:1 to a lateral offset': without being an offset-tracking expert, I

must say that I don't get this.

- l.24-25: if median values are safer, why not have used these for the point clusters?

- l.33-34: here comes quickly what I commented for p.8, l.12: maybe it could come before in the methods?

—

Page 23:

- l.23: 'acceleration between 1964-2017'

- l.26-29: make two sentences. For ex: the lowest rock glacier part deforms under a regime. . . It responds rather passively to. . .

—

Page 24:

- l.5: 'the latter part, under a strong compressive creep regime, reaches and diverts the river. . .'

- l.28: 'are the only systems'. 3 out of 6. . . sounds weird to say 'the only'. See main comment 3.

—

Page 25:

- l. 24: maybe 'is aligned with findings' instead of 'confirms'?

—

Page 26:

- l.11: 'an inventory of active rock glaciers and other periglacial landforms'? or 'a slope movement inventory including rock glaciers and other periglacial landforms'?

- l.16: well, maybe it is me being picky, but 85 cm is not a meter. And again, what about the use of ERS 1-day? See main comment 1.

- l.28: 'A possible explanation [. . .] is. . .'

—

Page 27:

- l.2: '. . .more passively to...'

- l.4: Not in line with l.28-29 p.24. See main comment 3.

- l.10-12: this could be more discussed in 5.4. See main comment 3.

- l.12-15: The last sentence is too long and hard to understand. I would suggest as prospect to include not only glacial-derived and landforms with lower averaged velocity, to confirm the accelerating trend.

—

Page 29:

- Table A1: Why no 2017 Sentinel-1 image?

---

## Author Comment (AC1) · 11 Dec 2020

The Cryosphere        tc-2020-109
**Inventory, motion and acceleration of rock glaciers in Ile Alatau and Kungöy Ala-Too, northern Tien Shan, since the 1950s**
Andreas Kääb, Tazio Strozzi, Tobias Bolch, Håkon Trefall, Markus Stoffel, Alexander Kokarev

**Response to referees**

**General response**

We would like to thank the three referees for their positive, very constructive and detailed reviews that will certainly help to improve the paper!

We agree with most of the comments made (as detailed below), and will modify our manuscript accordingly. In summary: we will clarify/modify descriptions and give at a number of places more details; we will balance the conclusions better with respect to the measurement results, in particular considering their statistical significance; and we will avoid discussions around glacier feeding of rock glaciers.

Referee comments are in *italic*, and our response and what kind of revisions we plan to do in normal font.

**Response to individual referees**

Referee #1     Alessandro Cicoira

*General comments:*
*This manuscript investigates the distribution and the motion of a large sample of active rock glaciers (551 landforms) in norther Tien Shan. The rock glacier inventory is not exhaustive and does not include inactive and relict landforms. In addition, the movement of some other landforms, such as debris-covered glaciers and ice-cored moraines, have been classified (900 landforms in total). A combination of satellite based radar interferometry and feature tracking of optical imagery allows the classifi- cation of activity classes for all the landforms mapped. For six rock glaciers the authors also perform the calculation of inter-annual variations in surface speed over a period of almost 70 years. By doing so, they provide the first long-term regional investigation of rock glacier motion in the Tien Shan. The methodology is thoroughly described and the uncertainties properly addressed, and the authors discuss the limitations of the manuscript, overall with high scientific rigor. The conclusions are reasonable and co- herent with previous research.*

*However, the conclusions are supported by little and at times controversial evidence and should be discussed with more caution.*

*A strong link to Sorg et al. (2015) is evident and well explained in the introduction, but the discus- sion is only briefly addressing the relation between the two manuscripts. In general, a more detailed discussion would help placing the manuscript in the context of current research*

*and highlight its novelty (extension of rock glacier kinematic observations in the Tien Shan massive).*

*Several minor revisions and a few possible additions to the manuscript are suggested in the specific comments below. Concluding, I consider the manuscript well suited for publication in The Cryosphere after minor revisions.*

Thanks for these very detailed and constructive overall comments. We will clarify the relation of this manuscript to Sorg et al. 2015 better. Main focus of Sorg et al. was on dendrochronology, not photogrammetry and no InSAR was used. The present study uses improved processing; more time stamps including more recent data; more rock glaciers; and combination with an InSAR kinematic inventory.

*Specific comments:*

We agree with all specific comments below and will implement them, unless discussed

*Page 1, Line 23: Please delete "very" in "very high resolution data".*

*Page 1, Line 23: Most of them reads three out of six. It would be very interesting to include an analysis of the regional scale evolution of surface speed, which would largely improve the confidence in the general conclusions. This might be possible on the basis of the readily available InSar or optical data. The resolution of such an analysis might be lower compared to the one provided for the six study cases investigated in detail. If this analysis is not done, it should be made clear in the text that the conclusions are still rather speculative and based on preliminary results and similitudes to other regions (Alps).*

Agreed, we will make our conclusions more specific and balanced throughout the paper, or justify it better, respectively. We don't see a possibility to extent the time series to more rock glaciers, though. InSAR data extend not much back in time, and processing and combining many displacement rates from different sensors is a massive work. Processing more optical time series requires more modern data, which are very expensive, and more Soviet era air photos. The latter are very tricky to obtain, and we have no coverage over other rock glaciers available. Also, processing of these airphotos was laborious as we didn't have access to the original raw data. We will clarify that better in the revision.

*Page 1, Line 26: The comparison between rock glacier and glacier sediment transfer is poorly constrained in the manuscript. Please provide more information and a context. Please add some relevant references from the literature, and extend the discussion. This might require some additional analysis and a quantitative regional-scale compari- son of the two contributions.*

We will explain better and modify our conclusion about rock glacier sediment transfer, include more relevant literature, and discuss the preliminary nature of our proposal better.

*Page 1, Line 27: The relation between the stress regime (compressive flow regime) and changes in speed upstream is mentioned, but only superficially discussed, without any reference to rock glacier dynamics and their mass fluxes.*

Good point, we will incorporate the stress regime of the rock glaciers investigated better in the discussion and interpretation of flows, fluxes and the changes found.

*Page 1, Line 28: The conclusion that the Gorodetsky Rock Glacier does not show an acceleration due to the decoupling from its rooting zone contradicts the previous point. It also partly contradicts the hypothesis that rock glacier acceleration is due to warming air and ground temperatures. The two mechanisms are not exclusive, but their interaction is not straight forward and requires a more detailed discussion in the manuscript (and possibly additional analysis).*

We will clarify that the decoupling is one possible explanation, and discuss the potential effects on RG speed, but also its uncertainty and other effects that may play, and that one case is not really a strong evidence for a process.

*Page 1, Entire Abstract: I suggest to rewrite the abstract in a more concise and specific fashion, in order to highlight the novelty and the merit of the manuscript.*

Will be done.

*Page 2, Line 2: Rock glacier move due to the deformation of debris-ice mixtures (as you also mention several times later in the text). Please correct this first sentence, which now suggests that they move due to the deformation of the frozen debris only.*

Agreed

*Page 2, Line 3: Please correct the repetition: "rock glaciers, or (rock glaciers)." I can imagine that it is a typo for the one-word version "rockglaciers".*

Agreed

*Page 2, Line 10: Due to the ongoing debate about the genesis of rock glaciers and the origin of their constitutive material, I find the text insufficient for an introduction. I would avoid confusion and keep this topic out of the intro, also because it is not needed nor addressed further in the manuscript. If the authors believe that this is essential for their manuscript, I suggest that they extend the introduction and address the topic in more detail.*

Agreed, we will remove.

*Page 2, Line 17: The concept of destabilization might be easily misunderstood and confused for structural instabilities. This being not the case, some additional explana- tion might be needed to present the concept of rock glacier destabilization. However, here again, I suggest to omit this point, because it is not needed in the manuscript. Alternatively, introduce it in more detail.*

Agreed, we will remove. Not relevant for the rock glaciers investigated.

*Page 2, Line 21: please add citations to Arenson and Springman (2005)a-b.*

Agreed

*Page 2, Line 25: please be careful when considering the different processes. The re- sponse of permafrost creep to increasing temperature is thought to follow a power law. In addition, the role of water can enhance, also in a non-linear fashion, the response of creep to temperatures approaching melting conditions.*

We will specify

*Page 2, Line 26: please consider the following references, which are in my opinion amongst the most important publications regarding this topic: Ikeda et al, (2008), Buchli et al, (2018) and Cicoira et al, (2019)b.*

Agreed

*Page 2, Line 29: please specify "impacts".*

We will specify. We thought e.g. about slope instabilities, infrastructure stability or mass transport.

*Page 2, Line 30: inventorying rock glaciers is typically done on the basis of a combi- nation of optical and topographical data. Kinematic information (e.g. InSar) remains an optional and non-sufficient data source. Please be more precise in the text. The authors might refer to the Baseline Concepts for inventorying rock glaciers which are currently being elaborated within the International Permafrost Association IPA.*

We will specify. But we think InSAR is an important method for that purpose, and one that will become even more important. The 2nd co-author is key member of the IPA/ESA process mentioned. We will clarify that neither optical data and InSAR is sufficient alone, and that rather the combination is optimal, and thus used in our study.

*Page 3, Line 19: please be more specific in the wording. "followed" is not a technical term and might be misunderstood. I suggest to use the key-terms: qualitative – similar patterns, statistical correlation, phase lag, thermal offset, non-linear.*

Agreed

*Page 3, Line 20: The influence of temperature forcing through heat conduction on rock glacier dynamics has been quantitatively investigated in detail in e.g. Kääb et al, (2007) and Cicoira et al, (2019)a-b. The authors might want to discriminate between qualitative and quantitative studies that have investigated the processes controlling variations in rock glacier creep and include the state-of-the-art knowledge on the topic.*

We will specify

*Page 3, Line 21: The influence of variations in ground temperature through melt water advection has been shown to be negligible for the case of the Furggwanghorn in Buchli et al, (2018) and for the Ritigraben Rock Glacier in Cicoira et al, (2019). I am not aware of any study where the hypothesis (in the submitted manuscript) was tested on the basis of observational data nor modelling studies. If such study exist, please include the reference in the text in an explicit fashion. The study of Ikeda (cited in the text), also concurs to the hypothesis that rock glacier creep is controlled by variations in the effective stresses, rather than variations in ground temperatures (being these close to the melting point).*

Good point, we will clarify/correct.

*Page 3, line 21: as a general comment, I see the need of general revisions of the text about the processes controlling rock glacier creep.*

Agreed, and we will try to combine with shortening.

*Page 3, Line 24: At this point, the reader would expect temperature, precipitation and snow cover data to be analysed along the creep rates. I believe that the study presents enough new insights and does not need this additional step, but I suggest the authors to explain why it has not been done.*

We will do, and check if we can get hold of more data (not trivial in the region) or if climate reanalysis data could be useful.

*Page 4, Line 15: please specify the depth of the ground temperature measurements.*

Will do

*Page 5, Line 7: please consider replacing "rock" with "boulder" in all the appropriate cases.*

Agreed

*Page 5, Line 8: add a point at the end of the sentence.*

*Page 5, Line 24: Please explain the reasons for the choice of the six rock glaciers.*

We will clarify (previously investigated and suitable data available).

*Page 6, Line 1: I am not sure what "frozen snow" means. Please consider replacing this formulation with a more specific terminology.*

Will do. We mean non-melting snow and will use the term "dry snow" as common in radar remote sensing.

*Page 6, Line 12: The choice of assigning a polygon to the next class when the velocities are close to its upper limit seems unjustified to me. Also, what is the reasoning behind the division for the first classes (0-2, 2-10)? The next two classes are (half) an order of magnitude, so I wonder why also the first two are not consistent.*

The reason for the classification in the higher class lies in the under-estimation of the surface flow velocity due to the 1d-LOS sensitivity of the InSAR data. We thus accounted for the correction factor to be applied. This will be clarified in the text.

The choice of the division between the classes 0-2 and 2-10 originates from the criteria for the determination of intensity in the "Guideline for the integrated hazard management of landslides, rockfall and hillslope debris flows" of the Swiss Federal Office of Environment. The IPA Action Group "Rock glacier inventories and kinematics" in the definition of its practical guidelines for rock glacier inventory using InSAR (kinematic approach), see https://www3.unifr.ch/geo/geomorphology/en/research/ipa-action-group-rock-glacier, also noticed that inconsistency and is now proposing a different division (< 1 cm/yr, 1-3 cm/yr, 3-10 cm/yr, ...). A revision of the rock glacier inventory in Ile Alatau and Kungöy Ala-Too is ongoing following these guidelines. The submitted work is based on the previous division and we will include a statement about the ongoing revision in the outlook. The classification used has no impact on the conclusions from our work.

*Page 6, Line 13: please specify the nature of the mentioned variations. (spatial varia- tions?)*

Will do. We mean temporal variations.

*Page 6, Line 15: is there a reason why the vector of the observed displacements has not been corrected according to topography? Maybe extend on this point.*

The reason that it was not applied (explicitly) in this case was the elaboration of a classified InSAR derived inventory based on expert decision. An a-posteriori calculation of the specific correction factors was not possible, since no date-interval and precise LOS velocity was recorded during the elaboration. One lesson learned was, that the additional effort in recording the actual scene that determined the velocity (sensor, date1, date2, and LOS velocity) was recorded for other studies. However, the vector was actually considered in the selection of the faster velocity class, when close to the class boundary (see your comment above to *Page 6, Line 12:).*

*Page 6, Line 23: is this sentence a list of the criteria used for identifying and locate the rock glaciers? Currently, the sentence is somehow lost in the text. Please consider rephrasing it.*

Agreed. Will rephrase:
"E.g. rock glaciers usually show an increase of the velocity to the inner region of the mapped polygon, have a lobe-structure and clear fronts."
To "E.g. rock glaciers usually show an increase of the velocity towards the front and inner region of the mapped polygon."

*Page 6, Line 24: short-term variations in rock glacier velocity have been observed at many sites worldwide. Seasonal and even weekly oscillations are observed con- sistently. The statement is therefore unjustified. I suggest to support it with specific evidence for the study area (if available) or to discuss it in more detail. See Haeberli, (1985), Wirz et al, (2016), Strozzi et al, (2020).*

We will clarify this criterion

*Page 6, Line 25: it is now not clear to me weather the analysis has been conducted over multiple time steps. Maybe this has not been explained clearly enough in the text,  or I just misunderstood it here.*

Agreed. Temporal variations are included only implicitly since the InSAR data availability did not allow a thorough analysis of the temporal behaviour. Eg. when single early summer scenes show decorrelation due to faster movements and late autumn scenes show coherent and slower motion in the same extent, this can be qualified as seasonal variations.

*Page 6, Line 19-30: is this paragraph a list of the criteria used to classify a rock glacier  and distinguish it from a rock glacier? Please be more specific and explain in detail the  concepts that have been used. I suggest to reference to the ongoing action group on  rock glacier inventories and kinematics of the IPA.*

Agreed, will clarify. In this paragraph, some typical motion patterns determined from InSAR were pinned to specific landforms / surface motion processes. This is to extend the morphological approach based on the available imagery.

*Page 8, Line 11: please explain why the measure of accuracy was performed on stable  ground only for three of the six field sites.*

We will explain. The other rock glaciers are surrounded by steep slopes where orthorectification DEM errors, shadows and other topographic effects make such test difficult systematically (and automatically) for a large number of points. The three rock glaciers chosen are surrounded by terrain with similar properties than the rock glaciers themselves and we view the test there to be representative for all the rock glacier surfaces.

*Page 8, Line 18-24: The results of this very interesting analysis are only briefly de- scribed in the manuscript. Also the discussion seem to me not sufficient to address this point. I suggest to include more details in the manuscript.*

Given the upper-limit length of the current manuscript we preferred to mention this aspect only shortly. Thoroughly covering it would require even more analyses, figures/tables, and descriptions and discussions. We will try to give more details without adding much length.

*Page 10, Line 10: please specify that only (part of ) the labelled rock glaciers are further investigated in the photogrammetric analysis, and not all the visible polygons in  the figure.*

Will do.

*Page 10, Line 10: please consider indicating the value of the wavelength for the data  in the figure.*

Will do, assuming the referee means the Sentinel-1 C-band radar.

*Page 10, Line 10: it is impossible from the figure to distinguish between the different  classes of the polygons. This information is present only in Fig. 1 at a very low reso- lution. Consider improving the level of detail in this (Fig. 2) or in the following figures (Fig. 3-7).*

We will try to add more details/labels to Fig.2, make sure Fig. 1 has high resolution so that readers can zoom in, and will mention that the inventory data are available online.

*Page 11, Line 3: consider replacing "photogrammetric velocities" with a more detailed terminology. (such as "Surface velocities calculated by offset tracking").*

Will do.

*Page 11, Line 4-12: the determination of the origin of the ice and sediment constituting the rock glacier requires more than the observation of spatial connection, or as in this case, the (legit) supposition of past spatial connection. The state of inactivity of the current glacier forefield (called in the text "zone between glacier and main rock glacier") only shows that the connection is not currently present, but is not sufficient to imply that this (dynamical- and sedimentological) connection was present in the past. Even in this case, it would have been limited to the period when the glacier advanced to its maximum (LIA). No information on the climatic and sedimentological setting of the rock glacier is provided in order to commence such an analysis. Without entering in too much detail, I suggest to limit the discussion to the spatial connection (glacier forefield connected, according to Delaloye and … 2018) and avoid speculation regarding the "nourishment" and genesis of the landform.*

Agreed, we will modify accordingly and in line with above comment to leave touching the origin of rock glaciers more than necessary.

*Page 11, Line 15: please provide quantitative evaluation of the observed trend and its statistical significance.*

Giving useful and reliable numbers about statistical significance will be hard to obtain due to the small and uneven number of measurement years possible. We will give a more specific description of trend significance, though.

*Page 11, Line 18: as above, consider removing the concept of "nourishment" from the paragraph.*

Agreed

*Page 11, Line 19: consider removing "striking" or replace it with a more technical and specific adjective.*

Will do.

*Page 11, Line 25: please describe in more detail the differences between the surface speed for this early period.*

Will do.

*Page 11, Line 27: I suggest to improve this paragraph and give a better summary of the results for this analysis. Also, write explicitly what the calculated ice-content would be. It is implicit in the ratio, but the reader might be helped by some repetition here. Consider calculating it for all the available time steps and include an estimation of the uncertainty in the results.*

We will give more details. The ratio between advance rate and surface speed is a function of ice content and vertical profile of horizontal velocities. We cannot quantitatively separate between both. The images available are not of sufficient quality to measure advance rates for every time step with useful accuracy.

*Page 14, Line 11: In figure 4c, the debris-covered glacier and (for what I can see) the glacier forefield are not shown and it is impossible to verify the presence of a material flux from the*

*rock wall to the glacier. As previously, I consider the observational evidence insufficient to support the statement. I suggest to simply avoid the point, which is in my opinion not important for the present manuscript, or to include more data and analysis to support this thesis.*

We will avoid the point. Fig 4c shows the entire system, as does Fig 5a. Fig 5c shows only parts of the uppermost part.

*Page 14, Line 24: the deformation profile (on the vertical dimension) also has an im- portant influence on this calculation. Why is it not mentioned here? Please consider spending some words about this point to make the text clearer.*

Yes, agreed, see above comment. But if the ratio is close to 1 then there has to be little decrease of speed with depth and little ice content. We will explain better.

*Page 15, Figure 5: it would be interesting to see the displacement on stable terrain.*

Stable terrain displacements are shown in Fig. 5c, but not included in 5b for the reasons given above. But we will add numbers in the text. 5c reflects that measurements around the rock glacier are much more difficult than on it and thus of limited use.

*Page 16, Line 2: what is the mass flux at the boundary between the glacier and the rock glacier? What is the absolute value of the surface speed? It is very hard to see this from the figure provided. In general, a similar comment as for the nourishment above.*

We will avoid the nourishment concept.

*Page 18, Line 15: here the authors state that the sediment transfer between glacier and rock glacier cannot be determined with the available data. I agree with the conclusion, but still, I would like a more quantitative discussion. Otherwise, I suggest again to completely avoid this point. A possible analysis would investigate the relation between acceleration and max fluxes along flow lines.*

Agreed to avoid the point.

*Page 18, Line 15: more information about the vegetation could be interesting. What is the size and what are the species growing on the rock glacier? Is this information available from field expeditions?*

We will refer to Sorg et al. 2015

*Page 18, Line 21: the fact that the observed signal from feature tracking is lower than the noise does not allow to conclude that the rock glacier has accelerated. I don't understand why the authors mention "statistical significance" in their argument.*

We negated this statement, "cannot … be considered … significant". We will reformulate.

*Page 22, Line 1: I agree with the statements, but I would like a more quantitative evaluation of the different error types.*

We will add more numbers to the different error types.

*Page 23, Line 2: I agree with the authors. Still, it would be interesting to see the values of the errors on stable ground.*

We will add numbers. See also several above comments.

*Page 23, Line 7: the divergence of a vector field is a well defined term and as far as I understand this is not what the authors mean. I suggest to replace this substantive with a more pertinent description of the differences observed in the velocity field.*

We will replace the term.

*Page 23, Line 10: I have not found any values for the statistical analysis of the velocity time series. I warmly suggest to implement this analysis in a quantitative way. If the authors prefer not to, I would be much more careful talking about statistical significance.*

The number of measurement years available is too low to do meaningful trend analyses, we suggest. We will instead modify statements about statistical significance, and the term,

*Page 23, Line 25: it would be valuable to have a better quantification of the "strong compressional regime" by means of e.g. strain rates (accompanied by proper interpre- tation or even with the calculation of the internal stresses).*

We will compute strain rates, and give and interpret the numbers.

*Page 23, Line 27-29: First, it would be very welcome to see the original results for the upper part. Second, I don't agree with the statement that the lower part responds passively. It is a matter of dynamics, thus of mass and momentum fluxes. I suggest to change the formulation, possibly mentioning that the dampened response (I would appreciate an illustration of this) is due to topographical setting and the correspond- ing dynamic behaviour of the rock glacier. (In detail it could be that the mass flux is mostly compensated by variations in thickness or in mass input rather than variations in velocities, but such a statement should be supported by more evidence.)*

Agreed, we will modify the formulations. We will also add results of the upper part.

*Page 23, Line 29: please repeat the advance rate and the surface velocity, and con- sider discussing the result in more detail – also with a possible range of quantitative values of the ice content.*

We will repeat and give more quantification.

*Page 23, Line 32: given the strong similarity between the two publications, this point might require some more discussion. I would suggest also one or two figures in the appendix or some additional comparisons.*

We will describe the differences between the publications better (see above comment).

*Page 24, Line 10: If I understand this correctly, it means that it is not possible to conclude which one of the two studies is more accurate. If this is the case, please state it more clearly in the text.*

We will explain better. The new study is more accurate. The statement about other measurements to compare with is not related to the difference between this study and Sorg et al. We will clarify this.

*Page 24, Line 24: consider citing Cicoira et al., 2019b, where it has been shown that the seasonal and inter-annual variations in rock glacier flow are mostly controlled by variations in snow melt rates and liquid precipitation, rather than in air temperature. Other very relevant citations are Buchli et al., 2018 and Ikeda et al., 2008.*

We will do.

*Page 25: in general, in the "5.3 Speed time series" paragraph, more discussion relative to the results and their validity would enhance the validity of the manuscript. Most of the discussion is a very precise comparison to Gorbunov et al., (1992), which could be probably summarized in one or maximum two paragraphs. I suggest to highlight more the originality of the manuscript and discuss better its strength and weaknesses.*

We will modify accordingly.

*Page 25, Line 26: I don't see the link the negative glacier mass balance. Please avoid this point or argument in more detail the linking mechanism.*

We will clarify the climatic significance of the glacier mass balances.

*Page 25, Line 29: it would be very interesting to quantify this sediment transfer. I suggest using a simple assumption for the rock glacier thickness (e.g. constant value of 20 meters, see Cicoira et al., 2020) and estimate the overall ice/sediment transport rates for the periglacial environment. This would be a major result, and is not very difficult to calculate (although the uncertainty will be large).*

We will add such an initial estimate.

*Page 25, Line 31: such a statement definitely requires a quantification of both the sediment transfer.*

We will specify better what we mean.

*Page 26, Line 4: this point is very interesting but insufficiently discussed. I suggest the authors to implement it both in a qualitative and in a quantitative fashion.*

We will try to specify more without adding much length and without repeating much of the dedicated study referred to.

*Page 26, Line 5: This statement appears unjustified to me. As far as I know, no quantitative (and conclusive) evidence that a rock glacier derived from a glacier exist. As for this manuscript, there is not sufficient evidence supporting the statement. Often, an interaction (more or less important) has happened during the LIA. I suggest to rewrite this last paragraph with more focus on the novelty of the manuscript (it is not the geomorphological genesis of the rock glaciers).*

Agreed, we will modify accordingly and leave out the nourishment concept.

*Page 26, Line 18: it is not so clear to me which assumptions. Please be more explicit in the conclusions.*

Agreed. Will clarify:
"However, the outlines and lower limit of the velocity of the surface can be identified using the assumptions stated above." to "However, the outlines can be drawn using the decorrelation pattern and velocity can be assumed as minimum velocity of e.g. 1 m/a for a 12-day Sentinel-1 interferogram.

*Page 26, Line 26: the time scale considered in the manuscript is (almost) only decen- nial. I would rephrase this sentence and highlight the fact that the original observations in speed also show past periods of acceleration at the investigated temporal scale.*

We will modify accordingly.

*Page 27, Line 4: I am not completely convinced by this statement. The quantification of the trends and their significance for each rock glacier might make this point more convincing and increase the confidence in the results and the conclusions.*

We will modify along the above comments.

*Page 27, Line 10: I don't agree at all with this statement. This is very speculative and not supported sufficiently by the evidence provided in the study. As above, I suggest to discuss it in more detail or to avoid this point, which is in my opinion not relevant for the manuscript.*

Agreed, we will avoid the point.

*Page 27, Line 10-15: on the contrary I welcome the topic as an outlook for future studies. With this last comment, I thank the authors for an interesting piece of research.*

We will modify accordingly.

*References:*
*Arenson, L.U., Springman, S.M., 2005a. Mathematical descriptions for the behaviour of ice-rich frozen soils at temperatures close to 0âŮ̧C. Can. Geotech. J.42, 431–442. https://doi .org /10 .1139 /t04 -109.*
*Arenson, L.U., Springman, S.M., 2005b. Triaxial constant stress and constant strain rate tests on ice-rich permafrost samples. Can. Geotech. J.42, 412–430. https://doi .org /10 .1139 /t04 -111.*
*Cicoira, A., Beutel, J., Faillettaz, Vieli, A., 2019b. Water controls the seasonal rhythm of rock glacier flow. EPSL 528, https://doi.org/10.1016/j.epsl.2019.115844*
*Cicoira, A., Marcer, M., Faillettaz, J., Gärtner-Roer, I., Bodin, X., Arenson, L. U., Vieli, A., 2020. A general theory of rock glacier creep based on in-situ and remote sensing observations. PPP (in review).*
*Delaloye et al., 2018. Rock glacier inventories and kinematics: a new IPA Action Group. EUCOP5.*
*Buchli, T., Kos, A., Limpach, P., Merz, K., Zhou, X., Springman, S.M., 2018. Kine-matic investigations on the Furggwanghorn Rock Glacier, Switzerland. Permafr. Periglac. Process.29, 3–20.*
*Haeberli, W., 1985. Creep of Mountain Permafrost. Mitteilungen der Versuchsanstalt für Wasserbau, Hydrologie und Glaziologie der ETH Zürich, vol.77.*
*Ikeda, A., Matsuoka, N., Kääb, A., 2008. Fast deformation of perennially frozen debris in a warm rock glacier in the Swiss Alps: an effect of liquid water. J. Geophys. Res., Earth Surf.113. https://doi .org /10 .1029 /2007JF000859 .f01021.*
*Wirz, V., Gruber, S., Purves, R.S., Beutel, J., Gärtner-Roer, I., Gubler, S., Vieli, A., 2016. Short-term velocity variations at three rock glaciers and their relationship with meteorological conditions. Earth Surf. Dyn.4, 103–123. https://doi .org /10 .5194 /esurf -4 -103 -2016.*

| Referee #2     Philippe  Schoeneich |
| --- |

*It is a very high quality paper, written by some of the best specialists of the topic and of the methods used. The paper provides a valuable and comprehensive dataset, on an area for which few data were available so far.*

*The work presented uses a combination of various methods and various types of optical and satellite imagery. A combination that was rarely achieved at such a level in a single study. This allows a  cross-checking and validation of the results, which therefore appear as very robust. The use of  image archives allows a back-analysis from the 1950s onwards and the interpretation of the  evolution of velocities over time. If this is not new, the above mentionned combination of various  image types and methods allows a higher and more detailed temporal resolution, and reveals short  lived velocity changes that could not be observed by using a single type of images.*

*The paper provides therefore an innovative contribution both for the results and the methodological approach.*

*Therefore, there are no fundamental comments, and the paper should be accepted with the minor improvements listed below or proposed by others.*

*p. 4 – l. 6 « landslides and rock avalanches » : there is a problem of vocabulary. In English, the  word « landslide » is used as generic term for all types of mass movements on slopes, and thus  includes rock avalanches. The authors probably wanted to distinguish slides (= « Rutschungen », « glissements ») from rock avalanches. The adequate term in this case is « slide ». If they mean  deep seated slope movements (= « Sackungen », « Talzuschub », « tassements », « glissement  rocheux ») the most adequate concept would be DSGSD (for Deep Seated Gravitational Slope  Deformation). As generic term for designating all mass movements, we recommend to use « mass  movements » instead of « landslide », the latter term causing much confusion among french or  german speaking readers.*

Thanks, we will change

*Fig. 1 + part 4.1 : the figure legend indicates 1 « landslide » (see also comment above), but this is  not mentionned in the text.*

Agreed. The Landslide was an unresolved relict in the Figure output. We will adapt the figure.

*Fig. 1 is small and hardly lisible. It should be provided in full format as downloadable  supplementary data*

Agreed. We will make sure the figure is available in high-res in the paper. The inventory is also available from an open archive. See also comment from Ref #1.

*Figures 3, 4, 5, 6 and 7 : the velocity color scale is not the same in all figures, which hinders a direct comparison of the different cases and can lead to misinterpretation. In figure 4 for instance, at a first and quick view, the Morenny RG could appear as slower than the Archaly RG, which is not the   case. A common velocity color scale would be better.*

We believe the velocities are too different among the rock glaciers to make a common scale useful. With a max of 6.5 m/a many details wouldn't be recognizable anymore for the slower rock glaciers. We will perform some colour experiments and try to include a figure in the supplement with comparable color scales.

*p. 24, l. 20-25 : you mention the influence of snow thickness/insulation and amount of meltwater as factors possibly explaining the acceleration in the 1960s. I agree with this general statement, but it  can*

*be refined. The data series of the Laurichard rock glacier (see Bodin et al. 2009, you already have in your references) and ground surface temerature measurements at many places, show that the most relevant factor is the onset date of the snow cover. An early onset, for instance in October, before the coldest days of November-December, prevents the ground from cooling and keeps the accumulated summer heat in the ground. On the opposite, a late onset of the snow cover, by end of December of even January, will allow a strong cooling of the ground surface and a deep seasonal frost in non permafrost areas. The Laurichard time series shows that the reaction is assymetric : it needs several « warm » years (years allowing a warming of the ground, with hot summer and/or early snow cover) to induce an acceleration, but a single « cold » winter (with late onset of the snow cover) is sufficient to reduce velocities. Do your meteorological data allow to establish the onset date of the snow cover (insulation is provided with a minimal thickness in the order of 60-80 cm) ?       If yes, it could reinforce your interpretation.*

We will try to get more detailed met-data. If we don't succeed, we will modify the text along the above considerations, with which we agree.

*Author contributions : there are two AK among the authors ! Who is AK ? And what was the contribution of the second one ?*

Good point, thanks for spotting! We will specify.

*Acknowledgements : the two first lines are duplicated in the Financial support section. Remove from here, and leave under Financial support.*

We will revise

*References*

We will fix the below issues. Thanks for checking the details!

*Check formatting of abridged journal names :*
- *for PPP : your text mentions Permafrost Periglac. (Processes is missing) – numerous references*
- *p. 35 Kääb et al. 2017 : NHESS (Sciences missing)*

*p. 33 - Gorbunov 1983 : volume missing*

*p. 33 - Haeberli 1985 : volume number in the collection missing*

*p. 34 – Kaufmann 2012 : journal missing*

*p. 36 – Roer et al. 2008 : incomplete (volume, pages, ...)*

Referee #3      Line Rouyet

*Kääb et al. inventoried active rock glaciers and other periglacial landforms in Tien Shan using InSAR and further analyzed the spatial-temporal patterns of six fast-moving glacial-derived rock glaciers using optical offset tracking. They present a nice piece of work that shows the value of radar and optical remote sensing for mapping and moni- toring creeping permafrost landforms over a large and hard-to-access areas. The study focuses on a region that had not been intensively documented before. The contribution is well suitable for a publication in The Cryosphere and likely to become a reference in remote sensing of mountain permafrost.*
*I have no major concern regarding the way the study has been designed but I think the manuscript could benefit from some clarifications about the inventory procedure, a better visualization of the results and an extended discussion on the findings. Here I develop three main elements. I also listed some complementary suggestions of cor- rection at the end of the review.*

*–Main comment 1. Inventory and movement classification–*

*For the movement types, I think it should rather be named 'landform types' as e.g. a rock glacier or a moraine is not a process but a landform that has been shaped by a periglacial or glacial process. Consider also renaming 'solifluction / debris movement' (not sure what debris movement means; a rock glacier is also composed of debris. Scree deposit?). 'Dead ice / subsidence' could just be 'thermokarst'?*

We will rename following your suggestions. Note on the "debris movement": the meaning was simple gravitational (settling) motion on scree slopes as opposed to solifluction. Both are meant to cover shallow debris motion. The legend in Fig 1 will be adapted.

*About the velocity classes in the inventory: The choice of the velocity classes needs at least to be explained. 0-2, 2-10, 10-50, 50-100, >100 does not look so intuitive to me.*
*Why not equal intervals: e.g. 0-25, 25-30, 50-75, 75-100. Or 0-3, 3-10, 10-30, 30-100, >100? Or just an order-of-magnitude (cm/yr, cm-dm/yr, dm/yr, dm/yr-m/yr, m/yr), as now recommended by the IPA action group 'rock glacier inventories and kinematics'? The criteria for defining the class are also not fully clear: when writing that 'if two or more classes were present during the observation time-span, the higher displacement rate was used to determine the velocity class' (l.14-15, p.6), does it mean that a little fast-moving part of the rock glacier could lead to classify the whole landform as for ex. >1m? If so, that sounds a bit bold to me.*

Agreed, that would indeed lead to a false representation of the actual kinematics. Usually, when a classification was done, a "representative" part of the polygon must show similar speeds. Usually small spots within the polygon with significantly higher velocity (such as accelerating or failing fronts) were either ignored or outlined separately. Therefore, from this direction, no over-estimation is expected.

*Maybe it should just be acknowledged in the discussion that it could lead to a rate overestimation. Similarly, at l.3-4 (p.7), it is acknowledged that the max. measurable rate for a C-band with 12d repeat-pass is 85 cm/a. To my understanding, a decorrelated part in a 12d-interferogram means >85 cm/a, but so not necessarily >100, right?*

From the calculation point of view, 85 cm/a is seen as 2.8 cm in a 12 day S1 interferogram. But in reality, given the size of the area under motion, this could be recorded coherently, when fringe visibility is good. This value even can be exceeded.
Therefore reason 1) for classification in the > 100 cm class is, that motion decorrelation usually is under higher displacements as the minimum value. And 2), the topographic correction factor is in realistic cases not lower than 1.1-1.2, making a LOS displacement rate of 85 cm/a rather belonging to a landform of > 100 cm/a class.
However, in the work carried out, we did not further group the decorrelated (and probably exceeding 100 cm/a) from the other "undefined" class.

*In these cases, were the 1d ERS interfero- grams used to ensure a correct classification? If yes, it could just be mentioned.*

The few good 1-day ERS interferograms we had, did not show good signatures. Probably because due to the main temporal coverage from March-Early June, with snow melt and other snow-related changes. So, from these data no additional information could be drawn unfortunately.

*–Main comment 2. Figures–*

*Most of the figures could benefit from slightly more job to clearly disseminate the find- ings. Considering the volume of work that the study represents, it is a bit a pity if the presentation does not fully help to maximize the understanding of the results.*

*Figure 1: This map is hard to read. I understand the wish to be comprehensive, but it is most likely too much information in once and the color contrast does not allow for differentiating the velocity classes for specific landforms. I would suggest making two maps: one with only the different movement types, without any velocity classes. And another focusing on rock glaciers (the main point of the article) with a different color for each velocity class (for ex. green to red, blue to red). A similar map for the other landform types can potentially be placed in Supplementary.*

We plan to revise the figure as follows: Two-panel figure with 1) two color schemes (e.g. black / gray) for rock glacier / non rock glacier types. And 2) a color map for only the rock glacier classes in overview. And a zoom for the "time-series" rock-glaciers in a 2nd figure.

*Figure 2: Add the equivalent in cm of the phase color scale. Missing information about SAR geometry (LOS arrow or mention of the ascending/descending geometry in leg- end).*

Will be done

*Figures 3-7: It would be more comfortable for the reader if the optical images always had the same extent and scale than the velocity field maps. The error bars (especially the red) are hard to see due to the transparency. The arrows of the photogrammetry results are sometimes impossible to see and interpret (Fig. 5-7). Less importantly: it could be nice to have altitude references at some locations on the optical images.*

We prefer to have the optical images giving the full overview and the velocity figures to zoom in. But we will indicate the velocity zooms in the overview images. We will make sure the figures are in high-res to give the reader the possibility to zoom in. We will improve the error bars. We will add elevation points, good idea!

*Figure 8: Add the equivalent in cm of the phase color scale. Missing information about SAR geometry. Here, as the background map from the interferogram is not directly related to the size of the arrows (contrary to Fig. 3-7), it would be good to add a legend of the arrow length. It is a really nice figure to show the agreement of both methods, as well as the value of combining them. What about adding the same for the five other rock glaciers (potentially in Supplementary)?*

We will modify the figure along your suggestions. We will also consider to add similar figures for the other rock glaciers in the Supplement.

*–Main comment 3. Time series and acceleration–*

*There is something that does not completely add up in the interpretation of the temporal variations: contradictory information that gets the reader a bit lost at the end of the paper. The title includes 'acceleration of rock glaciers' and it is presented in the abstract that five of the six temporally investigated rock glaciers exhibit acceleration (l.23-24, p.1). However, at several locations, the authors acknowledge that the statis- tical significance is not always good enough to confirm the trends (several cases with std dev of speed differences greater than the actual measured difference, e.g. Fig. 7c). As explained at l.28-29 (p.24), 'Gorodetsky, Morenny and Archaly rock glaciers […] are not showing a clear increase of speed over the study period'. Gorodetsky, Moreeny and Archaly are three out of six rock glaciers, which does not fit anymore with the findings' summary in the abstract (l.28, p.1) or the conclusion (l.4, p.27). Maybe I misunderstood something, but in that case, it is probably possible to be clearer in the text.*

Yes, agreed, Ref #1 rised similar concerns. We will revise.

*The statement 'the behaviour observed in the study area confirms findings […] that rock glacier creep speed increase overall under atmospheric warming' (l.23-24, p.25) sounds a bit too bold to me considering that no comparison with temperature is pro- vided in the study. Other elements could be discussed in greater details. In 5.4, the authors remind that 'all rock glaciers studies in detail in this study are derived from contemporary or former glaciers' (l.5, p.26). Some fuzzy thoughts here: When did the glacial direct connection expected to have stopped? During the time span of the study (50th-70th)? If yes, are some of the variations related to the glacial dynamics instead of the permafrost creep, or due to the transition between the two flow types? What if you had also analyzed talus-derived and slower landforms? Not to say that it should be done here, but it could be discussed as a prospect. Some of the conclusions here are maybe only valid for glacial and extreme (>1m) cases.*

Yes, also this concern overlaps with Ref #1 and we will revise, mostly by avoiding this discussion, but mention in an outlook.

*I don't think this comment requires critical changes. The results are well described in Section 4, but the discussion could be improved (especially 5.4) and the ab- stract/conclusion (and even the title?) better matched the actual findings.*

We agree and will revise accordingly.

*–Complementary  comments–*

All agreed and to be revised/clarified.

*Page 1:*
-        *Title: 'acceleration'. Maybe consider 'Inventory and spatial-temporal trends of…' See main comment 3.*

-        *l.23 and l.28: 'most of them'… 'The only rock glacier'… Maybe just write the actual number in both cases: five of them – the only, if this is really the case. See also main comment 3*

-        *l.30: maybe missing a general sentence in the abstract about what does it tell, what is the relevance of the study in a broad sense?*
—

*Page 2:*

- *l.3: rock glaciers (or rock-glaciers) I guess*

- *l.9: 'In a similar way, the ice content […] can be directly incorporated…'*

- *l.19: As rock glacier distribution is also a valuable environmental indicator, I would suggest removing the four first words of the sentence.*

- *l.21: '…although it is also influenced by…'*

- *l.22: '…temporal and spatial variations of ice content…' I guess lateral variations are as important as vertical?*

- *l.25: '…variable around thawing conditions'. Maybe 'sensitive' instead?*

- *l.25: '…i.e. in cases the ice content starts to degrade…'*

- *l.29: 'impacts from spatio-temporal variations'. Impacts on what in this context?*

—

*Page 3:*

- *l.3-5: Start of the sentence ('in addition to …,') does not sound necessary to me. Instead, maybe: 'The categorization of landforms into classified movement rates or orders of magnitude (e.g. cm/day, ….)'*

- *l.8: (more details below) is not really useful.*

- *l.17: 'constant speeds' instead of 'stable speeds'?*

- *l.19-20: repetition in these two sentences (creep velocity <-> temperature)*

- *l.24: 'for the first time' sounds a bit weird considering the extensive literature presented in review 2 and reference to Sorg et al. article. What about 'The aim of the present study is to systematically inventory contemporary rock glacier crepp velocities…'*

- *l.27-30: last two sentences are maybe not necessary.*

—

*Page 4:*

- *l.5-6: potential to rephrase here 'with the largest of magnitude8.0 and more […] and stronger earthquakes…' Stronger than 8 and more? …*

- *l.9-10: Sth unclear with that sentence: '…and for elevations of about 3000 m asl.' Does it mean 'For elevations of about 3000 m asl, mean annual precipitation exceeds 1000 mm on the northern slopes and is less than 800 mm on the southern ridges.'?*

- *l.18: larger than what? '…permafrost is very likely although the fronts of the large rock glaciers are located…'*

- *l.25: 'inactive ones': maybe wise to define here the activity categories. What about relict landforms? Are they considered in the study? In the activity morphologically determined or using InSAR/photogrammetry?*

No, only patterns with some sort of visible motion are included. We will add this info.

- *l.33: 'Blöthe et al. (2019) mapped…'*

—

*Page 5:*

- *l.7: 'Displacement rate of individual blocks vary greatly…'*

- *l.9: 'blocks'? Replace 'down-valley' by 'slope direction'?*

- *l.17-18: New sentence at 'Rock glacier activity/tree growth anomalies…'? Respectively? To what? Not clear.*

- *l.21: 'different statuses of the rock glaciers on their path towards…' -> 'different stages'?*

- *l.25: remove 'for the offset tracking work realized here.'*

- *l.30: 'short spatial baseline and time intervals between 1 day and three years…' According to your table A1, the max. interval is over one year.*

—

*Page 6:*

- *l.5: I guess $\frac{1}{4}$ of a phase cycle' will be abstract for most of the readers. Maybe add what it means in terms of displacement?*

- *l.9: I think that 1-2 sentences could added here to quickly explain the principle. Count the fringes on coherent parts and relate it to velocity considering the wavelength in- terval between pairs + use decorrelation information. In the paper, you show only 12d Sentinel examples, but it is important to be clear that you used a large stack with different wavelengths/intervals/periods to get different detection capability and avoid misinterpretation of short-term velocity variations. Someone who does not know the technique could misunderstand if not quickly explained here.*

This will be explained

- *l.11: Why these classes? See main comment 1.*
- *l.14: 'the highest displacement rate…' See also main comment 1.*

- *l.15-16: 'We empirically considered the line of sight in the assessment': It could be explained what it practically means. If the slopes significantly deviating from LOS and between two classes, the highest cause it is considered as underestimated?*

Will be clarified, see above.

- *l.16-18: last sentence of paragraph could be moved after first sentence of the next one (l.20).*

- *l.25: Check this sentence. I doubt you get coherent information on supraglacial ponds.*

- *l.27-28: Similar appearance to what? Check this sentence. I don't see why solifluction would give spot-like fringe patterns. Also: debris movement does sounds like right terminology. See main comment 1.*

Will be clarified.

- *l.30: Redundant. Remove 'also optical images were not of help due to'.*

- *l.32: Consider using everywhere 100 cm/yr instead of 1 m/yr, as in Fig.1*

—

*Page 7:*

- *l.2-3: >85 could also mean 90, which is in 50-100 class. See main comment 1.*

As stated above, it is unlikely that a decorrelated (in a 12 day interferogram) area will end up in the 50-100 cm/a class.

- *l.30 and 32: 'the above': not necessary.*

—

*Page 8:*

- *l.4: thresholds on correlation coefficients -> the actual values of the thresholds could be documented.*

- *l.12: the point (ii) sounds odd to me: looking at the chosen areas on the figures, they are significant velocity variations that are likely to be natural (for ex. C on Fig.6). Not possible to select areas with more similar patterns? What about the correlation*

*coefficients, the average correlation cannot be used as an accuracy estimate?*

These are important points that we will describe in more detail.

—

*Page 10:*

- *Legend Figure 2: 'Wrapped orthorectified interferogram'? Raw sounds weird, and not especially right considering the corrections described in Section 3.1.*

—

*Page 11:*

- *l.5: 'agree well with the interferograms'. Plural? To emphasize that several interfero-grams have been investigated.*

- *l.20: 'they show coherent fringes'*

- *l.24: 'the central part of Morenny rock glacier speeds are in the order of 1 m/a: It does not look like that in Fig.4, most of the rock glacier is blue and the averaged time series show values clearly under 1, even for the highest parts of the series. . .* We will specify

—

*Page 14:*

- *l.16: see main comment 1: in theory, noise could also mean 90 cm/yr, so can we really say consistent?* We will specify

- *l.24: 'This points to block movement of the entire rock glacier column. . .' Sounds odd, potential for rephrasing.*

—

*Page 16:*

- *l.6: ref. to Fig. 6e*

- *l.12: 'because of the many different frontal parts' -> 'because of heterogeneities'?*

—

*Page 18:*

- *l.19-20: I don't see the point of adding zone B if the results are not shown. Either the tseries are shown somewhere (potentially in Supplementary) or you maybe don't mention this*

*part at all? About 'changes documented for zone B [..] are within their error bars': looks like there is the similar problem for A since the 90th (Fig.7 c).*

—

*Page 19:*

*Figure 7: Remove area B) if not used? See also main comment 2.*  Agreed

—

*Page 20:*

-         *l.4: 'reasonably very' -> just reasonably?*

-         *l.6-7: 'all rock glaciers […] were classified correctly from the interferograms': does not sound right for Morenny (see comment page 11, l.24)*

-         *l.22-23: see main comment 1: so 85 cm/yr is approximate as 1m/yr?*

—

*Page 22:*

-         *l. 2: missing space 'very-highresolution'*

-         *l.4: 'more difficult to quantify precisely': Sounds weird cause I don't think biases a) have been quantify in the manuscript.*

-         *l.6: 'could translate 1:1 to a lateral offset': without being an offset-tracking expert, I must say that I don't get this.*

-         *l.24-25: if median values are safer, why not have used these for the point clusters?*

-         *l.33-34: here comes quickly what I commented for p.8, l.12: maybe it could come before in the methods?*

—

*Page 23:*

-         *l.23: 'acceleration between 1964-2017'*

-         *l.26-29: make two sentences. For ex: the lowest rock glacier part deforms under a regime… It responds rather passively to…*

—

*Page 24:*

-         *l.5: 'the latter part, under a strong compressive creep regime, reaches and diverts the river…'*

-         *l.28: 'are the only systems'. 3 out of 6… sounds weird to say 'the only'. See main comment 3.*

—

*Page 25:*

-         *l. 24: maybe 'is aligned with findings' instead of 'confirms'?*

—

*Page 26:*

-         *l.11: 'an inventory of active rock glaciers and other periglacial landforms'? or 'a slope movement inventory including rock glaciers and other periglacial landforms'?*

- *l.16: well, maybe it is me being picky, but 85 cm is not a meter. And again, what about the use of ERS 1-day? See main comment 1.*

No area wide additional information could be derived from ERS 1-day interferograms. Actually, only 1 RG showed a motion signature that could be used. See also above

- *l.28: 'A possible explanation […] is…'*

—

*Page 27:*

- *l.2: '…more passively to...'*

- *l.4: Not in line with l.28-29 p.24. See main comment 3.*

- *l.10-12: this could be more discussed in 5.4. See main comment 3.*

- *l.12-15: The last sentence is too long and hard to understand. I would suggest as prospect to include not only glacial-derived and landforms with lower averaged velocity, to confirm the accelerating trend.*

—

*Page 29:*

- *Table A1: Why no 2017 Sentinel-1 image?*

Every project has its own history of definition, work execution, and revision. Within the ESA GlobPermafrost project we completed our first inventory based on 2015-2016 Sentinel-1 images. While writing the paper we revised the inventory and included some newly available Sentinel-1 interferograms from 2018 to clarify some open issues. These images were archived and with a very good quality, so no further data were necessary for this work. However, the ongoing revision of the rock glacier inventory in Ile Alatau and Kungöy Ala-Too following the IPA guidelines (see above) will include more Sentinel-1 images also from more recent years. And important, these revisions will have no significant impact on the conclusions of the present paper.

---

## Author Response (AR1)

The Cryosphere        tc-2020-109
**Inventory, motion and acceleration of rock glaciers in Ile Alatau and Kungöy Ala-Too, northern Tien Shan, since the 1950s**
Andreas Kääb, Tazio Strozzi, Tobias Bolch, Håkon Trefall, Markus Stoffel, Alexander Kokarev

**Response to referees**

**General response**

We would like to thank the three referees for their positive, very constructive and detailed reviews that will certainly help to improve the paper!

We agree with most of the comments made (as detailed below), and modified our manuscript accordingly. In summary: we clarified/modified descriptions and give at a number of places more details; we balanced the conclusions better with respect to the measurement results, in particular considering their statistical significance; and we avoided discussions around glacier feeding of rock glaciers.

Referee comments are in *italic*, and our response and what revisions we did in normal font.

**Response to individual referees**

Referee 1: page 1
Referee 2: page 12
Referee 3: page 14

Referee #1     Alessandro Cicoira

*General comments:*
*This manuscript investigates the distribution and the motion of a large sample of active rock glaciers (551 landforms) in norther Tien Shan. The rock glacier inventory is not exhaustive and does not include inactive and relict landforms. In addition, the movement of some other landforms, such as debris-covered glaciers and ice-cored moraines, have been classified (900 landforms in total). A combination of satellite based radar interferometry and feature tracking of optical imagery allows the classifi- cation of activity classes for all the landforms mapped. For six rock glaciers the authors also perform the calculation of inter-annual variations in surface speed over a period of almost 70 years. By doing so, they provide the first long-term regional investigation of rock glacier motion in the Tien Shan. The methodology is thoroughly described and the uncertainties properly addressed, and the authors discuss the limitations of the manuscript, overall with high scientific rigor. The conclusions are reasonable and co- herent with previous research.*

*However, the conclusions are supported by little and at times controversial evidence and should be discussed with more caution.*

*A strong link to Sorg et al. (2015) is evident and well explained in the introduction, but the discus- sion is only briefly addressing the relation between the two manuscripts. In general, a more detailed discussion would help placing the manuscript in the context of current research and highlight its novelty (extension of rock glacier kinematic observations in the Tien Shan massive).*

*Several minor revisions and a few possible additions to the manuscript are suggested in the specific comments below. Concluding, I consider the manuscript well suited for publication in The Cryosphere after minor revisions.*

Thanks for these very detailed and constructive overall comments. We clarified the relation of this manuscript to Sorg et al. 2015 better. Main focus of Sorg et al. was on dendrochronology, not photogrammetry, and no InSAR was used. The present study uses improved processing; more time stamps including more recent data; more rock glaciers; and combination with an InSAR kinematic inventory. We now added related text and comparisons at several places in sections 2 and 5.3.

*Specific comments:*

We agree with all specific comments below and implemented them, unless discussed.

*Page 1, Line 23: Please delete "very" in "very high resolution data".*

*Page 1, Line 23: Most of them reads three out of six. It would be very interesting to include an analysis of the regional scale evolution of surface speed, which would largely improve the confidence in the general conclusions. This might be possible on the basis of the readily available InSar or optical data. The resolution of such an analysis might be lower compared to the one provided for the six study cases investigated in detail. If this analysis is not done, it should be made clear in the text that the conclusions are still rather speculative and based on preliminary results and similitudes to other regions (Alps).*

Agreed, we made our conclusions more specific and balanced throughout the paper, or justified it better, respectively. We don't see a possibility to extent the time series to more rock glaciers, though. InSAR data extend not much back in time, and processing and combining many displacement rates from different sensors is a massive work. Processing more optical time series requires more modern data, which are very expensive, and more Soviet era air photos. The latter are very difficult to obtain, and we have no coverage over other rock glaciers available. Also, processing of these airphotos was very laborious as we didn't have access to the original raw data. We clarified that better in the revision. We also added a sentence in the conclusions making these different circumstances to studies from European Alps clear.

*Page 1, Line 26: The comparison between rock glacier and glacier sediment transfer is poorly constrained in the manuscript. Please provide more information and a context. Please add some relevant references from the literature, and extend the discussion. This might require some additional analysis and a quantitative regional-scale compari- son of the two contributions.*

We modified our conclusion about rock glacier sediment transfer, being more specific to what conclusion our study enables.

*Page 1, Line 27: The relation between the stress regime (compressive flow regime) and changes in speed upstream is mentioned, but only superficially discussed, without any reference to rock glacier dynamics and their mass fluxes.*

This aspect is modified in the main text, but removed from the abstract.

*Page 1, Line 28: The conclusion that the Gorodetsky Rock Glacier does not show an acceleration due to the decoupling from its rooting zone contradicts the previous point. It also partly contradicts the hypothesis that rock glacier acceleration is due to warming air and ground temperatures. The two mechanisms are not exclusive, but their interaction is not straight forward and requires a more detailed discussion in the manuscript (and possibly additional analysis).*

This aspect was removed from the abstract, and else in the manuscript.

*Page 1, Entire Abstract: I suggest to rewrite the abstract in a more concise and specific fashion, in order to highlight the novelty and the merit of the manuscript.*

Done.

*Page 2, Line 2: Rock glacier move due to the deformation of debris-ice mixtures (as you also mention several times later in the text). Please correct this first sentence, which now suggests that they move due to the deformation of the frozen debris only.*

Done

*Page 2, Line 3: Please correct the repetition: "rock glaciers, or (rock glaciers)." I can imagine that it is a typo for the one-word version "rockglaciers".*

Done

*Page 2, Line 10: Due to the ongoing debate about the genesis of rock glaciers and the origin of their constitutive material, I find the text insufficient for an introduction. I would avoid confusion and keep this topic out of the intro, also because it is not needed nor addressed further in the manuscript. If the authors believe that this is essential for their manuscript, I suggest that they extend the introduction and address the topic in more detail.*

Removed

*Page 2, Line 17: The concept of destabilization might be easily misunderstood and confused for structural instabilities. This being not the case, some additional explana- tion might be needed to present the concept of rock glacier destabilization. However, here again, I suggest to omit this point, because it is not needed in the manuscript. Alternatively, introduce it in more detail.*

Removed

*Page 2, Line 21: please add citations to Arenson and Springman (2005)a-b.*

Included

*Page 2, Line 25: please be careful when considering the different processes. The re- sponse of permafrost creep to increasing temperature is thought to follow a power law. In addition, the role of water can enhance, also in a non-linear fashion, the response of creep to temperatures approaching melting conditions.*

Specified

*Page 2, Line 26: please consider the following references, which are in my opinion amongst the most important publications regarding this topic: Ikeda et al, (2008), Buchli et al, (2018) and Cicoira et al, (2019)b.*

Included

*Page 2, Line 29: please specify "impacts".*

Specified

*Page 2, Line 30: inventorying rock glaciers is typically done on the basis of a combi- nation of optical and topographical data. Kinematic information (e.g. InSar) remains an optional and non-sufficient data source. Please be more precise in the text. The authors might refer to the*

*Baseline Concepts for inventorying rock glaciers which are currently being elaborated within the International Permafrost Association IPA.*

We specified. We think InSAR is an important method for that purpose, and one that will become even more important. The 2nd co-author is key member of the IPA/ESA process mentioned. We clarified that neither optical data and InSAR is sufficient alone. We would also like to point out that our statement was not about "inventorying rock glaciers" but about "inventorying rock glacier motion", no changed to "kinematics", the term used in the working group.

*Page 3, Line 19: please be more specific in the wording. "followed" is not a technical term and might be misunderstood. I suggest to use the key-terms: qualitative – similar patterns, statistical correlation, phase lag, thermal offset, non-linear.*

Removed

*Page 3, Line 20: The influence of temperature forcing through heat conduction on rock glacier dynamics has been quantitatively investigated in detail in e.g. Kääb et al, (2007) and Cicoira et al, (2019)a-b. The authors might want to discriminate between qualitative and quantitative studies that have investigated the processes controlling variations in rock glacier creep and include the state-of-the-art knowledge on the topic.*

We specified that we here mean the observational long-term data referred to in the sentence before, not theoretical considerations that are given 3 paragraphs above.

*Page 3, Line 21: The influence of variations in ground temperature through melt water advection has been shown to be negligible for the case of the Furggwanghorn in Buchli et al, (2018) and for the Ritigraben Rock Glacier in Cicoira et al, (2019). I am not aware of any study where the hypothesis (in the submitted manuscript) was tested on the basis of observational data nor modelling studies. If such study exist, please include the reference in the text in an explicit fashion. The study of Ikeda (cited in the text), also concurs to the hypothesis that rock glacier creep is controlled by variations in the effective stresses, rather than variations in ground temperatures (being these close to the melting point).*

We removed this part, rather referring to the treatment of the topic further up in the intro.

*Page 3, line 21: as a general comment, I see the need of general revisions of the text about the processes controlling rock glacier creep.*

Revised at places.

*Page 3, Line 24: At this point, the reader would expect temperature, precipitation and snow cover data to be analysed along the creep rates. I believe that the study presents enough new insights and does not need this additional step, but I suggest the authors to explain why it has not been done.*

We explain now that the focus of the study is on kinematics measurements, and that the meteorological data availability does not enable spatially and temporally detailed analyses.

*Page 4, Line 15: please specify the depth of the ground temperature measurements.*

Done

*Page 5, Line 7: please consider replacing "rock" with "boulder" in all the appropriate cases.*

Done

*Page 5, Line 8: add a point at the end of the sentence.*

*Page 5, Line 24: Please explain the reasons for the choice of the six rock glaciers.*

We clarified (previously investigated and suitable data available).

*Page 6, Line 1: I am not sure what "frozen snow" means. Please consider replacing this formulation with a more specific terminology.*

Will do. We mean non-melting snow and used the term "dry snow" as common in radar remote sensing.

*Page 6, Line 12: The choice of assigning a polygon to the next class when the velocities are close to its upper limit seems unjustified to me. Also, what is the reasoning behind the division for the first classes (0-2, 2-10)? The next two classes are (half) an order of magnitude, so I wonder why also the first two are not consistent.*

The reason for the classification in the higher class lies in the under-estimation of the surface flow velocity due to the 1d-LOS sensitivity of the InSAR data. We thus accounted for the correction factor to be applied. This is now clarified in the text.

The choice of the division between the classes 0-2 and 2-10 originates from the criteria for the determination of intensity in the "Guideline for the integrated hazard management of landslides, rockfall and hillslope debris flows" of the Swiss Federal Office of Environment. The IPA Action Group "Rock glacier inventories and kinematics" in the definition of its practical guidelines for rock glacier inventory using InSAR (kinematic approach), see https://www3.unifr.ch/geo/geomorphology/en/research/ipa-action-group-rock-glacier, also noticed that inconsistency and is now proposing a different division (< 1 cm/yr, 1-3 cm/yr, 3-10 cm/yr, ...). A revision of the rock glacier inventory in Ile Alatau and Kungöy Ala-Too is ongoing following these guidelines. The submitted work is based on the previous division and we will include a statement about the ongoing revision. The classification used has no impact on the conclusions from our work.

*Page 6, Line 13: please specify the nature of the mentioned variations. (spatial varia- tions?)*

Done. We mean temporal variations.

*Page 6, Line 15: is there a reason why the vector of the observed displacements has not been corrected according to topography? Maybe extend on this point.*

The reason that it was not applied (explicitly) in this case was the elaboration of a classified InSAR derived inventory based on expert decision. An a-posteriori calculation of the specific correction factors was not possible, since no date-interval and precise LOS velocity was recorded during the elaboration. One lesson learned was, that the additional effort in recording the actual scene that determined the velocity (sensor, date1, date2, and LOS velocity) was recorded for other studies. However, the vector was actually considered in the selection of the faster velocity class, when close to the class boundary (see your comment above to Page 6, Line 12:). Explained now in the text.

*Page 6, Line 23: is this sentence a list of the criteria used for identifying and locate the rock glaciers? Currently, the sentence is somehow lost in the text. Please consider rephrasing it.*

Rephrased:
"E.g. rock glaciers usually show an increase of the velocity to the inner region of the mapped polygon, have a lobe-structure and clear fronts."
To "E.g. rock glaciers usually show an increase of the velocity towards the front and inner region of the mapped polygon."

*Page 6, Line 24: short-term variations in rock glacier velocity have been observed at many sites worldwide. Seasonal and even weekly oscillations are observed con- sistently. The*

*statement is therefore unjustified. I suggest to support it with specific evidence for the study area (if available) or to discuss it in more detail. See Haeberli, (1985), Wirz et al, (2016), Strozzi et al, (2020).*

Clarified by focussing on the other criteria.

*Page 6, Line 25: it is now not clear to me weather the analysis has been conducted over multiple time steps. Maybe this has not been explained clearly enough in the text, or I just misunderstood it here.*

Temporal variations are included only implicitly since the InSAR data availability did not allow a thorough analysis of the temporal behaviour. Eg. when single early summer scenes show decorrelation due to faster movements and late autumn scenes show coherent and slower motion in the same extent, this can be qualified as seasonal variations. See previous revision comment.

*Page 6, Line 19-30: is this paragraph a list of the criteria used to classify a rock glacier and distinguish it from a rock glacier? Please be more specific and explain in detail the concepts that have been used. I suggest to reference to the ongoing action group on rock glacier inventories and kinematics of the IPA.*

Clarified. In this paragraph, some typical motion patterns determined from InSAR were pinned to specific landforms / surface motion processes. This is to extend the morphological approach based on the available imagery.

*Page 8, Line 11: please explain why the measure of accuracy was performed on stable ground only for three of the six field sites.*

We explained now and added data for the other rock glaciers for the last period. The other rock glaciers are surrounded by steep slopes where orthorectification DEM errors, shadows and other topographic effects make such test difficult systematically (and automatically) for a large number of points. The three rock glaciers chosen are surrounded by terrain with similar properties than the rock glaciers themselves and we view the test there to be representative for all the rock glacier surfaces.

*Page 8, Line 18-24: The results of this very interesting analysis are only briefly de- scribed in the manuscript. Also the discussion seem to me not sufficient to address this point. I suggest to include more details in the manuscript.*

Given the upper-limit length of the current manuscript we preferred to mention this aspect only shortly. We added now explanation and discussion at a number of places .

*Page 10, Line 10: please specify that only (part of ) the labelled rock glaciers are further investigated in the photogrammetric analysis, and not all the visible polygons in the figure.*

Done

*Page 10, Line 10: please consider indicating the value of the wavelength for the data in the figure.*

Done, assuming the referee means the Sentinel-1 C-band radar.

*Page 10, Line 10: it is impossible from the figure to distinguish between the different classes of the polygons. This information is present only in Fig. 1 at a very low reso- lution. Consider improving the level of detail in this (Fig. 2) or in the following figures (Fig. 3-7).*

The figure is completely revised, and mentioned that the inventory is available online.

*Page 11, Line 3: consider replacing "photogrammetric velocities" with a more detailed terminology. (such as "Surface velocities calculated by offset tracking").*

"Photogrammetric velocities" explained.

*Page 11, Line 4-12: the determination of the origin of the ice and sediment constituting the rock glacier requires more than the observation of spatial connection, or as in this case, the (legit) supposition of past spatial connection. The state of inactivity of the current glacier forefield (called in the text "zone between glacier and main rock glacier") only shows that the connection is not currently present, but is not sufficient to imply that this (dynamical- and sedimentological) connection was present in the past. Even in this case, it would have been limited to the period when the glacier advanced to its maximum (LIA). No information on the climatic and sedimentological setting of the rock glacier is provided in order to commence such an analysis. Without entering in too much detail, I suggest to limit the discussion to the spatial connection (glacier forefield connected, according to Delaloye and … 2018) and avoid speculation regarding the "nourishment" and genesis of the landform.*

Agreed, we modified accordingly and in line with above comment avoided touching the origin of rock glaciers.

*Page 11, Line 15: please provide quantitative evaluation of the observed trend and its statistical significance.*

We make now clear that our goal is not to derive linear or low-order "trends" but changes. Giving useful and reliable numbers about statistical significance for an overall trend over the entire time series is hard to obtain due to the small and uneven number of measurement years possible and the unknow trend model. We believe thus that error bars for the average speeds per time step and for the changes between time steps is a good way to quantify uncertainty.

*Page 11, Line 18: as above, consider removing the concept of "nourishment" from the paragraph.*

Done

*Page 11, Line 19: consider removing "striking" or replace it with a more technical and specific adjective.*

Replaced by "distinct".

*Page 11, Line 25: please describe in more detail the differences between the surface speed for this early period.*

We discovered an error in our Figure. 1953 image data actually cover only Gorodetsky, so that the time series for the remaining years for all three rock glaciers is similar. This error also has repercussions on the discussion of the high 1964-1971 speeds later in the paper, which was also corrected.

*Page 11, Line 27: I suggest to improve this paragraph and give a better summary of the results for this analysis. Also, write explicitly what the calculated ice-content would be. It is implicit in the ratio, but the reader might be helped by some repetition here. Consider calculating it for all the available time steps and include an estimation of the uncertainty in the results.*

The ratio between advance rate and surface speed is a function of ice content and vertical profile of horizontal velocities. We cannot disentangle these. The images available are not of sufficient quality to measure advance rates for every time step with useful accuracy. This is now clarified in the method section.

*Page 14, Line 11: In figure 4c, the debris-covered glacier and (for what I can see) the glacier forefield are not shown and it is impossible to verify the presence of a material flux from the rock wall to the glacier. As previously, I consider the observational evidence insufficient to support the statement. I suggest to simply avoid the point, which is in my opinion not important for the present manuscript, or to include more data and analysis to support this thesis.*

We will avoid the point. Fig 4c shows the entire system, as does Fig 5a, which is the idea behind these panels. Fig 5c shows only parts of the uppermost part. Else, we removed this aspect.

*Page 14, Line 24: the deformation profile (on the vertical dimension) also has an im- portant influence on this calculation. Why is it not mentioned here? Please consider spending some words about this point to make the text clearer.*

Yes, agreed, see above comment. But if the ratio is close to 1 then there has to be little decrease of speed with depth and little ice content. Explained now in the methods. Note also that the ratio we calculate is based on two measurements and free of any hypothesis regarding velocity profile or ice content.

*Page 15, Figure 5: it would be interesting to see the displacement on stable terrain.*

Stable terrain displacements are shown in Fig. 5c, but not included in 5b for the reasons given above. 5c reflects that measurements around the rock glacier are much more difficult than on it and of limited use. Numbers for 2009-2017 now given and error bar added.

*Page 16, Line 2: what is the mass flux at the boundary between the glacier and the rock glacier? What is the absolute value of the surface speed? It is very hard to see this from the figure provided. In general, a similar comment as for the nourishment above.*

Numbers of speeds between glacier and rock glacier now given. Nourishment concept avoided.

*Page 18, Line 15: here the authors state that the sediment transfer between glacier and rock glacier cannot be determined with the available data. I agree with the conclusion, but still, I would like a more quantitative discussion. Otherwise, I suggest again to completely avoid this point. A possible analysis would investigate the relation between acceleration and max fluxes along flow lines.*

Removed to avoid the point.

*Page 18, Line 15: more information about the vegetation could be interesting. What is the size and what are the species growing on the rock glacier? Is this information available from field expeditions?*

The only information available to us is the one given in the cited Sorg et al. 2015. This study was about trees, not vegetation in general, though.

*Page 18, Line 21: the fact that the observed signal from feature tracking is lower than the noise does not allow to conclude that the rock glacier has accelerated. I don't understand why the authors mention "statistical significance" in their argument.*

We negated this statement, "cannot … be considered … significant". We reformulated to make clearer.

*Page 22, Line 1: I agree with the statements, but I would like a more quantitative evaluation of the different error types.*

We added now more numbers here, and for the individual rock glaciers in the result section.

*Page 23, Line 2: I agree with the authors. Still, it would be interesting to see the values of the errors on stable ground.*

See several above responses on this topic. We believe we have now provided a thorough error budget assessment and discussion, exceeding what is typically done in comparable studies.

*Page 23, Line 7: the divergence of a vector field is a well defined term and as far as I understand this is not what the authors mean. I suggest to replace this substantive with a more pertinent description of the differences observed in the velocity field.*

We replaced the term by "spreading out".

*Page 23, Line 10: I have not found any values for the statistical analysis of the velocity time series. I warmly suggest to implement this analysis in a quantitative way. If the authors prefer not to, I would be much more careful talking about statistical significance.*

The number of measurement years available (and in particular when talking about the speed increase during recent years) is too low to do meaningful trend analyses, we suggest. We clarified now that we refer to the amount of speed increases relative to their error bars.

*Page 23, Line 25: it would be valuable to have a better quantification of the "strong compressional regime" by means of e.g. strain rates (accompanied by proper interpre- tation or even with the calculation of the internal stresses).*

*Page 23, Line 27-29: First, it would be very welcome to see the original results for the upper part. Second, I don't agree with the statement that the lower part responds passively. It is a matter of dynamics, thus of mass and momentum fluxes. I suggest*
*to change the formulation, possibly mentioning that the dampened response (I would appreciate an illustration of this) is due to topographical setting and the correspond- ing dynamic behaviour of the rock glacier. (In detail it could be that the mass flux is mostly compensated by variations in thickness or in mass input rather than variations in velocities, but such a statement should be supported by more evidence.)*

Given the large errors bars, we realized that a comparison of our results to Gorbunov remain speculation, and removed this part of the paper. We also revised the discussion of "dampened" response. We prefer to avoid discussions and speculations that lead too far away from the focus of our paper on the measurement of kinematics and speed time series. The order of strain rates was added. As we removed much of the comparison with Gorbunov, we don't add the weakly defined speeds from the middle part in the paper, but show it just below for information.

[Figure]

[Figure]

Figure: As Fig 5b of the manuscript, but for points further up the rock glacier. Horizontal green bars are the median, blue bars the mean speeds.

*Page 23, Line 29: please repeat the advance rate and the surface velocity, and con- sider discussing the result in more detail – also with a possible range of quantitative values of the ice content.*

Advance rates added. For the reason given above, the lack of possibility to disentangle ice content and vertical profile of speeds, we prefer to not discuss scenarios of these two components.

*Page 23, Line 32: given the strong similarity between the two publications, this point might require some more discussion. I would suggest also one or two figures in the appendix or some additional comparisons.*

We don't think these two publications are "strongly similar" (see also response above). This was now made clearer in section 2, and the text here was revised accordingly.

*Page 24, Line 10: If I understand this correctly, it means that it is not possible to conclude which one of the two studies is more accurate. If this is the case, please state it more clearly in the text.*

We meant the new study is more accurate. The statement about other measurements to compare with is not related to the difference between this study and Sorg et al. We reformulated.

*Page 24, Line 24: consider citing Cicoira et al., 2019b, where it has been shown that the seasonal and inter-annual variations in rock glacier flow are mostly controlled by variations in snow melt rates and liquid precipitation, rather than in air temperature. Other very relevant citations are Buchli et al., 2018 and Ikeda et al., 2008.*

Done.

*Page 25: in general, in the "5.3 Speed time series" paragraph, more discussion relative to the results and their validity would enhance the validity of the manuscript. Most of the discussion is a very precise comparison to Gorbunov et al., (1992), which could be probably summarized in one or maximum two paragraphs. I suggest to highlight more the originality of the manuscript and discuss better its strength and weaknesses.*

We modified accordingly.

*Page 25, Line 26: I don't see the link the negative glacier mass balance. Please avoid this point or argument in more detail the linking mechanism.*

Removed.

*Page 25, Line 29: it would be very interesting to quantify this sediment transfer. I suggest using a simple assumption for the rock glacier thickness (e.g. constant value of 20 meters, see Cicoira et al., 2020) and estimate the overall ice/sediment transport rates for the periglacial environment. This would be a major result, and is not very difficult to calculate (although the uncertainty will be large).*

We added such an order-of-magnitude of magnitude estimate for the large/fast rock glaciers in the region, but prefer to not extrapolate our time series to all rock glaciers, in particular the smaller ones.

*Page 25, Line 31: such a statement definitely requires a quantification of both the sediment transfer.*

We formulated more careful.

*Page 26, Line 4: this point is very interesting but insufficiently discussed. I suggest the authors to implement it both in a qualitative and in a quantitative fashion.*

We added more text here and for the individual rock glaciers.

*Page 26, Line 5: This statement appears unjustified to me. As far as I know, no quantitative (and conclusive) evidence that a rock glacier derived from a glacier exist. As for this manuscript, there is not sufficient evidence supporting the statement. Often, an interaction (more or less important) has happened during the LIA. I suggest to rewrite this last paragraph with more focus on the novelty of the manuscript (it is not the geomorphological genesis of the rock glaciers).*

Entire paragraph removed.

*Page 26, Line 18: it is not so clear to me which assumptions. Please be more explicit in the conclusions.*

Clarified.

*Page 26, Line 26: the time scale considered in the manuscript is (almost) only decen- nial. I would rephrase this sentence and highlight the fact that the original observations in speed also show past periods of acceleration at the investigated temporal scale.*

Reformulated.

*Page 27, Line 4: I am not completely convinced by this statement. The quantification of the trends and their significance for each rock glacier might make this point more convincing and increase the confidence in the results and the conclusions.*

Entire paragraph removed.

*Page 27, Line 10: I don't agree at all with this statement. This is very speculative and not supported sufficiently by the evidence provided in the study. As above, I suggest to discuss it in more detail or to avoid this point, which is in my opinion not relevant for the manuscript.*

Removed

*Page 27, Line 10-15: on the contrary I welcome the topic as an outlook for future studies. With this last comment, I thank the authors for an interesting piece of research.*

Rewritten.

*References:*
*Arenson, L.U., Springman, S.M., 2005a. Mathematical descriptions for the behaviour of ice-rich frozen soils at temperatures close to 0âŮçC. Can. Geotech. J.42, 431–442. https://doi.org /10 .1139 /t04 -109.*
*Arenson, L.U., Springman, S.M., 2005b. Triaxial constant stress and constant strain rate tests on ice-rich permafrost samples. Can. Geotech. J.42, 412–430. https://doi .org /10 .1139 /t04 -111.*
*Cicoira, A., Beutel, J., Faillettaz, Vieli, A., 2019b. Water controls the seasonal rhythm of rock glacier flow. EPSL 528, https://doi.org/10.1016/j.epsl.2019.115844*
*Cicoira, A., Marcer, M., Faillettaz, J., Gärtner-Roer, I., Bodin, X., Arenson, L. U., Vieli, A., 2020. A general theory of rock glacier creep based on in-situ and remote sensing observations. PPP (in review).*
*Delaloye et al., 2018. Rock glacier inventories and kinematics: a new IPA Action Group. EUCOP5.*
*Buchli, T., Kos, A., Limpach, P., Merz, K., Zhou, X., Springman, S.M., 2018. Kine-matic investigations on the Furggwanghorn Rock Glacier, Switzerland. Permafr. Periglac. Process.29, 3–20.*
*Haeberli, W., 1985. Creep of Mountain Permafrost. Mitteilungen der Versuchsanstalt für Wasserbau, Hydrologie und Glaziologie der ETH Zürich, vol.77.*
*Ikeda, A., Matsuoka, N., Kääb, A., 2008. Fast deformation of perennially frozen debris in a warm rock glacier in the Swiss Alps: an effect of liquid water. J. Geophys. Res., Earth Surf.113. https://doi .org /10 .1029 /2007JF000859 .f01021.*
*Wirz, V., Gruber, S., Purves, R.S., Beutel, J., Gärtner-Roer, I., Gubler, S., Vieli, A., 2016. Short-term velocity variations at three rock glaciers and their relationship with meteorological conditions. Earth Surf. Dyn.4, 103–123. https://doi .org /10 .5194 /esurf -4 -103 -2016.*

Referee #2     Philippe  Schoeneich

*It is a very high quality paper, written by some of the best specialists of the topic and of the methods used. The paper provides a valuable and comprehensive dataset, on an area for which few data were available so far.*

*The work presented uses a combination of various methods and various types of optical and satellite imagery. A combination that was rarely achieved at such a level in a single study. This allows a  cross-checking and validation of the results, which therefore appear as very robust. The use of   image archives allows a back-analysis from the 1950s onwards and the interpretation of the  evolution of velocities over time. If this is not new, the above mentionned combination of various  image types and methods allows a higher and more detailed temporal resolution, and reveals short  lived velocity changes that could not be observed by using a single type of images.*

*The paper provides therefore an innovative contribution both for the results and the methodological approach.*

*Therefore, there are no fundamental comments, and the paper should be accepted with the minor improvements listed below or proposed by others.*

*p. 4 – l. 6 « landslides and rock avalanches » : there is a problem of vocabulary. In English, the  word « landslide » is used as generic term for all types of mass movements on slopes, and thus  includes rock avalanches. The authors probably wanted to distinguish slides (= « Rutschungen », « glissements ») from rock avalanches. The adequate term in this case is « slide ». If they mean  deep seated slope movements (= « Sackungen », « Talzuschub », « tassements », « glissement  rocheux ») the most adequate concept would be DSGSD (for Deep Seated Gravitational Slope  Deformation). As generic term for designating all mass movements, we recommend to use « mass  movements » instead of « landslide », the latter term causing much confusion among french or   german speaking readers.*

Changed to mass movements.

*Fig. 1 + part 4.1 : the figure legend indicates 1 « landslide » (see also comment above), but this is  not mentionned in the text.*

Figure changed now to show rock glaciers and all other movements as summary category.

*Fig. 1 is small and hardly lisible. It should be provided in full format as downloadable  supplementary data*

We will make sure the figure is available in high-res in the paper. The inventory is also available from an open archive as now also mentioned in the caption. See also comment from Ref #1.

*Figures 3, 4, 5, 6 and 7 : the velocity color scale is not the same in all figures, which hinders a direct comparison of the different cases and can lead to misinterpretation. In figure 4 for instance, at a first and quick view, the Morenny RG could appear as slower than the Archaly RG, which is not the   case. A common velocity color scale would be better.*

We believe the velocities are too different among the rock glaciers to make a common scale useful. With a max of 6.5 m/a many details wouldn't be recognizable anymore for the slower rock glaciers. We included a figure in the supplement with equal colour scales instead.

*p. 24, l. 20-25 : you mention the influence of snow thickness/insulation and amount of meltwater as factors possibly explaining the acceleration in the 1960s. I agree with this general statement, but it  can be refined. The data series of the Laurichard rock glacier (see Bodin et al. 2009, you already  have in*

*your references) and ground surface temerature measurements at many places, show that the   most relevant factor is the onset date of the snow cover. An early onset, for instance in October,   before the coldest days of November-December, prevents the ground from cooling and keeps the   accumulated summer heat in the ground. On the opposite, a late onset of the snow cover, by end of December of even January, will allow a strong cooling of the ground surface and a deep seasonal   frost in non permafrost areas.  The Laurichard time series shows that the reaction is assymetric : it   needs several « warm » years (years allowing a warming of the ground, with hot summer and/or   early snow cover) to induce an acceleration, but a single « cold » winter (with late onset of the snow   cover) is sufficient to reduce velocities. Do your meteorological data allow to establish the onset date   of the snow cover (insulation is provided with a minimal thickness in the order of 60-80 cm) ?        If yes, it could reinforce your interpretation.*

Data on snow onsets in the 1960s are not available to us. Also, it seems Ref 1 and 2 don't agree what could cause rock glacier speed changes (see comment on same topic from Ref 1). We formulated now more open including both possibilities.

*Author contributions : there are two AK among the authors ! Who is AK ? And what was the contribution of the second one ?*

Good point, thanks for spotting! We specified.

*Acknowledgements : the two first lines are duplicated in the Financial support section. Remove  from here, and leave under Financial support.*

Revised

*References*

We fixed issues. Thanks for checking the details!

*Check formatting of abridged journal names :*
*-   for PPP : your text mentions Permafrost Periglac. (Processes is missing) – numerous references*
*-*
*-   p. 35 Kääb et al. 2017 : NHESS (Sciences missing)*

*p. 33 - Gorbunov 1983 : volume missing*

*p. 33 - Haeberli 1985 : volume number in the collection missing*

*p. 34 – Kaufmann 2012 : journal missing*

*p. 36 – Roer et al. 2008 : incomplete (volume, pages, ...)*

Referee #3     Line Rouyet

*Kääb et al. inventoried active rock glaciers and other periglacial landforms in Tien Shan using InSAR and further analyzed the spatial-temporal patterns of six fast-moving glacial-derived rock glaciers using optical offset tracking. They present a nice piece of work that shows the value of radar and optical remote sensing for mapping and moni- toring creeping permafrost landforms over a large and hard-to-access areas. The study focuses on a region that had not been intensively documented before. The contribution is well suitable for a publication in The Cryosphere and likely to become a reference in remote sensing of mountain permafrost.*
*I have no major concern regarding the way the study has been designed but I think the manuscript could benefit from some clarifications about the inventory procedure, a better visualization of the results and an extended discussion on the findings. Here I develop three main elements. I also listed some complementary suggestions of cor- rection at the end of the review.*

*–Main comment 1. Inventory and movement classification–*

*For the movement types, I think it should rather be named 'landform types' as e.g. a rock glacier or a moraine is not a process but a landform that has been shaped by a periglacial or glacial process. Consider also renaming 'solifluction / debris movement' (not sure what debris movement means; a rock glacier is also composed of debris. Scree deposit?). 'Dead ice / subsidence' could just be 'thermokarst'?*

Fig. 1 is now completely changed. Movements other than rock glaciers are now only summarized as one category.

*About the velocity classes in the inventory: The choice of the velocity classes needs at least to be explained. 0-2, 2-10, 10-50, 50-100, >100 does not look so intuitive to me.*
*Why not equal intervals: e.g. 0-25, 25-30, 50-75, 75-100. Or 0-3, 3-10, 10-30, 30-100, >100? Or just an order-of-magnitude (cm/yr, cm-dm/yr, dm/yr, dm/yr-m/yr, m/yr), as now recommended by the IPA action group 'rock glacier inventories and kinematics'? The criteria for defining the class are also not fully clear: when writing that 'if two or more classes were present during the observation time-span, the higher displacement rate was used to determine the velocity class' (l.14-15, p.6), does it mean that a little fast-moving part of the rock glacier could lead to classify the whole landform as for ex. >1m? If so, that sounds a bit bold to me.*

Agreed, that would indeed lead to a false representation of the actual kinematics. Usually, when a classification was done, a "representative" part of the polygon must show similar speeds. Usually small spots within the polygon with significantly higher velocity (such as accelerating or failing fronts) were either ignored or outlined separately. Therefore, from this direction, no over-estimation is expected. We rewrote now parts of the text and also included explanation about the choice of classes (the work was done before the IPA working group).

*Maybe it should just be acknowledged in the discussion that it could lead to a rate overestimation. Similarly, at l.3-4 (p.7), it is acknowledged that the max. measurable rate for a C-band with 12d repeat-pass is 85 cm/a. To my understanding, a decorrelated part in a 12d-interferogram means >85 cm/a, but so not necessarily >100, right?*

From the calculation point of view, 85 cm/a is seen as 2.8 cm in a 12 day S1 interferogram. But in reality, given the size of the area under motion, this could be recorded coherently, when fringe visibility is good. This value even can be exceeded.
Therefore reason 1) for classification in the > 100 cm class is, that motion decorrelation usually is under higher displacements as the minimum value. And 2), the topographic correction factor is in realistic cases not lower than 1.1-1.2, making a LOS displacement rate of 85 cm/a rather belonging to a landform of > 100 cm/a class. This was now added in the method section. However, in the work

carried out, we did not further group the decorrelated (and probably exceeding 100 cm/a) from the other "undefined" class.

*In these cases, were the 1d ERS interfero- grams used to ensure a correct classification? If yes, it could just be mentioned.*

The few good 1-day ERS interferograms we had, did not show good signatures. Probably because due to the main temporal coverage from March-Early June, with snow melt and other snow-related changes. So, from these data no additional information could be drawn unfortunately. Mentioned now in the text.

*–Main comment 2. Figures–*

*Most of the figures could benefit from slightly more job to clearly disseminate the find- ings. Considering the volume of work that the study represents, it is a bit a pity if the presentation does not fully help to maximize the understanding of the results.*

*Figure 1: This map is hard to read. I understand the wish to be comprehensive, but it is most likely too much information in once and the color contrast does not allow for differentiating the velocity classes for specific landforms. I would suggest making two maps: one with only the different movement types, without any velocity classes. And another focusing on rock glaciers (the main point of the article) with a different color for each velocity class (for ex. green to red, blue to red). A similar map for the other landform types can potentially be placed in Supplementary.*

We revised the figure as follows: Two-panel figure with 1) two color schemes (e.g. black / gray) for rock glacier / non rock glacier types. And 2) a color map for only the rock glacier classes in overview.

*Figure 2: Add the equivalent in cm of the phase color scale. Missing information about SAR geometry (LOS arrow or mention of the ascending/descending geometry in leg- end).*

Done

*Figures 3-7: It would be more comfortable for the reader if the optical images always had the same extent and scale than the velocity field maps. The error bars (especially the red) are hard to see due to the transparency. The arrows of the photogrammetry results are sometimes impossible to see and interpret (Fig. 5-7). Less importantly: it could be nice to have altitude references at some locations on the optical images.*

We prefer to have the optical images giving the full overview and the velocity figures to zoom in. We indicated the velocity zooms in the overview images. We will make sure the figures are in high-res to give the reader the possibility to zoom in. We improved the error bars. We added elevation points, good idea!

*Figure 8: Add the equivalent in cm of the phase color scale. Missing information about SAR geometry. Here, as the background map from the interferogram is not directly related to the size of the arrows (contrary to Fig. 3-7), it would be good to add a legend of the arrow length. It is a really nice figure to show the agreement of both methods, as well as the value of combining them. What about adding the same for the five other rock glaciers (potentially in Supplementary)?*

We modified the figure, but did not add an arrow legend as we think estimating the arrow length from figure is difficult. The arrows are linearly scaled and we added maximum speed in the legend. We added similar figures for the other rock glaciers in the Supplement.

*–Main comment 3. Time series and acceleration–*

*There is something that does not completely add up in the interpretation of the temporal variations: contradictory information that gets the reader a bit lost at the end of the paper. The title includes 'acceleration of rock glaciers' and it is presented in the abstract that five of the six temporally investigated rock glaciers exhibit acceleration (l.23-24, p.1). However, at several locations, the authors acknowledge that the statis- tical significance is not always good enough to confirm the trends (several cases with std dev of speed differences greater than the actual measured difference, e.g. Fig. 7c). As explained at l.28-29 (p.24), 'Gorodetsky, Morenny and Archaly rock glaciers [...] are not showing a clear increase of speed over the study period'. Gorodetsky, Moreeny and Archaly are three out of six rock glaciers, which does not fit anymore with the findings' summary in the abstract (l.28, p.1) or the conclusion (l.4, p.27). Maybe I misunderstood something, but in that case, it is probably possible to be clearer in the text.*

Ref #1 raised similar concerns. We revised at several places.

*The statement 'the behaviour observed in the study area confirms findings [...] that rock glacier creep speed increase overall under atmospheric warming' (l.23-24, p.25) sounds a bit too bold to me considering that no comparison with temperature is pro- vided in the study. Other elements could be discussed in greater details. In 5.4, the authors remind that 'all rock glaciers studies in detail in this study are derived from contemporary or former glaciers' (l.5, p.26). Some fuzzy thoughts here: When did the glacial direct connection expected to have stopped? During the time span of the study (50th-70th)? If yes, are some of the variations related to the glacial dynamics instead of the permafrost creep, or due to the transition between the two flow types? What if you had also analyzed talus-derived and slower landforms? Not to say that it should be done here, but it could be discussed as a prospect. Some of the conclusions here are maybe only valid for glacial and extreme (>1m) cases.*

Yes, also this concern overlaps with Ref #1 and we revised, mostly by avoiding the discussion about glacial influence, as suggested by Ref 1.

*I don't think this comment requires critical changes. The results are well described in Section 4, but the discussion could be improved (especially 5.4) and the ab- stract/conclusion (and even the title?) better matched the actual findings.*

We agree and revised accordingly.

*–Complementary  comments–*

All agreed and revised/clarified.

*Page 1:*
*-        Title: 'acceleration'. Maybe consider 'Inventory and spatial-temporal trends of...' See main comment 3.* Changed

*-        l.23 and l.28: 'most of them'... 'The only rock glacier'... Maybe just write the actual number in both cases: five of them – the only, if this is really the case. See also main comment 3.* Revised

*-        l.30: maybe missing a general sentence in the abstract about what does it tell, what is*

*the relevance of the study in a broad sense?* Added

—

*Page 2:*

- *l.3: rock glaciers (or rock-glaciers) I guess.* Revised

- *l.9: 'In a similar way, the ice content […] can be directly incorporated…'* Revised

- *l.19: As rock glacier distribution is also a valuable environmental indicator, I would suggest removing the four first words of the sentence.* Revised

- *l.21: '…although it is also influenced by…'* Revised

- *l.22: '…temporal and spatial variations of ice content…' I guess lateral variations are as important as vertical?* Revised

- *l.25: '…variable around thawing conditions'. Maybe 'sensitive' instead?* Revised

- *l.25: '…i.e. in cases the ice content starts to degrade…'* Revised

- *l.29: 'impacts from spatio-temporal variations'. Impacts on what in this context?* Added

—

*Page 3:*

- *l.3-5: Start of the sentence ('in addition to …,') does not sound necessary to me. Instead, maybe: 'The categorization of landforms into classified movement rates or orders of magnitude (e.g. cm/day, ….)'* Revised

- *l.8: (more details below) is not really useful.* Removed

- *l.17: 'constant speeds' instead of 'stable speeds'?* Revised

- *l.19-20: repetition in these two sentences (creep velocity <-> temperature)* Revised

- *l.24: 'for the first time' sounds a bit weird considering the extensive literature presented in review 2 and reference to Sorg et al. article. What about 'The aim of the present study is to systematically inventory contemporary rock glacier crepp velocities…'* Revised

- *l.27-30: last two sentences are maybe not necessary.* Revised

—

*Page 4:*

- *l.5-6: potential to rephrase here 'with the largest of magnitude8.0 and more […] and stronger earthquakes…' Stronger than 8 and more? …* Revised

- *l.9-10: Sth unclear with that sentence: '…and for elevations of about 3000 m asl.' Does it mean 'For elevations of about 3000 m asl, mean annual precipitation exceeds 1000 mm on the northern slopes and is less than 800 mm on the southern ridges.'?* Revised

- *l.18: larger than what? '…permafrost is very likely although the fronts of the large rock glaciers are located…'* Revised

- *l.25: 'inactive ones': maybe wise to define here the activity categories. What about relict landforms? Are they considered in the study? In the activity morphologically determined or using InSAR/photogrammetry?*

No, only patterns with some sort of visible motion are included in this study. We added this info. However, the sentence of concern refers to an existing study that uses the term 'inactive'. We hesitate to re-interpret their terminology, also as we don't investigate non-moving landforms.

- *l.33: 'Blöthe et al. (2019) mapped...'* Revised

—

*Page 5:*

- *l.7: 'Displacement rate of individual blocks vary greatly...'* Revised

- *l.9: 'blocks'? Replace 'down-valley' by 'slope direction'?* Revised

- *l.17-18: New sentence at 'Rock glacier activity/tree growth anomalies...'? Respectively? To what? Not clear.* Revised

- *l.21: 'different statuses of the rock glaciers on their path towards...' -> 'different stages'?* Revised

- *l.25: remove 'for the offset tracking work realized here.'* Revised

- *l.30: 'short spatial baseline and time intervals between 1 day and three years...' According to your table A1, the max. interval is over one year.* Revised

—

*Page 6:*

- *l.5: I guess $\frac{1}{4}$ of a phase cycle' will be abstract for most of the readers. Maybe add what it means in terms of displacement?* Added

- *l.9: I think that 1-2 sentences could added here to quickly explain the principle. Count the fringes on coherent parts and relate it to velocity considering the wavelength in- terval between pairs + use decorrelation information. In the paper, you show only 12d Sentinel examples, but it is important to be clear that you used a large stack with different wavelengths/intervals/periods to get different detection capability and avoid misinterpretation of short-term velocity variations. Someone who does not know the technique could misunderstand if not quickly explained here.* Added

- *l.11: Why these classes? See main comment 1.* Explained
- *l.14: 'the highest displacement rate...' See also main comment 1.* Revised

- *l.15-16: 'We empirically considered the line of sight in the assessment': It could be explained what it practically means. If the slopes significantly deviating from LOS and between two classes, the highest cause it is considered as underestimated?*

We clarified, see above.

- *l.16-18: last sentence of paragraph could be moved after first sentence of the next one (l.20).* Revised

- *l.25: Check this sentence. I doubt you get coherent information on supraglacial ponds.* Revised

- *l.27-28: Similar appearance to what? Check this sentence. I don't see why solifluction would give spot-like fringe patterns. Also: debris movement does sounds like right terminology. See main comment 1.* Revised

- *l.30: Redundant. Remove 'also optical images were not of help due to'.* Revised

- *l.32: Consider using everywhere 100 cm/yr instead of 1 m/yr, as in Fig.1* Revised

—

*Page 7:*

- *l.2-3: >85 could also mean 90, which is in 50-100 class. See main comment 1.*

As stated above, it is unlikely that a decorrelated (in a 12 day interferogram) area will end up in the 50-100 cm/a class. Text added.

- *l.30 and 32: 'the above': not necessary.* Revised

—

*Page 8:*

- *l.4: thresholds on correlation coefficients -> the actual values of the thresholds could be documented.* Revised

- *l.12: the point (ii) sounds odd to me: looking at the chosen areas on the figures, they are significant velocity variations that are likely to be natural (for ex. C on Fig.6). Not possible to select areas with more similar patterns? What about the correlation*

*coefficients, the average correlation cannot be used as an accuracy estimate?*

We added text in sections 3.2 and 5.2 to explain the choice of measure (ii) and discuss the choice of correlation coefficient as uncertainty measure. Smaller clusters give larger homogeneity at the cost of a smaller number of measurements. Given the low matching accuracy based on the low-quality air and satellite photos, a larger number of points turned out to be more important based on initial tests we did.

—

*Page 10:*

- *Legend Figure 2: 'Wrapped orthorectified interferogram'? Raw sounds weird, and not especially right considering the corrections described in Section 3.1.* Revised

—

*Page 11:*

- *l.5: 'agree well with the interferograms'. Plural? To emphasize that several interferograms have been investigated.* Revised

- *l.20: 'they show coherent fringes'* Revised

- *l.24: 'the central part of Morenny rock glacier speeds are in the order of 1 m/a: It does not look like that in Fig.4, most of the rock glacier is blue and the averaged time series show values clearly under 1, even for the highest parts of the series. . .* Location described.

—

*Page 14:*

- *l.16: see main comment 1: in theory, noise could also mean 90 cm/yr, so can we really say consistent?* Described now in section 3.1, see also above.

- *l.24: 'This points to block movement of the entire rock glacier column. . .' Sounds odd, potential for rephrasing.* "Block movement" is a common term in glacier physics. Explained now.

—

*Page 16:*

- *l.6: ref. to Fig. 6e* Revised

- *l.12: 'because of the many different frontal parts' -> 'because of heterogeneities'?* Revised

—

*Page 18:*

- *l.19-20: I don't see the point of adding zone B if the results are not shown. Either the tseries are shown somewhere (potentially in Supplementary) or you maybe don't mention this part at all? About 'changes documented for zone B [..] are within their error bars': looks like there is the similar problem for A since the 90th (Fig.7 c).* We refer now more to zone B, and prefer to show where this zone is.

—

*Page 19:*

*Figure 7: Remove area B) if not used? See also main comment 2.* Kept (see above) but outline changed.

—

*Page 20:*

- *l.4: 'reasonably very' -> just reasonably?* Revised

- *l.6-7: 'all rock glaciers […] were classified correctly from the interferograms': does not sound right for Morenny (see comment page 11, l.24)* Specified above. Also Morenny is classified correctly.

- *l.22-23: see main comment 1: so 85 cm/yr is approximate as 1m/yr?* Explained now in detail in section 3.1

—

*Page 22:*

- *l. 2: missing space 'very-highresolution'* Revised

- *l.4: 'more difficult to quantify precisely': Sounds weird cause I don't think biases a) have been quantify in the manuscript.* Revised

- *l.6: 'could translate 1:1 to a lateral offset': without being an offset-tracking expert, I must say that I don't get this.* Explained

- *l.24-25: if median values are safer, why not have used these for the point clusters?* Explained

- *l.33-34: here comes quickly what I commented for p.8, l.12: maybe it could come before in the methods?* Text changed now.

—

*Page 23:*

- *l.23: 'acceleration between 1964-2017'* Now obsolete

- *l.26-29: make two sentences. For ex: the lowest rock glacier part deforms under a regime… It responds rather passively to…* Revised

—

*Page 24:*

- *l.5: 'the latter part, under a strong compressive creep regime, reaches and diverts the river…'* Revised

- *l.28: 'are the only systems'. 3 out of 6… sounds weird to say 'the only'. See main comment 3.* Revised

—

*Page 25:*

-        *l. 24: maybe 'is aligned with findings' instead of 'confirms'?* Revised

—

*Page 26:*

-        *l.11: 'an inventory of active rock glaciers and other periglacial landforms'? or 'a slope movement inventory including rock glaciers and other periglacial landforms'?* Revised

-        *l.16: well, maybe it is me being picky, but 85 cm is not a meter. And again, what about the use of ERS 1-day? See main comment 1.*

No area wide additional information could be derived from ERS 1-day interferograms. Actually, only 1 RG showed a motion signature that could be used. We explained in the text, see also responses above

-        *l.28: 'A possible explanation [...] is...'* We think this is correct and will leave to the language editing.

—

*Page 27:*

-        *l.2: '...more passively to...'* Revised

-        *l.4: Not in line with l.28-29 p.24. See main comment 3.* Obsolete

-        *l.10-12: this could be more discussed in 5.4. See main comment 3.* Obsolete, removed. See Ref 1

-        *l.12-15: The last sentence is too long and hard to understand. I would suggest as prospect to include not only glacial-derived and landforms with lower averaged velocity, to confirm the accelerating trend.* Obsolete, removed. See Ref 1

—

*Page 29:*

-        *Table A1: Why no 2017 Sentinel-1 image?*

Every project has its own history of definition, work execution, and revision. Within the ESA GlobPermafrost project we completed our first inventory based on 2015-2016 Sentinel-1 images. While writing the paper we revised the inventory and included some newly available Sentinel-1 interferograms from 2018 to clarify some open issues. These images were archived and with a very good quality, so no further data were necessary for this work. However, the ongoing revision of the rock glacier inventory in Ile Alatau and Kungöy Ala-Too following the IPA guidelines (see above) will include more Sentinel-1 images also from more recent years. And important, these revisions will have no significant impact on the conclusions of the present paper.